# A Survey of Agent Memory in the Second Half: Towards Self-Evolving and Long-Horizon Agents

**Wei-Chieh Huang**[1,†], **Weizhi Zhang**[1,†,‡], **Yueqing Liang**[2,†], **Yuanchen Bei**[3,*], **Yankai Chen**[1,*], **Tao Feng**[3,*], **Xinyu Pan**[4,*], **Zhen Tan**[5,*], **Yu Wang**[14,*], **Tianxin Wei**[3,*], **Shanglin Wu**[6,*], **Ruiyao Xu**[7,*], **Liangwei Yang**[20,*], **Rui Yang**[3,*], **Wooseong Yang**[1,*], **Chin-Yuan Yeh**[1,*], **Hanrong Zhang**[1,*], **Haozhen Zhang**[8,*], **Siqi Zhu**[3,*], **Henry Peng Zou**[1], **Wanjia Zhao**[19], **Song Wang**[9], **Wujiang Xu**[10], **Zixuan Ke**[20], **Zheng Hui**[11], **Dawei Li**[5], **Yaozu Wu**[13], **Langzhou He**[1], **Chen Wang**[1], **Xiongxiao Xu**[2], **Baixiang Huang**[6], **Juntao Tan**[20], **Shelby Heinecke**[20], **Huan Wang**[20], **Caiming Xiong**[20], **Ahmed A. Metwally**[21], **Jun Yan**[21], **Chen-Yu Lee**[21], **Hanqing Zeng**[22], **Yinglong Xia**[22], **Xiaokai Wei**[23], **Ali Payani**[24], **Yu Wang**[25], **Haitong Ma**[12], **Wenyada Wang**[8], **Chenguang Wang**[15], **Yu Zhang**[16], **Xin Wang**[17], **Yongfeng Zhang**[11], **Jiaxuan You**[3], **Hanghang Tong**[3], **Xiao Luo**[4], **Yizhou Sun**[18], **Wei Wang**[18], **Julian McAuley**[14], **James Zou**[19], **Jiawei Han**[3], **Philip S. Yu**[1,¶], **Kai Shu**[6,¶]

[†] *Co-first authors* [*] *Core contributors (listed alphabetically)* [‡] *Project organizer* [¶] *Core supervisors*

[1] *UIC* [2] *IIT* [3] *UIUC* [4] *UW–Madison* [5] *ASU* [6] *Emory* [7] *Northwestern* [8] *NTU* [9] *UCF* [10] *Rutgers* [11] *Cambridge* [12] *Harvard* [13] *UTokyo* [14] *UCSD* [15] *UCSC* [16] *TAMU* [17] *UCSB* [18] *UCLA* [19] *Stanford* [20] *Salesforce AI Research* [21] *Google* [22] *Meta* [23] *Roblox* [24] *Cisco Research* [25] *Capital One*

**Reviewed on OpenReview:** *https://openreview.net/forum?id=XycbogUAeJ*

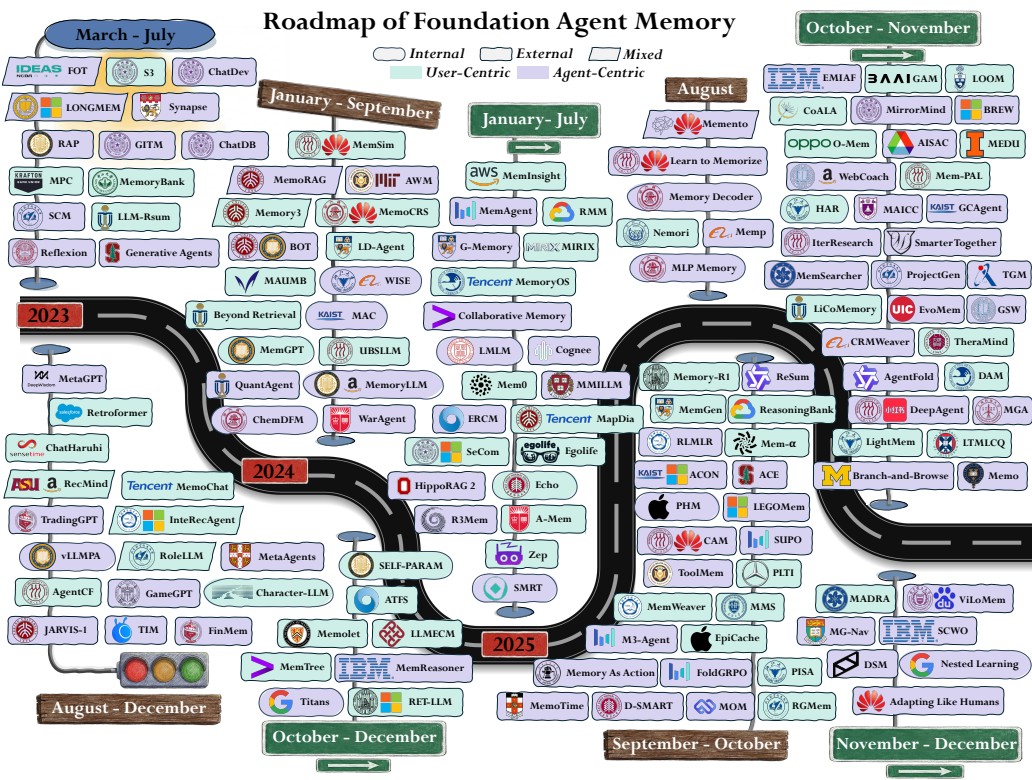

**AgentMemoryWorld/Awesome-Agent-Memory**

Figure 1: **Roadmap of Foundation Agent Memory**. A timeline illustrating the trend of foundation agent memory frameworks released between 2023 and 2025, categorized along two of the three taxonomy dimensions: memory substrates (internal, external, or mixed) and memory subjects (user- or agent-centric).

# Contents

## Abstract

Research in artificial intelligence is undergoing a paradigm shift from prioritizing model innovations and benchmark scores towards emphasizing problem definition and rigorous real-world evaluation. As the field enters the "second half," the central challenge becomes real utility in long-horizon, dynamic, and user-dependent settings such as agentic coding, deep research, and computer use, where LLM-based agents face context explosion beyond fixed context windows and must continuously accumulate, manage, and selectively reuse large volumes of information across extended interactions. Memory, with hundreds of papers released in 2025, therefore emerges as the critical solution to fill the utility gap. Beyond serving as passive storage, memory is increasingly the substrate through which agents self-evolve: short-term memory gates which experiences are perceived, selected, and abstracted during execution, while long-term memory accumulates and consolidates them into reusable knowledge and skills, forming the loop through which agents improve from their own experience and sustain continual learning. In this survey, we provide a unified view of foundation agent memory along three dimensions: *memory substrate* (internal parametric state and external retrieval-augmented stores), *cognitive mechanism* (sensory, working, episodic, semantic, and procedural), and *memory subject* (user-centric personalization and agent-centric experience). We then analyze how memory is operated under single- and multi-agent topologies and highlight learning policies over memory operations, showing how memory management itself is becoming a trainable, self-evolving capability that spans reinforcement-learned context curation, experience consolidation at decision time, and the emerging ecosystem of explicit, portable, and shareable agent skills surfaced through agent harnesses, context engineering, and standardized tool-mediation protocols. Finally, we review evaluation benchmarks and metrics for assessing memory utility, and outline various open challenges and future directions.

## 1 Introduction

The landscape of Artificial Intelligence (AI) has now undergone a fundamental paradigm shift: from prioritizing foundation model architecture and simplified benchmark performance to emphasizing problem definition and rigorous real-world evaluations. This marks the end of the "first half" of AI development, in which progress was primarily driven by training methods (Liu et al., 2024a), scaling (He et al., 2016; Achiam et al., 2023; Gu et al., 2025), and model architectures (Vaswani et al., 2017), which repeatedly pushed higher scores on standardized benchmarks (Wang et al., 2024m; Krizhevsky et al., 2012). During this period, the field converged on a dominant paradigm of scaling data and model size. This paradigm follows a general recipe of massive pre-training (Minaee et al., 2024) followed by a post-training process (Ouyang et al., 2022), which solves traditional benchmarks with remarkable accuracy. Under well-defined training pipelines, Large Language Models (LLMs) and agents can achieve over 90% accuracy on benchmarks such as MMLU (Hendrycks et al., 2021b) or MATH (Hendrycks et al., 2021c). As a result, LLMs and agents have rapidly evolved from static predictors, like conventional machine learning models, into general-purpose agents capable of complex reasoning (Wu et al., 2025f), planning (Li et al., 2025i), and tool use (Huang et al., 2025a; Zou et al., 2025b) in various tasks and environments. In this survey, we define a **foundation agent** as an autonomous or semi-autonomous system driven by a foundation model (e.g., LLM, Vision-Language Model (VLM), or world model), augmented with perception, reasoning, planning, tool use, and memory management (see Section 2 for more details).

Despite the impressive capabilities demonstrated on standard benchmarks, a significant gap remains between the reported performances and the utility in many real-world tasks and environments (Yu et al., 2024b). The majority of evaluation protocols largely simplify experimental assumptions and design static, pre-defined rules, with relatively short and isolated task settings (Cobbe et al., 2021; Chen et al., 2021; Budzianowski et al., 2018). In particular, most recent agent evaluation benchmarks are coupled with short agentic execution times without multi-turn, long-term interaction (Wang et al., 2024j; Lu et al., 2025a). As a result, these

## Foundation Agent Memory System

Figure 2: The taxonomy of Foundation Agent Memory. The **Memory Substrate** (**what form is represented**) for foundation agents includes the internal and external memory (§3.1), where internal memory covers weights, latent states, and KV caches (§3.1.2), and external memory covers vector indices, text records, and structural/hierarchical stores (§3.1.1). In the **Memory Cognitive Mechanism** (**how memory functions**) perspective, memories are categorized into *sensory*, *working*, *episodic*, *semantic*, and *procedural* memory (§3.2; see §3.2.1–§3.2.5 for each type). Based on the **Memory Subject** (**who is supported**), the memory is characterized into *user-centric* and *agent-centric* perspectives (§3.3).

evaluations no longer reflect the foundation agent's ability in reality, where interactions are inherently long-horizon, long-context, and deeply user-dependent with high-level complexity. As the field transitions towards more realistic settings, such as embodied agents (Li et al., 2024b), GUI automation (Ye et al., 2025a), agentic coding and computer use, deep research (Huang et al., 2025c), personal health-care (Zhan et al., 2024), and human-agent collaborations (Feng et al., 2024; Zou et al., 2025c), the complexity of the operational environment explodes, exposing agents to exceptionally large and dynamic contexts. In such settings, static, one-shot capabilities are insufficient. Instead, agents must accumulate, retain, and selectively reuse information across interactions. Memory thus emerges as the critical and natural solution to bridge the gap between idealized benchmark performance and real-world implementation and environment (Zhang et al., 2025q). Unlike fixed-task AI systems where memory primarily optimizes for efficiency and cost, foundation agents require memory that supports continuous accumulation, selective reuse, cross-session retention, and user adaptation in dynamic, long-horizon environments.

As the field enters the "second half" of AI development, the focus shifts from improving training recipes to solving the critical utility problem in reality (Bell et al., 2025; Yao et al., 2025). How to design a benchmark to evaluate an agent in the real environment has become one of the most important challenges (Xu et al., 2025c), particularly as agents strive to adapt along two primary empirical dimensions: **user-facing**

**personalization** (Cai et al., 2025; Zhang et al., 2025o; Wu et al., 2025e) and **task-oriented specialization** (Ling et al., 2025; Zhang et al., 2025i). In both dimensions, the interaction contexts of a specific user's long-term history or the vertical tasks like coding (Islam et al., 2024) and web search (He et al., 2024a) expand far beyond what can be accommodated by prompt-based mechanisms alone. As multi-session data from daily interactions or accumulated context from project work expands exponentially, reliance on a static memory mechanism is insufficient. As a result, memory architectures evolve from a static, predefined, and simple mechanism (Hao et al., 2023; Hu et al., 2023) towards a self-adaptive, self-evolving, and flexible unit (Liu et al., 2025g;f), to intelligently store, load, summarize, forget, and refine memory so as to retain informative experience for downstream tasks.

This shift forms a closed loop across the memory hierarchy: working memory gates what enters and remains in the active context, with curation policies increasingly trained end-to-end via reinforcement learning rather than hand-designed heuristics (Zhang et al., 2025p; Yuan et al., 2025a; Zhou et al., 2025c; Yu et al., 2025b); the retained experiences accumulate as episodic records that reflection and consolidation distill into semantic facts and procedural skills (Yang et al., 2025a; Liu et al., 2025b; Ouyang et al., 2025), which are further externalized as explicit, composable, and shareable skills that agents autonomously induce, revise, and maintain, and that modern agent harnesses expose through context engineering and standardized tool-mediation protocols (Wang et al., 2025c; Anthropic, 2025; Belikova et al., 2026; Song et al., 2026; Zhang et al., 2026a). Memory thus serves not merely as an interface to an agent's history, but as the substrate of agent self-evolution itself (Gao et al., 2025a), a thread that runs through cognitive mechanisms (Section 3.2), memory operations (Section 4), and learning policies (Section 5).

Although the rapid growth of foundation agent memory research has produced several surveys, important gaps remain in how agent memory is analyzed from a system-design perspective in real-world agent utility settings. Such memory design has shifted from short, isolated prompts to long-horizon interaction, where agents must operate over exploding context windows, multi-session workflows, and persistent real-world user relationships. Early works often organize memory primarily by task applications or management strategies (Zhang et al., 2025q; Du et al., 2025), or adopt neuroscience-inspired perspectives that project AI memory onto human memory through functional analogies and memory lifecycles (Wu et al., 2025g; Liang et al., 2025). While these approaches provide useful conceptual grounding, they do not systematically characterize underlying memory substrates or explicitly model the subject that memory serves within an agent system, which are significant when the context exceeds the foundation model's limitations. As a result, they fall short of distinguishing the optimization goals of agent memory and overlook the connections across complementary dimensions that are essential for designing and deploying autonomous agent systems in real-world applications. More recent work begins to broaden this conceptual landscape. In particular, Hu et al. (2025d) organizes agent memory along forms, functions, and temporal dynamics. Despite offering a valuable consolidation of memory, it remains largely partial, focusing mainly on how memory functions in agent-centric tasks rather than how memory should be designed, optimized, and deployed for users. The motivation for this survey arises from the fragmentation of current literature across retrieval-augmented generation (RAG) systems, long-context modeling, cognitive memory architectures, personalization, continual learning, reflection, and agent evaluation, making it difficult for researchers to understand how different memory mechanisms relate and how to design memory systems for realistic deployments. Our survey addresses this gap with three contributions: (1) a three-dimensional taxonomy connecting substrate, mechanism, and subject—dimensions often discussed separately in prior reviews; (2) a system-design analysis of how memory is instantiated, operated, and evaluated across single- and multi-agent topologies; and (3) positioning memory as the central mechanism for real-world utility and for agent self-evolution and self-improvement in the "second half" of AI development, with six open challenges to guide future research. To this end, we analyze foundation agent memory across hundreds of papers from three major complementary perspectives that connect memory with system-level design choices in increasingly complex environments.

Specifically, we introduce a unified taxonomy that is organized around three core design dimensions: the **memory substrate**, the **cognitive mechanism**, and the **memory subject**, as shown in Figure 2. We classify existing foundation agent memory works by substrate into external and internal, by cognitive mechanism into functional categories such as sensory, working, episodic, semantic, and procedural memory, and by subject into user-centric versus agent-centric subjects. From a system perspective, we further analyze

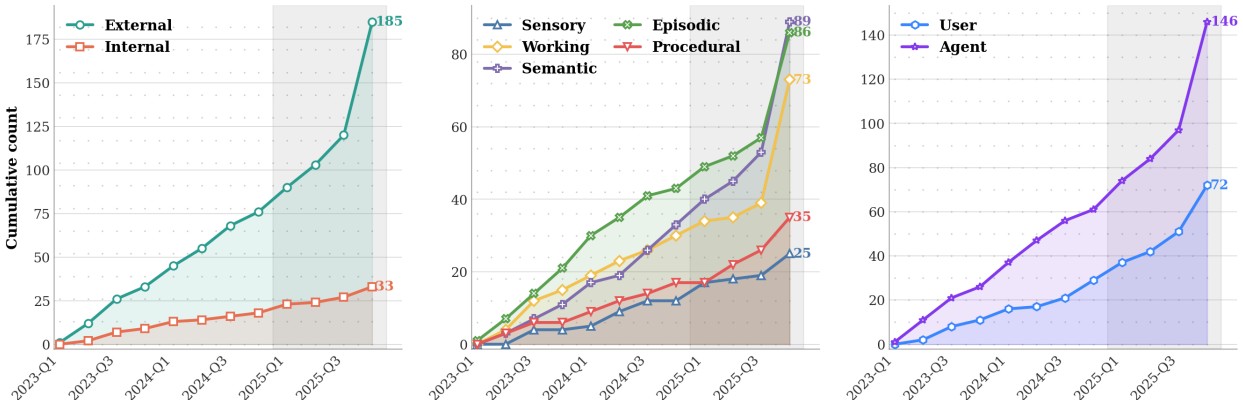

Figure 3: Cumulative publication trends of memory-related research in LLM agents (2023 Q1 – 2025 Q4). The plots illustrate the distribution of 218 collected papers across three key dimensions: memory substrate (left), memory cognitive mechanism (middle), and memory subject (right). The shaded region highlights the rapid acceleration of research output observed in 2025. In the left panel, external and internal refer to memory stored outside the model (§3.1.1) and memory held in model weights and inference-time states (§3.1.2), respectively. The middle and right panels correspond to cognitive mechanisms (§3.2) and memory subjects (§3.3). A single paper may exhibit several cognitive mechanisms, so the middle panel sums to more than the 218 collected papers, whereas the left and right panels partition them.

how these foundation agent memory systems are operated under different agent topologies, distinguishing the fundamental memory operations in single-agent systems from those memory routings in multi-agent settings. Furthermore, we highlight the growing role of learning policies, showing how agents are increasingly trained to master memory management itself and thus to learn from their interaction histories and self-evolve over time. These learned policies close the loop between short-term selection and long-term consolidation, turning memory from a passively managed resource into an actively evolving capability. To reflect the impact of shifting environments on foundation agent memory design, we discuss scalability issues across interaction horizon, environment complexity, and the number of interacting agents and systems, and we review the evaluation method and metrics used to measure memory performance and utility. Finally, we outline six open challenges in foundation agent memory to guide the next generation of foundation agent memory design.

## 2 Background

### 2.1 Large Language Models and Foundation Agents

Recent advances have pushed large language models beyond one-shot question answering into foundation agents (Park et al., 2023; Xi et al., 2025a; Luo et al., 2025; Liu et al., 2025a): systems capable of perceiving environments, reasoning through complex objectives, and executing actions to achieve assigned goals. Unlike traditional "one-shot" chatbots, an agent operates in a full loop: it interprets instructions, selects actions, observes outcomes, and updates its internal state or memory. This iterative interaction makes agents particularly well-suited to long-horizon, dynamic problem solving. This trend is already visible in emerging agentic products such as Deep Research and Manus, which emphasize multi-step execution and tool-augmented decision-making over single-turn responses (Zhang et al., 2025l).

A foundation agent is typically an autonomous or semi-autonomous system that uses foundation models, such as LLMs, as its core decision modules, augmented with mechanisms for state estimation, action execution, and memory management. In this survey, we use the term *foundation agent* to denote an AI agent whose core decision-making is driven by a general-purpose foundation model, including large language models, vision–language models, or learned world models. Core capabilities commonly include planning for task decomposition and decision-making (Yao et al., 2023; Huang et al., 2024b), tool use through external functions or models (Wu et al., 2025c; Yuan et al., 2025c; Lu et al., 2025a), multimodal perception (Liu et al., 2023a;

Bai et al., 2025), and memory that spans both short-term context and longer-term horizons (Lumer et al., 2025; Wang et al., 2025t). Memory (Zhang et al., 2025q; Xiong et al., 2025d) becomes especially important because context windows are limited and environments evolve over time; accordingly, many agents rely on external memory stores, coupled with summarization (Lu et al., 2025b), reflection (Renze & Guven, 2024), or consolidation procedures that compress experience into reusable knowledge (Kang et al., 2025c; Yu et al., 2025b). A growing line of work further frames foundation agents as self-evolving systems that improve from their own execution experience at deployment time, rather than solely through offline training (Gao et al., 2025a), and emerging benchmarks have begun to directly evaluate such experience-driven test-time learning over self-evolving memory (Wei et al., 2025d). Memory is the substrate that makes this self-evolution possible: without mechanisms to record, consolidate, and reuse experience, every episode leaves the agent unchanged, and long-horizon interaction degenerates into a sequence of isolated one-shot tasks. Multi-agent systems (Talebirad & Nadiri, 2023; Wu et al., 2024b) extend this paradigm by assigning specialized roles and memories to multiple agents that communicate and coordinate (Lan et al., 2024; Estornell et al., 2025).

Foundation agents are now being explored for different real-world applications, such as workflow automation (Xiong et al., 2025c), tutoring (Wang et al., 2025l), web and GUI interaction (He et al., 2024a; Wang et al., 2025e), embodied control in simulated or real environments (Fan et al., 2022; Yang et al., 2025c), and early forms of agentic scientific assistance (Ren et al., 2025; Pantiukhin et al., 2025; Zheng et al., 2025d). Despite this progress, several foundational challenges remain. First, long-horizon reliability (Huang et al., 2024b; Xi et al., 2025b) requires preventing compounding errors, behavioral loops, and unreliable replanning. Second, evaluation (Yehudai et al., 2025) should move beyond static QA toward benchmarks that measure dynamic, interactive capabilities, such as tool use, multi-step decision-making, and long-horizon feedback. Third, alignment and safety (Hua et al., 2024; Yuan et al., 2024a; Tian et al., 2023; Zhang et al., 2024d) become increasingly important as autonomy and tool access expand, demanding stronger guarantees of controllability. Addressing these challenges could ultimately turn today's tool-using assistants into agents that are capable enough to solve more complex tasks and trustworthy enough to stay predictable and controllable in real-world deployments.

## 2.2 Memory

Memory generally refers to a system's ability to retain, organize, and exploit information over time. In the context of LLM-based foundation agents, memory is used to explain how agents go beyond single-turn contexts to support long-term interaction, behavioral consistency, and experience accumulation (Luo et al., 2025). While biological studies often characterize memory as persistent, experience-driven neural change (e.g., synaptic plasticity and consolidation (Hebb, 2005; Bliss & Lømo, 1973; Tonegawa et al., 2015)), for agent systems, the more relevant insight lies in how memory is designed, realized, and used in practice to support different functions, representations, and targets.

In human cognitive models, memory is commonly understood as a set of interacting subsystems organized across different time scales. Short-term memory supports the temporary retention and manipulation of information during ongoing processing. Sensory memory briefly buffers raw perceptual input, enabling downstream processing (Sperling, 1960), while working memory operates under strict capacity constraints and supports online information manipulation, reasoning, and control (Baddeley, 2020; 2000). Long-term memory supports information retention over extended periods and comprises multiple functionally distinct systems. Episodic memory stores specific experiences situated in time and context, semantic memory accumulates abstract facts and conceptual knowledge (Tulving, 1972; 1985), and procedural memory captures skills, habits, and action policies that are typically expressed implicitly through performance rather than explicit recall (Cohen & Squire, 1980; Squire, 1992). Crucially, these subsystems do not operate in isolation. Short-term systems determine which experiences are perceived, selected, and abstracted during ongoing processing, while long-term systems accumulate these experiences and consolidate them into stable knowledge and skills; their interaction, rather than any individual store, constitutes the loop through which behavior improves with experience (Gao et al., 2025a). We adopt this systems view for foundation agents as well: agent memory is treated as an integrated hierarchy whose operations, from selection and retention to consolidation and skill formation, can themselves be designed, optimized, and, increasingly, learned from the agent's own interactions. In biological systems, functional distinctions in memory are ultimately grounded in physical substrates, such

as synaptic plasticity (Hebb, 2005; Bliss & Lømo, 1973; Martin et al., 2000) and circuit-level changes that give rise to enduring memory traces or engrams (Tonegawa et al., 2015; Josselyn & Tonegawa, 2020). While these cognitive and biological perspectives provide essential grounding, foundation agent memory systems introduce additional design considerations, including explicitly distinguishing whose information memory is designed to capture and support, what physical substrate it is realized in, and what cognitive role it plays, three questions that structure the taxonomy, together with how its operation policies are acquired (Section 5).

## 3 Taxonomy of Memory in Foundation Agents

We conducted a systematic literature collection by querying Google Scholar with memory-related keywords (e.g., agent memory, long-term memory, context management, personalization memory) and manually scrutinizing proceedings of major computer science conferences and journals (including top-tier NLP, ML, IR, and AI venues). From an initial pool of several hundred papers, we curated the most relevant contributions through iterative screening, ultimately selecting 218 key articles published between 2023 Q1 and 2025 Q4. The resulting trends, illustrated in Figure 3, demonstrate a dramatic escalation in research activity, and Figure 1 places the representative frameworks along a timeline. Most notably, the cumulative publication volume rises sharply throughout 2025, with the steepest growth in Q4. These trends indicate a growing need for more intelligent memory designs that enable agents to complete long-horizon, long-context tasks in increasingly complex environments.

To better distinguish memory design, we categorize memory in foundation agents along three orthogonal perspectives, as shown in Figure 2: (1) **Memory Substrates**, also referred to as storage formats, describing in what form memory is represented across different settings, are presented in Section 3.1; (2) **Memory Cognitive Mechanisms**, describing the functional role memory serves in the pipeline or workflow, are presented in Section 3.2; (3) **Memory Subjects**, whose information the memory is designed to capture and support, are elaborated in Section 3.3. In addition, we present the taxonomy in Figure 4. We then cover the operation and management of memory in single- and multi-agent systems (Section 4), learning policies (Section 5), scaling (Section 6), evaluation (Section 7), applications (Section 8), and open challenges (Section 9).

### 3.1 Memory Substrates

Memory substrates serve as the essential mechanisms for retaining historical knowledge produced during interactions between foundation agents and humans in various tasks and environments. Based on the framework, storage mediums, and persistence mechanisms of current research, we categorize these substrates into external and internal memory. The definitions and implementations of these categories are elaborated in Sections 3.1.1 and 3.1.2, respectively.

### 3.1.1 External Memory

> **External Memory**
>
> *External memory* refers to any memory substrate that stores the knowledge, information, and past experience **outside the agent model's parameter or state**. The agent can explicitly read from and write to the external memory via retrieval and update operations, enabling scalable, easy-to-update, cross-session retention of knowledge and interaction history.

The external memory represents information storage systems that store information in a **vector index**, **text-record**, **structural store**, and **hierarchical store** (Lewis et al., 2020; Zhang et al., 2023a). It operates independently of the gradient-updated weights within the neural network. And, it is characterized by a clear separation between the computation, performed within the LLM's internal parameters, and the knowledge, stored in an external database or memory module (Mallen et al., 2023; Zhong et al., 2024). This separated computation and retrieval design allows agents to access extensive and continually updated information without requiring expensive retraining of the foundation model (Omidi et al., 2025; Aratchige & Ilmini, 2025). In addition, external memory is both scalable and flexible. With just a few changes, it can be easily added, replaced, or inserted into most frameworks. Moreover, unlike internal memory (Saha et al., 2021), where

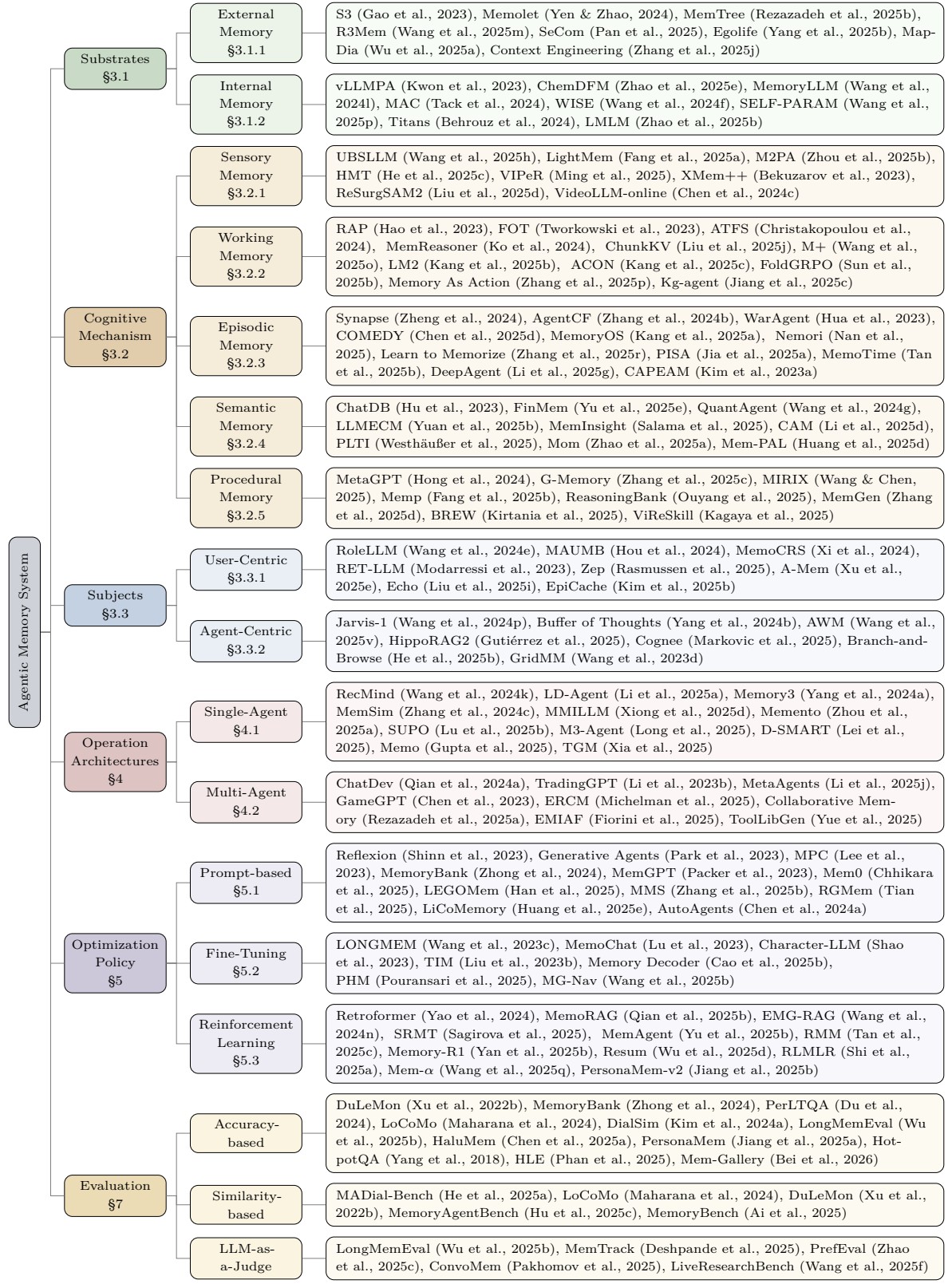

Figure 4: A taxonomy of the Foundation Agent Memory System

knowledge may be overwritten or adjusted due to weight updates from new experience, external memory can preserve past information in its original form, helping reduce hallucinations or knowledge cutoff by utilizing an additional storage module (Castrillo et al., 2025).

However, this design also has drawbacks and trade-offs. Normally, inference takes a longer time since the agent has to retrieve information from external storage. This problem becomes more severe when the agent runs in multi-turn iterations or in complex environments. In addition, the retrieval can be unreliable. If the similarity or ranking algorithms are not well-developed, they may provide results that are irrelevant or unhelpful for the current task. This irrelevant information would introduce noise that may reduce the agent's utility (Xu et al., 2025a). As the system operates for longer periods, with more turns, and when the memory grows, the escalating costs of storage and indexing may further diminish efficiency and performance (Chen et al., 2025e). As a result, a well-constructed external memory usually demands solid procedures for regulating memory size and quality, including summarization, selective retention, forgetting, deduplication, and periodic pruning of low-value or obsolete items (Du et al., 2025; He et al., 2024b).

**Vector Index.** Vector index is implemented by embedding memory items or queries into a shared vector space, running approximate nearest-neighbor search to fetch top-$k$ items, and appending those snippets to the prompt as grounding context (Lewis et al., 2020; Zou et al., 2025a). The dominant implementation of external memory in the current work of agents or LLMs is the vector database, utilized within the RAG framework (Quinn et al., 2025; Li et al., 2025i; Zhang et al., 2025l). This approach operates memory as a geometric problem. By embedding textual or multimodal information into high-dimensional vectors, it represents each memory item as a point in continuous space. New queries are also transformed into vectors with the same embedding mechanism, and the agents search for the closest points, the data representing the closest semantic information. Current techniques, such as hierarchical navigable small-world (HNSW) (Liu et al., 2025h), inverted file (IVF) (Rege et al., 2023), or product quantization (PQ) (Huang & Huang, 2024) indexing, are developed and optimized to find the most semantically similar documents or past interactions efficiently in a high-dimensional embedding space. This approximate-nearest-neighbor search retrieves a small set of context snippets, which are mapped back to their original form and fed into the agent or LLM to integrate its response with relevant, previously unseen information (Guu et al., 2020).

**Text Record.** Text-record memory treats an agent's long-term memory as a set of explicit text documents rather than embeddings or graphs (Zhang et al., 2025q). It is implemented as persistent human-readable text artifacts (e.g., a running "core summary" plus episodic logs) that are periodically summarized or edited, and selectively copied into the prompt when generating the next response. A common implementation keeps a persistent core summary and augments it with semantic and episodic lists. The semantic list stores discrete facts and can be modified via insert, update, and delete operations. In contrast, the episodic list is a chronological ledger of timestamped events and interactions. During memory updates, the agent modifies these structures (Zhu et al., 2025c; Wang & Chen, 2025). During response generation, it extracts a limited selection of pertinent information and instances, then reintegrates them into the prompt with the primary summary. The architecture facilitates transparency and fast integration by structuring memory as human-readable summaries and lists (Yue et al., 2025; Park et al., 2023). However, it requires meticulous summarization and pruning to maintain a succinct core summary and manageable lists.

**Structural Store.** Structural store is implemented by storing memories in explicit schemas and retrieving them via symbolic queries or traversal before formatting the returned records for the LLM's context. The structured memory architecture (Jia et al., 2025a; Bei et al., 2025), based on the topological design, can be partitioned into *relational tables*, *graph-based memories*, or *tree-based memories*. **Relational tables** use SQL to retrieve row records from a table. They can store facts and preferences as structured records, separate short-term transactions and long-term events into different tables, and use joins and indexes for efficient retrieval between different tables (Wang et al., 2024d). **Graph-based memory** represents episodes or pieces of information as nodes and edges within a knowledge graph (Anokhin et al., 2024). A knowledge graph is essentially a semantic network. It typically organizes information into nodes with entities and edges capturing the relationships between them. When new information arrives, the system combines semantic embeddings and keyword search with traversal to detect edge changes, resolve conflicts, and maintain the valid graph

structure (Jiang et al., 2025c). **Tree-based memory** organizes knowledge hierarchically, with each node in the tree storing an aggregated summary of textual content and a corresponding semantic embedding, and deeper branches representing more specific details (Sarthi et al., 2024). When new information arrives, the system traverses the tree and updates or creates nodes based on embedding similarity to decide where the information is stored. This dynamic mechanism supports multi-level abstraction and efficient retrieval to support long conversation tasks (Rezazadeh et al., 2025b). The structured memory, with its different types of formats, can capture sequential history and relationships and support multi-level abstraction. This technique offers the agent flexibility and control over how memories are stored, updated, and queried.

**Hierarchical Store.** Hierarchical memory separates memory into multiple storage units, each with its own schema and storage strategy. **Instead of keeping everything in a single flat archive, the agent maintains dedicated memory modules** (Rezazadeh et al., 2025b; Zhai et al., 2025). For instance, the agent may maintain a core memory for persistent persona and user facts, an episodic memory for time-sensitive events, a semantic memory for abstract concepts, a procedural memory for step-by-step instructions, a resource memory for documents and media, and a knowledge store for sensitive or private information (Yang et al., 2025a; Wang et al., 2025q). Each module utilizes a data structure suited to its content. For example, episodic memory might store records with `event_type`, `summary`, `details`, and `timestamp`; semantic memory may organize entries by `type`, `summary`, and `source`; procedural memory encodes workflows as JSON sequences; and the knowledge store applies strict access controls to secure credentials and API keys. A meta-memory manager (Wang & Chen, 2025; Zhang et al., 2024a) coordinates these modules, directing queries to the correct store and utilizing specialized retrieval methods (e.g., *embedding similarity* (Xu et al., 2025d), *BM25 match* (Hu et al., 2025c), *string match* (Alla et al., 2025)) for each storage unit. This multi-store substrate has advantages such as modularity, separation of concerns, and scalability, making it easier to design specialized schemas and retrieval paths while supporting nuanced personalization across long-term interactions. However, coordinating multiple memory modules increases the system's architectural complexity. It can potentially add latency and additional resource cost when querying several stores, and requires well-designed mechanisms to maintain data consistency and synchronize updates between different turns in the long-horizon task (Sun & Zeng, 2025; Li et al., 2025k; Maragheh & Deldjoo, 2025).

### 3.1.2 Internal Memory

> **Internal Memory**
>
> *Internal memory* refers to the information stored **directly within the model's architecture**, encompassing both the persistent knowledge embedded in its parameters (i.e., parametric memory) and the working states utilized during inference.

**Weights.** Weight memory is implemented by writing memory into parameters, so the model later recalls the information without retrieving external context. **Knowledge or experience is embedded directly into the neural network's parameters** (Mallen et al., 2023) through *pre-training*, *post-training*, or *targeted parameter editing* (Wang et al., 2025r; Ampel et al., 2025; Wang et al., 2024h). Because this knowledge is internal, these models recall facts and past events efficiently and robustly without communicating with an external store. However, maintaining and updating internal weights is challenging. Therefore, recent research has divided weight updates into three main strategies: *continual learning*, *model editing*, and *distillation*. **Continual learning** methods incrementally update the model's weights with the new information while attempting to avoid catastrophic forgetting (De Lange et al., 2021). Methods like regularizing changes to prevent overwriting important weights and applying soft masks to selectively update parameters can largely preserve the previous domain knowledge while recognizing new information (Serra et al., 2018). **Model editing** strategies treat specific factual associations as localized memories and adjust the corresponding parameters: one line of work locates decisive mid-layer weights and applies rank-one updates to change a single factual association, whereas another extends this idea to update thousands of memories by applying small updates across multiple layers (Meng et al., 2022; 2023). For instance, some methods show that carefully fine-tuning the model with augmented data can achieve comparable editing performance. Another method inserts a few new neurons to correct mistakes sequentially without affecting unrelated behavior. Another approach

adds calibration memory slots that store corrected facts without altering the original parameters (Chhetri et al., 2025; Luo & Specia, 2024). **Distillation** techniques compress contextual knowledge into the parameters themselves. They can internalize prompt- or context-dependent behaviors into model parameters by training a student to reproduce a prompted teacher's outputs (e.g., prompt-to-weights or in-context distillation). Concretely, some approaches optimize the model so it behaves as if a fixed prompt were implicitly present at inference time (Cao et al., 2025a; Padmanabhan et al., 2023). While such parameter-based memorization can make recall efficient at inference, modifying weights is computationally costly and may introduce interference, overwriting, or distortion of previously learned knowledge (Meng et al., 2023).

**Latent-State.** Latent-state memory is implemented by carrying forward and reusing intermediate hidden states across steps or segments. Earlier activations can directly impact later computation even when the raw tokens are no longer in the window. The intermediate activations or hidden-state tensors are produced as information flows through layers during the forward pass process (Ibanez-Lissen et al., 2025). These hidden states are the layer-wise representations that combine the fixed learned parameters with the current prompt, forming the model's working state that drives the next-token prediction. Unlike weight-based memory, state memory is not durable. It is usually created during runtime activations, then used, and typically discarded or reset at the end of a request. Therefore, it supports within-session coherence and reasoning but does not persist knowledge across sessions unless exported externally. The key trade-off is resource cost. The environment must hold not only parameters but also intermediate states in memory, and this activation value scales with model depth, batch size, and precision. Efficient and optimized latent-state management reduces inference latency and memory cost. One representative direction reconstructs hidden representations across layers to reinforce earlier context at a controllable cost (Dillon et al., 2025). In contrast to reconstruction, other work caches a compact subset of hidden-state vectors or memory tokens across segments, thereby extending the effective context length beyond a fixed window while reducing memory and computation costs (Dai et al., 2019). Beyond reusing existing states, latent-state memory can also be generated. A memory trigger and memory weaver synthesize machine-native latent tokens that are fed back into the model to enrich downstream reasoning (Zhang et al., 2025d). Meanwhile, structured modulation of hidden-state transitions maintains latent trajectories aligned with prior context, which can reduce semantic drift in long sequences (Carson & Reisizadeh, 2025). Additionally, some architectures integrate compressive memory into the attention mechanism to store and reuse key-value pairs from previous segments, enabling the processing of very long inputs with bounded memory (Munkhdalai et al., 2024). Other strategies treat the hidden state as a fast-weight updated through small optimization steps during inference to refine internal representations over extended contexts (Zhu et al., 2025b).

**KV Cache.** In transformer-based LLMs, the KV cache is a transient, inference-time memory that speeds up autoregressive decoding (Kwon et al., 2023; Liu et al., 2023c). It is implemented by caching per-layer attention keys and values from previous tokens during decoding and reusing them for subsequent tokens. During self-attention, each token is projected into keys and values (Ge et al., 2023; Pope et al., 2023). Without KV caching, these matrices are recomputed for all previous tokens at every step, leading to unwanted resource waste and slowing generation for long sequences (Cai et al., 2024; Feng et al., 2025b). The KV cache stores the keys and values from earlier tokens and, at each new step, only computes them for the freshly generated token and retrieves the rest from the cache. This mechanism significantly accelerates inference, especially on long outputs or large and deep models, but comes at the cost of higher memory usage and implementation complexity (Pope et al., 2023; Kwon et al., 2023). Recent research proposes several high-impact approaches for compressing the KV cache. One approach recognizes that a small portion of tokens contribute most to attention scores and dynamically evicts the rest, balancing "heavy hitters" with recently generated tokens. This method achieves large throughput improvements while retaining only a fraction of the cache (Zhang et al., 2023b). Another technique identifies consistent attention patterns within a prompt's observation window and clusters important features to enable substantial speed and memory savings for long input sequences (Li et al., 2024d). Another approach observes that layers differ in how many key-value vectors they truly need, and instead of giving every layer the same cache size, ranks vectors by importance and uses a binary search to decide how many to keep per layer under a global budget, so only the most informative vectors are retained (Wang et al., 2024a). These current studies demonstrate that the KV cache can be used efficiently to improve model performance significantly.

### 3.1.3 Trade-offs Across Memory Substrates

Different memory substrates can have different advantages and drawbacks. They have trade-offs in terms of access speed, scalability, adaptability, and reliability in ways that show up quickly on long-horizon tasks. In practice, the choice often comes down to fast, precise internal state versus scalable external storage that can become noisy as it grows. Table 1 summarizes these trade-offs across eight operational dimensions: retrieval latency (time cost of accessing stored information), storage cost (memory footprint required), update flexibility (ease of modifying stored content), context window overhead (additional burden on the context window), cross-session persistence (whether information survives across sessions), scalability (capacity to accommodate growing memory), forgetting risk (susceptibility to knowledge degradation upon modification), and retrieval precision (accuracy of targeting relevant stored information).

Table 1: Comparative analysis of memory substrate trade-offs across eight operational dimensions.

| Dimension | Internal Memory (§3.1.2) | External Memory (§3.1.1) |
|---|---|---|
| Retrieval Latency | Low; accessed through forward pass or GPU-resident caches | Variable; dependent on index structure, corpus size, and retrieval algorithm |
| Storage Cost | High; scales with model depth and sequence length | Moderate; scales with stored entries, independent of model architecture |
| Update Flexibility | Low; requires fine-tuning, model editing, or distillation | High; supports explicit insert, update, and delete without modifying parameters |
| Context Window Overhead | Significant for latent-state and KV cache substrates | Moderate; retrieved entries must be serialized into prompt |
| Cross-Session Persistence | Partial; weights persist but latent states and KV caches are discarded | Permanent; entries maintained across sessions |
| Scalability | Constrained; parametric capacity fixed at training time | High; storage grows independently of model architecture |
| Forgetting Risk | High for parametric substrates (catastrophic forgetting); KV caches lack durability | Negligible; original entries preserved unless explicitly modified |
| Retrieval Precision | Implicitly determined by learned associations; high within-session for KV cache | Variable; sensitive to embedding quality and ranking algorithms |
| Representative Works | MemoryLLM, WISE, SELF-PARAM (parametric); vLLMPA, Titans, ChunkKV, EpiCache (latent/KV) | Mem0, Zep, A-Mem, HippoRAG2, MIRIX, MemTree |

Internal memory, including parametric memory encoded in model weights, usually offers high-speed access and tight integration with reasoning. However, it is expensive to update and can suffer from catastrophic forgetting due to frequent modifications. It therefore fits stable, general-purpose knowledge better than rapidly changing or user-specific information. Latent substrates such as hidden activations or KV caches are fast and transient, making them well-suited to within-session state. However, their ephemeral nature and linear scaling with sequence length limit capacity and make them unsuitable for cross-session retention. Moreover, latent-state and KV cache substrates directly occupy the context budget, imposing significant context window overhead, and their storage cost scales with model depth, batch size, and precision.

External memory, consisting of vector databases or stores, scales naturally with experience and supports flexible editing without retraining. However, retrieval adds latency and makes performance sensitive to indexing and retrieval quality. Prior work shows that storing excessive or low-quality items can inject noise and increase the difficulty of targeting the informative items, which degrades tightly coupled decision-making (Liu et al., 2024b). Retrieval precision is further dependent on embedding quality, similarity metrics, and ranking algorithms, making it a key design consideration for external memory systems. To better tackle the issue, explicit memory management mechanisms such as pruning, summarization, and hierarchical organization are required and necessary.

Overall, no single substrate dominates across settings and environments, as illustrated by the complementary strengths and limitations summarized in Table 1. Effective system design increasingly adopts hybrid memory

architectures, using internal or parametric memory for inherent knowledge or facts, latent memory for fast short-term reasoning, and external memory for scalable experience storage. This pattern reflects a broader shift from a static or persistent, defined knowledge storage mechanism to a more dynamic and adaptive approach that can handle the complexities of real-world tasks.

### 3.2 Memory Cognitive Mechanisms

Human memory provides a conceptual scaffold for analyzing memory in LLM-based agents (Kim et al., 2023b; Li & Li, 2024; Sumers et al., 2023). Cognitive psychology distinguishes multiple interacting systems that explain how information is perceived, maintained, and reused (Baddeley, 2020; Tulving, 1972). While a wide range of cognitive memory types have been proposed in the literature, we focus on a set of five atomic cognitive memory systems that are particularly relevant for LLM-based agents: **sensory**, **working**, **episodic**, **semantic**, and **procedural memory**. Table 2 summarizes these five memory systems by mapping their core functional roles to representative research directions and illustrative agent-level implementations. These five systems constitute a minimal and architecturally complete decomposition of cognitive memory, whereas other memory constructs such as autobiographical (Conway & Pleydell-Pearce, 2000) or prospective memory (Einstein & McDaniel, 1990) can be understood as compositions or functional abstractions built upon them. Accordingly, our taxonomy is organized around these five atomic memory types, which cover both short-term and long-term memory mechanisms commonly realized, either explicitly or implicitly, in current agent architectures. Beyond their individual functions, these five systems play complementary roles in agent self-evolution: short-term systems (sensory and working memory) gate which experiences are perceived, selected, and abstracted during execution, while long-term systems (episodic, semantic, and procedural memory) accumulate these experiences and consolidate them into reusable knowledge and skills, together forming the loop through which agents improve from their own experience (Gao et al., 2025a).

#### 3.2.1 Sensory Memory

> **Sensory Memory**
>
> *Sensory memory* refers to the **temporary retention** of incoming perceptual signals, allowing attention and selection mechanisms to operate **before higher-level processing** occurs, by briefly holding raw inputs long enough for the system to decide what to attend to next.

In current foundation agents, sensory memory is typically not explicitly modeled. This is largely due to the highly abstracted nature of textual inputs, where perceptual processing has already been collapsed into symbolic or linguistic representations. In contrast to memory systems that encode stable knowledge, sensory memory functions as a transient interface between perception and cognition, operating over very short timescales and across multiple sensory modalities (Atkinson & Shiffrin, 1968). However, in multimodal or embodied agents, analogous mechanisms emerge in the form of short-lived perceptual buffers, such as caches of visual, auditory, or interaction embeddings. These buffers function as a sensory stage by temporarily retaining minimally processed observations before they are filtered, summarized, or routed to working memory for downstream reasoning and control.

Only a limited number of LLM agent works explicitly instantiate sensory memory as a distinct stage in their memory architectures. Hierarchical and multi-stage designs such as HMT (He et al., 2025c), LightMem (Fang et al., 2025a), and M2PA (Zhou et al., 2025b) model sensory memory as an initial buffer that retains recent, minimally processed inputs before selection, compression, or consolidation into downstream memory components. Beyond these explicit formulations, implicit sensory memory realizations are more common in streaming and embodied agents. Systems such as SAM2 (Ravi et al., 2024) and ReSurgSAM2 (Liu et al., 2025d) maintain short-term perceptual queues over recent video frames, while ReKV (Di et al., 2025) and V-Rex (Kim et al., 2025a) rely on streaming KV caches or memory queues to retain recent tokens or visual representations during online inference. Although these mechanisms are rarely labeled as sensory memory, they serve an analogous function by buffering recent observations to support perceptual continuity and computational efficiency, and are typically tightly coupled with working and semantic memory rather than implemented as standalone cognitive modules.

Table 2: Mapping five cognitive memory systems to their functional roles and corresponding memory design directions in foundation agents, with illustrative agent-level examples.

| Cognitive Type | Functional Role | Core research focuses in LLM agents (representative works) | Implementations |
|---|---|---|---|
| **Short-Term Memory** | | | |
| **Sensory Memory** | **What is perceived**. Brief retention of recent visual, auditory, or other sensory inputs before further processing. | **Perceptual buffering and lightweight caching** for multimodal streams, such as short-lived embedding queues and perceptual state buffers (He et al., 2025c; Fang et al., 2025a; Zhou et al., 2025b; Di et al., 2025). **Temporal gating and selection** mechanisms that stabilize noisy or high-bandwidth observations for downstream reasoning and control (Mon-Williams et al., 2025; Bjorck et al., 2025; Black et al., 2025; Ravi et al., 2024; Liu et al., 2025d). | **A short rolling buffer of the last 2–5 seconds of audio and video frames** (or recent sensor embeddings) to smooth perception and handle brief occlusions. |
| **Working Memory** | **What is currently handled**. Temporary holding and manipulation of current information. | **Pre-write representation shaping** that reduces what must enter the active context, including compression, folding, and abstraction-aware representations (Labate et al., 2025; Kang et al., 2025c; Wu et al., 2025d; Sun et al., 2025b; Ye et al., 2025b). **Online state maintenance and self-evolving curation** under fixed budgets, including update, eviction, and runtime control of context and KV states, with curation policies increasingly learned from the agent's own experience (Zhang et al., 2025p; Yuan et al., 2025a; Zhou et al., 2025c; Yu et al., 2025b; Zhang et al., 2025j; Kim et al., 2025b; Liu et al., 2025j; Kwon et al., 2023; Ni et al., 2025). | **An in-progress reasoning state (chain of thought):** "goal: refine the survey; earlier sections set the framing; the next revision should preserve framing consistency." |
| **Long-Term Memory** | | | |
| **Episodic Memory** | **What happened**. Contextual record of specific experiences. | **Episode recording and structuring**, including what to write, how to organize events and trajectories, and multi-scale episode formation (Rajesh et al., 2025; Yeo et al., 2025a;b; Anokhin et al., 2024). **Retrieval, reflection, and experience consolidation** at decision time, including adaptive triggering, episode selection, retention policies, and distilling episodes into reusable knowledge for self-evolution (Yeo et al., 2025b; Latimer et al., 2025; Li et al., 2025f; Sarin et al., 2025; Alqithami, 2025; Yang et al., 2025a; Liu et al., 2025b). | **A past interaction log:** "last time you preferred a 2-page summary; the previous plan failed due to missing API keys," stored with its time and situational context. |
| **Semantic Memory** | **What is known**. Conceptual and factual knowledge about the world. | **Knowledge induction and organization** into reusable representations, including memory graphs, schemas, and compact neural representations (Zhao et al., 2025a; Jia et al., 2025a; Li et al., 2025d; Rasmussen et al., 2025; Behrouz et al., 2024; Wang et al., 2024l; Pouransari et al., 2025). **Knowledge access and reliability control** during reasoning, including selective activation, validation, and continual revision under distribution shift (Jimenez Gutierrez et al., 2024; Wang et al., 2025m; Rezazadeh et al., 2025b; Yan et al., 2025b; Wang et al., 2024f; Alqithami, 2025). | **A knowledge base:** entities or facts (e.g., project info, preferences, definitions) retrieved by query and checked for reliability. |
| **Procedural Memory** | **How to act**. Skills and action patterns. | **Skill induction and packaging**, learning reusable procedures from experience, tools, or interaction traces, increasingly externalized as portable and shareable skills (Hong et al., 2024; Fang et al., 2025b; Han et al., 2025; Zhang et al., 2025d; Xia et al., 2025; Wang et al., 2025c; Anthropic, 2025; Xu & Yan, 2026). **Skill execution, composition, and adaptation**, invoking and refining procedures under changing contexts over long horizons, including self-evolving maintenance of skill libraries (Ouyang et al., 2025; Li et al., 2025h; Tablan et al., 2025; Terranova et al., 2025; Wang & Chen, 2025; Belikova et al., 2026; Song et al., 2026). | **A reusable workflow or tool skill:** "search → read → extract → cite," or "debug with sanitizer," invoked as a routine. |

Despite its limited explicit treatment in current LLM agent research, sensory memory is likely to become increasingly important as agents are deployed in multimodal, embodied, and robotic settings. In such environments, agents must process continuous, high-bandwidth sensory streams, such as video frames, audio signals, and proprioception, under real-time and memory constraints (Black et al., 2025; Bjorck et al., 2025; Mon-Williams et al., 2025). In these embodied applications, sensory memory is often instantiated through sensory-level buffering and gating mechanisms, such as short-lived perceptual embedding buffers, attention-driven filtering, and temporal integration, rather than explicitly labeled cognitive modules. These designs reduce redundant computation, stabilize partially observable environments, and support principled consolidation from raw sensory input into working and episodic memory. As a result, more explicit modeling of sensory memory may become a key design consideration for scalable embodied and robotic foundation agents.

### 3.2.2 Working Memory

> **Working Memory**
>
> *Working memory* refers to a short-term memory mechanism that supports the **temporary storage** and **active manipulation of information** necessary for complex tasks such as reasoning, comprehension, and learning, enabling information to be actively maintained during ongoing operations.

In the LLM-based agent setting, the core goal of working memory (Baddeley, 2020) is to maintain and manipulate task-relevant state under strict online capacity constraints, such that multi-step reasoning and action can proceed without interruption. Since LLMs are inherently stateless, working memory provides the mechanism through which such state is explicitly carried and updated across interaction steps. Beyond serving as a passive buffer, working memory management is itself increasingly treated as an adaptive capability of the agent: rather than relying solely on hand-designed truncation or summarization heuristics, recent agents learn from their own interaction experience how to curate what enters and remains in the active context, making working memory a direct object of agent self-evolution (Gao et al., 2025a).

Working memory in foundation agents is most commonly instantiated through the active context, which includes the prompt context, intermediate reasoning traces, tool outputs, and runtime states such as key-value caches that are accessible during inference. Within this instantiation, existing approaches differ in where they intervene during execution. One line of work focuses on how task-relevant state is represented before or as it is written into the active context. By compressing (Kang et al., 2025c; Wu et al., 2025d), restructuring (Sun et al., 2025b; Ye et al., 2025b), or abstracting interaction history (Labate et al., 2025), these methods aim to delay or avoid context saturation at the source. Specifically, Labate et al. (2025) replaces large intermediate outputs with lightweight references, while Kang et al. (2025c); Wu et al. (2025d); Sun et al. (2025b); Ye et al. (2025b) periodically summarize or fold completed reasoning segments to maintain a compact working context. A second line of work addresses the problem after working state has already accumulated. These approaches study how task-relevant state can be continuously maintained, updated, or evicted under a fixed online budget during execution. At the system level, runtime mechanisms manage states such as key-value caches and scheduling (Kim et al., 2025b; Liu et al., 2025j; Kwon et al., 2023; Ni et al., 2025). At the agent level, a growing body of work moves beyond fixed heuristics toward self-evolving working memory management, in which the curation policy itself is learned from the agent's own experience: agents are trained end-to-end, typically via reinforcement learning, to decide what to retain, consolidate, or discard as an integral part of the reasoning loop (Zhang et al., 2025p; Yuan et al., 2025a; Zhou et al., 2025c; Yu et al., 2025b). A complementary direction treats the working context itself as an evolvable substrate that improves across tasks, either by continually distilling execution experience into structured, reusable context (Zhang et al., 2025j; Suzgun et al., 2025), or by exposing runtime memory state to the agent so that it can autonomously archive and recover its own context during execution (Xu et al., 2026).

In summary, working memory in foundation agents serves as the agent's online workspace, realized through the active context under strict capacity constraints. Rather than treating longer context windows as the sole solution, existing work shows that sustaining coherent long-horizon reasoning depends on selectively retaining and manipulating task-relevant state during execution. This shift reframes working memory from a passive context buffer into a self-managed workspace: mirroring the broader evolution of agent memory

from static, predefined mechanisms toward self-adaptive and self-evolving designs, the policies that govern working memory are increasingly learned and refined from the agent's own experience rather than fixed in advance. Working memory further serves as the gateway of this self-evolution loop, since short-term state must pass through it before being consolidated into episodic, semantic, or procedural memory, and the selection and abstraction performed at this stage determine which experiences the agent can later learn from (Section 5). As agents scale to longer horizons and more complex interaction settings, progress in working memory will therefore hinge on principled and increasingly self-evolving state selection and transformation, rather than unbounded context expansion.

### 3.2.3 Episodic Memory

> **Episodic Memory**
>
> *Episodic memory* refers to a form of long-term memory dedicated to the persistent storage of an agent's **interactive experiences** across sessions. It records specific events situated in particular temporal and environmental contexts, typically organized as interaction trajectories, action sequences, and associated feedback.

The primary role of episodic memory in foundation agents is to preserve historical interaction contexts and outcomes over extended time horizons, enabling agents to reference past experiences when relevant to ongoing interactions or decision making (Tulving, 1983; 2002). By maintaining access to concrete past experiences, episodic memory supports background reconstruction and cross-session continuity in dynamic environments. This allows agents to ground their behavior in previously observed situations, maintain consistency across repeated interactions, and recover relevant context that would otherwise be lost between sessions.

Episodic memory in foundation agents is most commonly instantiated as an explicit experience repository that accumulates interaction histories across sessions and can be accessed by the agent when needed. From a methodological perspective, existing work on episodic memory in foundation agents can be grouped into two dominant lines of research based on their primary focus. One line of work focuses on episode recording, namely how historical interactions across sessions are organized into coherent episodic records that preserve event structure and situational context. For cross-session interaction histories, Rajesh et al. (2025) emphasizes selective episode writing and structured organization of past interactions, making what to store and how to organize episodic content explicit. For long video understanding, Yeo et al. (2025a) constructs episodic records by organizing events together with their temporal and causal relations, enabling coherent episode-level representations of extended visual experiences. More broadly, Yeo et al. (2025b) represents episodic events at multiple temporal scales, allowing episode formation and access to adapt to different levels of granularity. Anokhin et al. (2024) links episodic observations with semantic anchors, enabling episodic recall during planning. A second line of work focuses on retrieval and reflection, namely how stored episodes are triggered, selected, and leveraged at decision time. Yeo et al. (2025b) formulates episodic retrieval as an adaptive process that iteratively selects a memory source and temporal scale conditioned on the query and retrieval history. Latimer et al. (2025) defines explicit recall and reflection operations that retrieve episodes based on their relevance to the current reasoning context and use them to guide subsequent behavior. In tool-use settings, Li et al. (2025f) retrieves episodic experience by matching the current execution state against structured representations induced from past trajectories. For multi-session dialogue, Sarin et al. (2025) triggers episodic retrieval using session-level context and user state cues to recall relevant episodic summaries across sessions. Finally, Alqithami (2025) shows that retention policies under fixed memory budgets directly shape which episodes remain retrievable over long horizons.

Overall, episodic memory research in foundation agents centers on preserving concrete interaction experiences across sessions and enabling selective access to those experiences at decision time. Existing work primarily addresses two methodological questions: episode recording and episodic retrieval or reflection. From the self-evolving perspective, episodic memory further serves as the experiential substrate from which agent capabilities grow: accumulated episodes are not only retrieved to inform individual decisions, but are also distilled, through reflection and consolidation, into semantic facts and procedural skills, allowing agents to improve continually from their own execution history (Yang et al., 2025a; Liu et al., 2025b; Ouyang et al.,

2025). Open problems include how agents should define episode boundaries, regulate the influence of episodic recall during reasoning, and manage long-term retention as episodic memory scales.

### 3.2.4 Semantic Memory

> **Semantic Memory**
>
> *Semantic memory* refers to a form of long-term memory dedicated to the storage of **abstract facts**, **general concepts**, and **structured knowledge**. It provides agents with decontextualized information that remains stable over time and can be reused across different situations and objectives.

In foundation agent architectures, semantic memory functions as a stable knowledge base that supports **knowing-what** factual reasoning (Tulving, 1972). Its content is typically derived through the distillation and decontextualization of recurring facts accumulated in episodic memory. When an agent encounters similar knowledge patterns across multiple tasks, fragmented factual information from individual experiences is consolidated into universal concepts, entity relations, or attribute summaries via summarization mechanisms. This semanticization process equips the agent with a knowledge substrate analogous to an encyclopedia or technical manual. By enabling direct access to verified conceptual knowledge rather than repeated retrieval of raw interaction logs, semantic memory provides a coherent and reliable factual foundation for complex reasoning and decision-making. Importantly, in long-horizon agent settings, semantic memory is not static but subject to continual access, revision, and control as new information accumulates.

Existing semantic memory approaches mainly differ in how abstract knowledge is constructed and stabilized, and how it is selected, validated, and maintained as agents reason over long horizons. One line of work focuses on knowledge induction and organization, studying how stable, decontextualized knowledge is distilled from interactions, documents, or external evidence and stored in forms that can be reliably reused by agents. Representative approaches organize semantic knowledge using hierarchical schemas (Zhao et al., 2025a; Jia et al., 2025a), memory graphs (Rasmussen et al., 2025; Wang et al., 2024n), or entry-centric semantic structures (Xu et al., 2025e; Li et al., 2025d), supporting long-term maintenance and structured access during reasoning. Other approaches encode semantic knowledge into neural memory modules (Behrouz et al., 2024; Wang et al., 2024l) or auxiliary parameters (Wang et al., 2025p; Pouransari et al., 2025), emphasizing compact representations and fast reuse without relying on explicit external structures. A second line of work focuses on how semantic knowledge is activated, validated, and updated during agent reasoning as new evidence accumulates. Representative approaches treat semantic access as a decision-time control process in which abstract knowledge is selectively chosen, checked for applicability, and applied to the current reasoning state, rather than retrieved through a fixed similarity lookup (Jimenez Gutierrez et al., 2024; Wang et al., 2025m; Rezazadeh et al., 2025b; Yan et al., 2025b). Other work emphasizes long-term semantic reliability by introducing mechanisms for preference drift detection (Sun et al., 2025a), retention or forgetting policies (Alqithami, 2025), and continual knowledge editing (Wang et al., 2024f) to prevent outdated or inconsistent knowledge from degrading agent behavior.

In summary, semantic memory complements working and episodic memory by providing a stable yet revisable knowledge substrate. It serves as the primary site for distilling fragmented experiences into stable, reusable regularities and facts. By stripping universal knowledge ontologies from specific episodes, it equips the agent with the common-sense foundation necessary for cross-domain tasks. As research progresses, semantic memory is evolving from simple document storage toward self-evolving semantic networks, ensuring that agents maintain an accurate and consistent knowledge system over long-term operation.

### 3.2.5 Procedural Memory

> **Procedural Memory**
>
> *Procedural memory* refers to a form of long-term memory dedicated to **how to perform tasks**. It encodes operational skills, execution strategies, and automated routines for specific scenarios. Unlike memory that stores factual knowledge, it abstracts complex action sequences into reusable patterns, enabling agents to complete tasks efficiently and coherently.

In foundation agent architectures, procedural memory is typically reflected in the execution layer, where repeated action sequences, decision rules, or workflows shape how high-level decisions are carried out. Rather than being tied to a single memory substrate, such knowledge may be distilled from prior executions, learned through optimization, or shared across agents. As execution experience accumulates, short-lived action states can be consolidated into reusable skills or routines, allowing agents to execute tasks more consistently over long interaction horizons (Fang et al., 2025b).

The instantiation of procedural memory exhibits an evolution from explicit non-parametric templates toward implicit parametric neural policies through diverse mechanisms. Experience distillation and metacognitive control involve transforming historical trajectories into reusable schemes; for instance, From Experience to Strategy distills interactions into structured strategies (Xia et al., 2025), while Adapting Like Humans focuses on metacognitive routines for error correction at test time (Li et al., 2025h). Learning optimization and policy refinement emphasize the internalizing of skills into parametric weights; for example, Retroformer employs policy gradient optimization to refine agent actions (Yao et al., 2024), and BREW enhances task-handling through continuous training and refinement (Kirtania et al., 2025). Multi-agent coordination and shared practices establish common operational norms in collaborative environments, as explored in Smarter Together, MetaGPT, and MIRIX (Tablan et al., 2025; Hong et al., 2024; Wang & Chen, 2025). Workflow externalization and automation transform complex operations into explicit automation templates, as seen in Agent Workflow Memory and LEGOMem (Wang et al., 2025v; Han et al., 2025). Furthermore, self-evolving mechanisms in works like MemGen and ReasoningBank facilitate the accumulation of reasoning traces for long-term procedural evolution (Zhang et al., 2025d; Ouyang et al., 2025), while CoEvoSkills co-evolves a skill generator with a surrogate verifier so that skill packages can be refined without access to ground-truth tests (Zhang et al., 2026a). Finally, while work like Evaluating Long-Term Memory signals growing interest in assessment (Terranova et al., 2025), procedural memory evaluation is beginning to standardize around dedicated benchmarks that measure both the task-level utility of procedural knowledge and its transfer across tasks, roles, and models (Li et al., 2026; Belikova et al., 2026).

A notable recent development recasts procedural memory as explicit, shareable *skills*. Early skill libraries such as Voyager showed that agents can accumulate verified, reusable programs as an ever-growing behavioral repertoire (Wang et al., 2025c), and this idea has since matured into a broader ecosystem in which skills are packaged as composable bundles of instructions, code, and resources that agents load on demand (Anthropic, 2025; Xu & Yan, 2026; Wang et al., 2025u). Such externalization turns procedural memory from private agent state into portable, human-readable, and versionable infrastructure that can be inspected, shared, and reused across agents and models. From the self-evolving perspective, the skill library itself then becomes the object of evolution: agents revise skills from their own execution traces through iterative diagnose-and-revise cycles (Belikova et al., 2026), autonomously maintain library health by merging, repairing, or retiring skills to contain accumulated skill-level technical debt (Song et al., 2026), and learn which memory skills to induce and evolve over long horizons (Zhang et al., 2026b). Two caveats temper this trend. Empirically, self-generated skills still underperform human-curated ones, suggesting that fully autonomous skill induction remains an open challenge for self-evolving agents (Li et al., 2026); moreover, as skills circulate as shared artifacts, they introduce new security and governance risks, with a substantial fraction of community-contributed skills found to contain vulnerabilities (Liu et al., 2026).

In summary, procedural memory complements semantic and episodic memory by providing reusable action abstractions. It is currently undergoing a pivotal transition from explicit non-parametric instruction retrieval to implicit parametric neural strategies. This evolution not only supports skill acquisition but also ensures the stable and efficient implementation of complex decisions in long-horizon autonomous agents. At the same

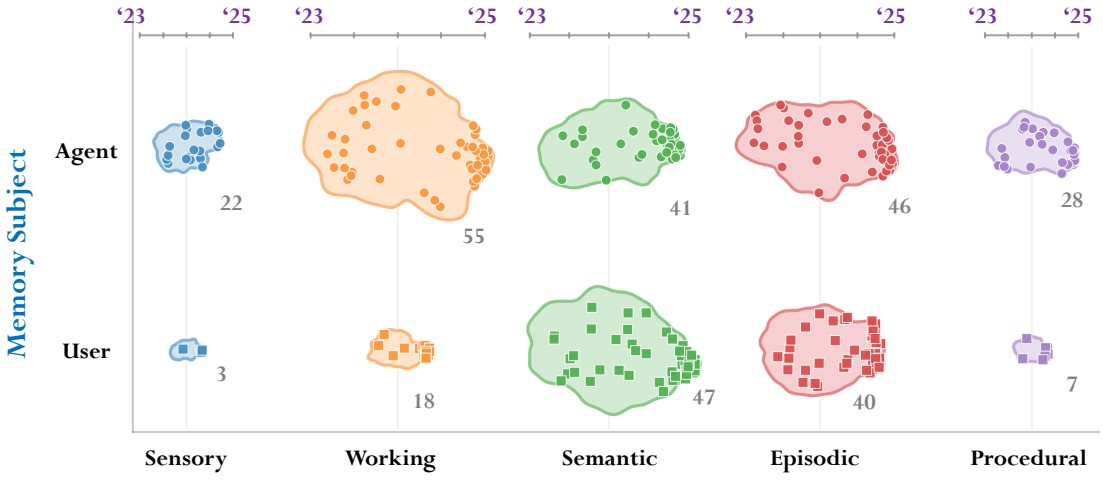

Figure 5: Connections between memory cognitive mechanism and memory subject. The figure is organized as a grid: columns correspond to the five cognitive mechanisms (§3.2), sensory, working, semantic, episodic, and procedural, while the two rows correspond to agent-centric (upper; §3.3.2) and user-centric (lower; §3.3.1) memory subjects. Each dot represents one paper, positioned along the horizontal axis by publication time (2023 to 2025). The cluster area is proportional to the paper count in each cell.

time, the rapid rise of the agent skills ecosystem highlights a complementary countertrend in which procedural knowledge is externalized as explicit, portable, and shareable artifacts; enabling agents to autonomously induce, evolve, and safely govern such skills, rather than relying on human curation, stands out as a central challenge for self-evolving agents.

## 3.3 Memory Subjects

From a self-evolving and long-horizon perspective, memory subjects characterize whom the memory primarily models and serves. This dimension is orthogonal to memory substrate and cognitive mechanism, and is critical for understanding the optimization goal of the memory in foundation agents. Figure 5 illustrates how this subject-level distinction connects with cognitive memory mechanisms. Working memory, procedural memory, and sensory memory are predominantly agent-centric, reflecting their roles in supporting many real-world downstream tasks. Semantic and episodic memory appear in both user- and agent-centric settings. For users, they encode longitudinal interaction histories and evolving preferences, whereas in agent-centric memory, they support world modeling, long-horizon task execution, and continual experience accumulation. Together, we may find that memory subjects interleave with cognitive categories, offering a complementary perspective for organizing memory research beyond the single classifications as in prior surveys (Zhang et al., 2025q; Du et al., 2025) alone. We note that user-centric and agent-centric memory are conceptual orientations rather than mutually exclusive categories. In practice, a single system may maintain user-specific preferences alongside reusable task strategies. For example, a coding assistant may simultaneously maintain user-specific coding style preferences (user-centric) and reusable debugging strategies accumulated across many users (agent-centric). Recent work such as MemSkill (Zhang et al., 2026b) evaluates memory across both orientations. However, current benchmarks rarely require both simultaneously, which naturally leads existing research to emphasize one over the other.

### 3.3.1 User-Centric Memory

> **User-Centric Memory**
>
> *User-centric memory* refers to an abstraction of **user-specific facts and preferences**, including biographical data, historical interactions, and expressed preferences, that could evolve across sessions and domains, to align with the user for coherent interactions and assistant task execution.

User-centric memory constitutes a foundational component of LLM agents for user-related tasks, particularly long-term personalization (Liu et al., 2025e; Chen et al., 2024b; Zhang et al., 2025o). Beyond general instruction prompts, which only inform temporal task execution status, user-centric memory underpins the agent's actions and responses by grounding them in the user's identity and evolving context across both short-term interactions and long-term histories (Tan et al., 2025c). This component is particularly critical in domains, including but not limited to counseling (Litam et al., 2021), recommendation (Chen et al., 2018; Zhang et al., 2025m; Shao et al., 2026), and personal healthcare (Velliangiri et al., 2022; Zhang et al., 2026c), where the agent must demonstrate both immediate and longitudinal awareness of the user's evolving preferences, intentions, and behavioral traits. Below, rather than imposing a strict taxonomy, we highlight the key objectives that accurate and persistent user-centric memory serves, enabling agents to continuously refine their understanding of users and maintain personalized alignment over long interaction horizons.

**Memory Management in Long-Horizon Dialogues.** The dialogue system is one of the most popular real-world application scenarios for user-agent interactions (Chen et al., 2017; Xu et al., 2025e). Within the LLM's finite context window, existing dialogue systems can generally only process multi-turn conversations in limited session contexts (Li et al., 2025a). Prompting LLM agents with all details in prior conversations is computationally infeasible and often counterproductive due to noise accumulation and topic drift. Effective user memory thus requires principled mechanisms for relevant conversation context retrieval (Maharana et al., 2024), update (Xu et al., 2025e), summarization (Chhikara et al., 2025; Salama et al., 2025), and forgetting (Zhong et al., 2024). Recent real-world memory systems such as MemGPT (Packer et al., 2023) explicitly formalize memory hierarchies to manage long-term conversational state. In parallel, OpenAI's ChatGPT memory feature implements a persistent memory mechanism that retains user-relevant details while giving users explicit control over what is stored or forgotten (OpenAI, 2025). These developments illustrate a central challenge in dialogue memory management: determining which pieces of information accumulated throughout long-horizon user-agent interactions should be retained, updated, or forgotten to support future dialogues. Such information may include persistent preferences, evolving goals, significant life events, and sensitive constraints. Consequently, memory management is not merely the preservation of past conversations, but a continual process through which the agent maintains a coherent yet adaptable representation of the user over time.

**Persistent User Simulation.** High-fidelity user simulators are essential for realistic interactive online platforms because they provide scalable, reproducible, and controlled environments for training and evaluation, without accessing extensive real-user data, thereby respecting privacy and reducing costs associated with live user studies and online A/B testing (Park et al., 2023; Zhu et al., 2024; Samuel et al., 2024; Zhang et al., 2025s). In online digital platforms such as recommender systems (Zhu et al., 2025a) and social networks (Gao et al., 2023), simulators help approximate long-term user behaviors and preferences, enabling evaluation of policy optimization, ranking strategies, and interventional impacts under dynamic conditions rather than static test collections (Jin et al., 2013; Zhang et al., 2025k). In such scenarios, user-centric memory supports both longitudinal consistency and fine-grained preference evolution, as simulators need to maintain personalized profiles and adaptive interaction patterns over extended horizons. Rather than reproducing a fixed persona, memory-enabled simulators should model how user states, preferences, and behaviors evolve in response to accumulated experiences and agent interventions.

**Long-Term Personalization.** Unlike user simulation, which primarily targets group- or platform-level benefits, and dialogue systems, which mainly capture preferences expressed within user-agent conversational contexts (Zhang et al., 2025n), long-term personalization focuses on optimizing the experience of a specific user over extended time horizons spanning days, months, or even years. By maintaining an explicit user

profile or persistent personal knowledge base across sessions (Zhang et al., 2026d), agents can adapt linguistic style, decision-making, and behavior in a manner that consistently supports the needs and preferences of the same individual over time. Importantly, such personalization should not treat the user profile as a static record. Instead, agents must continually reconcile historical evidence with newly observed interactions, distinguish persistent preferences from temporary states, and revise their memory when the user's goals or behaviors change. This process enables self-evolving preference alignment, in which the agent progressively adapts its understanding and behavior while preserving longitudinal coherence. Early work in long-term personalization (Salemi et al., 2024) generates contextually relevant responses through episodic memory construction. Subsequent research (Tan et al., 2024a) further explores encoding personal memory into parameter-efficient modules, such as LoRA-based parameters (Hu et al., 2022). However, these approaches either struggle to fully exploit rich personal user data or incur substantial computational overhead. More recent frameworks therefore emphasize efficient memory mechanisms that can reconcile newly observed behaviors with existing memory structures for personalized alignment (Cai et al., 2025; Zhang et al., 2025o). This direction shifts personalization from one-time profile construction toward continual, memory-driven adaptation throughout the user-agent relationship.

**Privacy-Preserving Memory.** Persistent user memory introduces substantial demands in privacy, security, and data governance, because sensitive attributes may be stored, retrieved, and inadvertently exposed through both training-time memorization and inference-time context leakage in memory-augmented agents (Mireshghallah & Li, 2025). Recent work has systematically demonstrated that memory modules in LLM agents are vulnerable to targeted extraction attacks (Wang et al., 2025a), and one can easily recover private user data stored in agent memory under a black-box threat model. In multi-agent settings, privacy risks are further compounded by heterogeneous agent roles and dynamic collaboration, which complicates the enforcement of consistent privacy protocols across interacting memory banks (Shi et al., 2025b). These risks become particularly significant for self-evolving agents because continual personalization requires memory to be repeatedly retrieved, updated, consolidated, and potentially shared across long-running interactions. To ensure safe deployment, agents must support selective memory retention, secure storage, user-controlled deletion, and transparent auditing of what information is remembered or forgotten. They should further provide users with control over how their evolving profiles are inferred and updated, rather than only allowing control over individual stored records. Practically, these safeguards are often combined with differential privacy mechanisms (especially in personalized or federated adaptation), encryption-based storage and retrieval through private vector databases, and explicit retention or access-control policies to mitigate leakage risks from both stored content and embeddings (Tran et al., 2025; Shi et al., 2025b).

### 3.3.2 Agent-Centric Memory

> **Agent-Centric Memory**
>
> *Agent-centric memory* refers to **distilled knowledge, skills, and operational task priors** that an agent accumulates through its own history of task execution or gained via environment interaction. This supports long-context, long-horizon, and long-running tasks across real-world environments.

Unlike user-centric memory, which preserves personalized information mainly about a corresponding user, agent-centric memory encodes more general lessons and experiences derived from the agent's own history of solving and interacting with real-world tasks (Luo et al., 2025; Shinn et al., 2023; Wang et al., 2025c; Zhang et al., 2025i). This memory captures lessons that are generally applicable rather than tied to any single user. In essence, it represents how an agent "learns from experience and environments," retaining important facts, strategies, and world knowledge gained through previous experiences to improve future performance (Wei et al., 2025e;d; Zhang et al., 2025i). Different from user-centric memory, which optimizes satisfaction for a particular user, agent-centric memory addresses broader real-world problems, where experiences and solutions are expected to remain useful across different tasks, environments, or users. This is crucial for long-horizon autonomy: an agent tackling complex multi-step tasks or lifelong learning (Zheng et al., 2025c;b) must be able to remember and build upon what it has encountered before. Beyond preserving experience, agent-centric memory provides the foundation for self-evolution: agents can distill successful and failed trajectories into domain knowledge, operational strategies, and executable skills; refine them through subsequent interactions;

and transfer reusable abstractions to previously unseen tasks. Below, we outline the key motivations and scenarios that necessitate agent-centric memory approaches.

**Long-Horizon Task Execution.** LLM agents frequently engage in tasks requiring hundreds or thousands of reasoning and action steps, such as coding (Jimenez et al., 2024), web navigation (Zhang et al., 2025l; Zhou et al., 2024), complex multi-turn decision-making (Shani et al., 2024), or sequential tool use (Qin et al., 2024). In these settings, immediate working memory is easily overwhelmed by accumulated observations and intermediate reasoning. Agent-centric memory provides an externalized mechanism for storing and retrieving key intermediate states, enabling agents to operate beyond their native context window. Such memory directly supports execution within an ongoing task by preserving subgoals, action outcomes, environmental states, unresolved failures, and dependencies among temporally distant steps. It therefore enables an agent to resume interrupted workflows, revise earlier decisions, and maintain coherent progress throughout long-running interactions. For instance, MEM1 (Zhou et al., 2025c) introduces an end-to-end reinforcement learning (RL) framework that maintains a compact internal state, enabling agents to consolidate relevant information while discarding redundant context and thereby operate with near-constant memory usage across arbitrarily long, multi-turn interactions. Complementary to MEM1, MemAgent (Yu et al., 2025b) proposes an RL-based memory agent tailored for long-text processing. It reads long inputs in segmented chunks and uses a fixed-length, overwritable memory that is updated incrementally. This design enables LLM agents to scale to extremely long contexts with linear complexity and minimal performance degradation. More recent approaches include hierarchical memory modules (Hu et al., 2025b), context folding schemes (Sun et al., 2025b), and learned memory controllers (Zhang et al., 2025p) that decide what to store and when to compress or discard outdated information. Collectively, these approaches extend memory beyond passive context retention toward active state management for sustained agent execution.

**Domain-Specific Long-Tail Solution Iteration.** Many real-world problems exhibit long-tail phenomena (Zhang et al., 2021; Kandpal et al., 2023; Park & Tuzhilin, 2008), where rare but important patterns, error cases, or domain-specific heuristics occur infrequently in the training data. Agent-centric experience memory supports the retention of these rare insights and knowledge, enabling agents to reuse them efficiently when similar or related cases arise in the future (Li et al., 2024a). Through repeated interactions within a domain, agents can progressively consolidate one-off solutions into domain-specific knowledge and refine their strategies according to observed successes, failures, and environmental feedback. This enables self-evolution beyond the knowledge and behaviors initially acquired during pretraining. For example, in software debugging (Jimenez et al., 2024), most errors follow common patterns, yet real-world systems often fail due to highly specific configuration issues, dependency conflicts, or environment-dependent bugs. Similarly, in scientific research (Ghafarollahi & Buehler, 2025), while general reasoning patterns and experimental procedures are often shared within a discipline, many sub-area-specific experimental setups (e.g., specialized channel-coding configurations in wireless communications) or highly domain-dependent troubleshooting practices (e.g., field-specific protocols in archaeology) are rarely encountered across researchers and are therefore unlikely to be sufficiently represented in pretraining data. Comparable long-tail dynamics also arise in complex information search (Wei et al., 2025a) and professional workflow automation (Zhang et al., 2025g), where agents must address narrowly scoped, context-dependent problems that benefit from storing customized one-off solutions and reapplying them over time (Yang et al., 2025a). As these experiences accumulate, memory allows the agent to update domain-specific heuristics, correct ineffective strategies, and develop increasingly specialized competence over long operational horizons.

**Cross-Task Skill Generalization.** Long-term memory enables agents to accumulate durable knowledge across tasks and episodes, supporting continual improvement without catastrophic forgetting (Hatalis et al., 2023). At a high level, cross-task skill generalization concerns how agents abstract and retain reusable knowledge and skills from diverse interaction trajectories, enabling generalization across heterogeneous tasks, domains, and environments rather than improving execution within a single setting or environment. For example, Agent KB (Tang et al., 2025b) constructs a cross-domain experience framework that aggregates high-level strategies and execution lessons distilled from heterogeneous agent trajectories into a shared knowledge base, enabling agents to retrieve and reuse transferable problem-solving knowledge when facing novel tasks across different domains. Unlike specific trajectory replay, the goal of cross-task memory is to

distill interaction experience into task-agnostic abstractions and reusable skill primitives that generalize across environments and objectives. Another representative example is the action–thought patterns used across WebShop (Yao et al., 2022) and ALFWorld (Shridhar et al., 2021) in ReAct (Yao et al., 2023). Such abstractions also include reusable tool-use patterns, such as stable tool-use strategies in Toolformer (Schick et al., 2023) and ToolLLM (Qin et al., 2024), and error-avoidance execution heuristics accumulated through repeated failures (Shinn et al., 2023). Rather than memorizing complete solutions, agents can identify recurring subproblems, extract invariant action patterns, and recombine previously learned skills to address new tasks. This enables LLM agents to progressively evolve into more capable and efficient problem solvers, exhibiting behavior that mirrors human-like expertise development across diverse tasks over time.

**Continual Strategy and Skill Evolution.** Complementary to cross-task generalization, continual strategy and skill evolution focuses on how environment-grounded procedural memories are created, refined, composed, and maintained through repeated execution. These memories encode how to efficiently execute multi-step actions within a specific interaction regime, such as a web interface, GUI system, or physical environment. For instance, web agents (Wei et al., 2025e) in environments such as WebArena (Zhou et al., 2024) learn to reuse and refine successful multi-step browsing policies rather than re-exploring interfaces from scratch. For GUI agents, such procedural memory stores interface-specific action sequences, including menu traversal, widget manipulation, and error-recovery strategies under the constraints of desktop or mobile environments (Qin et al., 2025; Wang et al., 2025e). In embodied agents, procedural memory manifests as executable skills and control policies that respect physical dynamics and action preconditions (Fung et al., 2025; Yang et al., 2025c). These stored experiences can be reused as templates, demonstrations, or priors for solving tasks more efficiently. However, self-evolving agents must move beyond simply storing successful trajectories: they need to induce skills from experience, evaluate their effectiveness, update them when environments change, decompose or compose them at different levels of abstraction, and remove obsolete or conflicting skills. Memory-based skill learning thus allows agents to refine effective behaviors over repeated episodes and internalize world models at the level of action–outcome regularities. Over time, the resulting skill library serves as an evolving procedural substrate that supports both increasingly efficient execution within familiar domains and adaptive generalization to new tasks. This capability is central to long-horizon autonomy and forms the basis for emerging research in lifelong learning (Zheng et al., 2025c) and long-running agents (Yang et al., 2025a).

## 4 Memory Operation Mechanism

Beyond how memory is represented and what cognitive role it plays, a memory system is defined by the operations that act on it. Figure 6 summarizes these operations for both single- and multi-agent settings. Section 4.1 covers the five core operations of a single agent, and Section 4.2 extends them to multi-agent systems, where architecture, routing, and isolation govern how experience is shared and kept consistent across agents.

### 4.1 Memory Operations for Single-Agent Systems

In a single-agent system, memory operation mechanisms define how a foundation agent actively constructs, updates, controls, and utilizes memory throughout long-horizon interaction and task execution. Rather than treating memory as a static repository, modern agents manipulate memory through a sequence of operations, including indexing, retrieval, updating, compression, summarization, forgetting, and pruning. Beyond managing information, self-evolving agents further consolidate interaction trajectories, reflect on execution outcomes, and distill recurring experience into reusable knowledge, strategies, and skills. These operations collectively regulate how past experience is incorporated into ongoing reasoning and future decision-making, forming the operational backbone of single-agent memory systems. Together, they establish a continual adaptation loop in which an agent retrieves relevant experience to guide current execution, evaluates resulting successes and failures, and subsequently revises its memory and skill repertoire. This loop allows the agent to maintain coherent state within long-running tasks while progressively improving across tasks and interaction episodes.

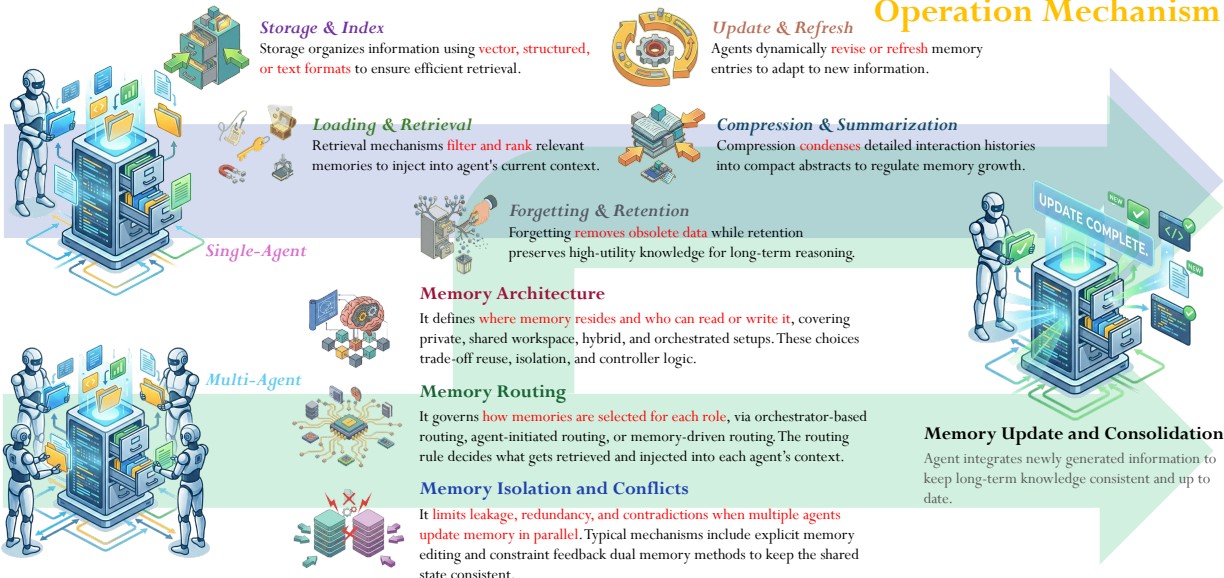

Figure 6: **The Operation Mechanism of the Foundation Agent Memory System**. The diagram illustrates the complete operation mechanism of the foundation agent memory system. For single-agent systems, it defines five core operations: *storage & index*, *loading & retrieval*, *update & refresh*, *compression & summarization*, and *forgetting & retention*, that govern how historical information is preserved and accessed to support downstream work. For multi-agent systems, the framework extends to address coordination challenges through *memory architecture definitions*, *routing protocols*, and *isolation and conflict resolution strategies*, ensuring data consistency and efficient collaboration across distributed agents.

### 4.1.1 Storage and Index

As an agent's memory grows over time, how information is stored and organized becomes essential for ensuring that relevant information can be efficiently and reliably retrieved when needed. In single-agent systems, memory is typically indexed at write time by associating each entry with semantic embeddings and auxiliary metadata such as timestamps, task identifiers, entities, or tool usage (Lewis et al., 2020; Zhang et al., 2023a; Rezazadeh et al., 2025b). Vector-based storage remains the dominant paradigm in non-parametric agent memory, enabling efficient approximate nearest-neighbor search over episodic or semantic memory in retrieval-augmented agents (Guu et al., 2020; Quinn et al., 2025). Beyond flat vector indices, agents increasingly adopt structured storage formats, including relational tables, graph-based memory, and hierarchical tree representations, which enable relational queries, multi-level abstraction, and schema-aware access aligned with task structure (Anokhin et al., 2024; Sarthi et al., 2024; Rezazadeh et al., 2025b). In parallel, some systems maintain text-record memories that store explicit, human-readable summaries and chronological logs, relying on keyword matching or lightweight string-based retrieval to prioritize transparency and controllability (Park et al., 2023; Zhang et al., 2025q). Finally, storage is not limited to external memory modules: parametric memory embedded in model weights, transient working memory in the context window, and inference-time internal states such as KV caches collectively serve as implicit storage substrates that influence retrieval behavior and reasoning dynamics (Mallen et al., 2023; Kwon et al., 2023; Pope et al., 2023). As memory scales across longer horizons, the choice of storage format and indexing strategy directly affects retrieval precision, computational overhead, and downstream reasoning reliability.

### 4.1.2 Loading and Retrieval

To utilize stored experience during ongoing reasoning and decision-making, an agent relies on retrieving task-relevant memories while limiting the influence of irrelevant or outdated information. In single-agent systems, this process typically begins with lightweight loading operations that filter or preselect memory entries based on metadata such as recency, task scope, or memory type, followed by similarity-based retrieval over vectorized representations (Lewis et al., 2020; Mallen et al., 2023; Zhang et al., 2025q). Loaded memories are then ranked or refined using semantic similarity, heuristic constraints, or budget-aware selection strategies before being injected into the model's prompt or working context, forming the primary interface between external memory and the LLM's inference process (Park et al., 2023; Wang et al., 2025q). Prior studies indicate that retrieval quality has a substantial impact on agent performance. Retrieving excessive memory can introduce noise and context overload, while overly restrictive retrieval may prevent access to critical historical information (Xu et al., 2025e; Hu et al., 2025c). Consequently, effective loading and retrieval mechanisms aim to balance relevance, diversity, and context budget to support coherent long-horizon behavior.

### 4.1.3 Update and Refresh

As agents interact with their environment over extended horizons, previously stored memory may become incomplete, outdated, or misaligned with newly observed information, making static or append-only memory representations insufficient. Update mechanisms enable a single agent to revise existing memory entries in response to new observations, external feedback, or reflective reasoning, allowing memory content to evolve rather than merely accumulate. In practice, updates are often triggered after task completion, evaluation signals, or detected inconsistencies, and may involve rewriting semantic summaries, merging overlapping episodic records, or adjusting the importance of stored information (Park et al., 2023; Tan et al., 2025c; Wang et al., 2023a). In parallel, *refresh* operations focus on adjusting the relative prominence and accessibility of memory without altering its core content, such as re-ranking salient entries, regenerating condensed summaries, or reinforcing frequently accessed memories to preserve their influence over future reasoning. Recent agent systems further demonstrate that reflective or self-evaluative processes can autonomously trigger both updating and refresh actions, leading to improved long-term coherence and task performance (Shinn et al., 2023; Ouyang et al., 2025). Together, these mechanisms allow memory representations to evolve dynamically, supporting adaptation in non-stationary environments while mitigating the accumulation of obsolete or misleading information.

### 4.1.4 Compression and Summarization

Long-horizon agent memory systems require mechanisms that regulate growth while preserving information essential for future reasoning and decision-making. Compression and summarization address this need by converting fine-grained episodic records into more compact and abstract representations that reduce redundancy and improve memory efficiency. In practice, many agent systems perform summarization periodically or after task completion, consolidating interaction histories into semantic or hierarchical memory suitable for long-term storage (Wang et al., 2025j; Chen et al., 2025d). Hierarchical consolidation further organizes compressed memory into multi-level or tree-structured representations, enabling scalable retrieval across different abstraction levels (Sarthi et al., 2024; Rezazadeh et al., 2025b). Dynamic Cheatsheet further instantiates this idea by maintaining a compact, task-adaptive summary that is continuously updated to surface only the most salient information for ongoing reasoning, reducing repeated large-scale retrieval and context expansion at inference time (Suzgun et al., 2025). While these mechanisms improve context utilization and scalability, prior work highlights an inherent trade-off between abstraction fidelity and long-term recall, making summarization strategies a critical design consideration in single-agent memory systems (Lee et al., 2024; Wu et al., 2025d).

### 4.1.5 Forgetting and Retention

In agent systems, memory relevance evolves over time as task objectives change and environments become non-stationary, making indiscriminate memory accumulation increasingly misaligned with effective long-term reasoning. Forgetting mechanisms address this challenge by reducing the influence of obsolete or low-utility

information through explicitly removing or suppressing memory entries that are no longer aligned with an agent's ongoing objectives. In practice, forgetting is commonly implemented via heuristic policies such as recency-based decay or importance thresholds, as well as learned strategies that optimize memory removal under explicit resource or efficiency constraints (Alla et al., 2025; Wang et al., 2025q). In contrast, retention mechanisms determine which memories should be preserved and prioritized over extended interaction horizons, ensuring that task-relevant knowledge remains accessible despite continual memory growth. Recent work emphasizes adaptive retention strategies, where agents dynamically adjust preservation priorities based on task context, feedback signals, or long-term performance objectives, enabling robust behavior in non-stationary environments (Yan et al., 2025b; Zheng et al., 2025a).

## 4.2 Memory Operations for Multi-Agent Systems

In a multi-agent system, memory operation mechanisms describe how several agents build and reuse memory together during collaboration: each agent can keep its own private memory, and they can also share experience through a shared workspace. As in single-agent systems, this mechanism still includes basic operations such as indexing, retrieval, updating, compression, summarization, forgetting, and pruning. It may additionally support the aggregation and distillation of experiences contributed by multiple agents into shared knowledge and skill repositories, allowing useful strategies discovered by one agent to benefit other agents or future collaborations. Beyond these basic operations, what matters more in multi-agent settings is cross-agent read and write rules: in each task, the system needs to choose suitable memory for agents with different roles. Such rules determine not only which memories each agent can access, but also how role-specific experience, specialized skills, and execution feedback are transferred across agents. In addition, the system often needs extra operations to remove redundancy, resolve conflicts, and keep the memory consistent. These operations must account for the provenance, reliability, and compatibility of memories generated by heterogeneous agents, especially when their observations or learned strategies conflict. Through selective sharing, validation, consolidation, and revision, multi-agent memory can support collective self-evolution while preserving specialized capabilities and coherent long-horizon collaboration.

### 4.2.1 Memory Architecture

In multi-agent systems, memory can be organized in different ways, varying in which layer it is stored, how read and write permissions are defined across agents, and whether the system introduces a controller. These design choices determine whether routing can construct an efficient memory view and whether systemic issues such as information leakage may arise.

**Private-only.** In a private-only architecture, each agent has its own independent memory, and its read and write rights only take effect inside that agent's memory. In multi-agent collaboration, agents can only rely on their own memory and the current task context to work. This setup gives strong isolation and makes the system easier to check. However, the same memories are often created again in multiple private spaces, which can waste resources. Representative examples include RecAgent (Wang et al., 2025h), which instantiates one agent per user and keeps a private memory to avoid mixing different users' histories to better protect privacy; TradingGPT (Li et al., 2023b), where each trading agent maintains its own memory so it can follow a consistent risk preference and sector focus, and collaboration mainly happens through exchanging selected viewpoints rather than sharing full memories; and MetaAgents (Li et al., 2025j), which equips each role with an isolated memory of its past thoughts and decisions so the role stays stable and consistent, while any information gained from dialogue is written back locally.

**Shared-workspace.** Unlike private-only memory, shared-workspace designs use a common pool that all agents can read and write. Agents share intermediate results through this shared state, so they do not need heavy peer-to-peer messaging, which reduces communication cost. However, the shared pool may quickly become noisy and therefore requires filtering mechanisms, as well as coordination strategies to avoid conflicts when multiple agents update the same information. As representative shared-workspace designs, MetaGPT (Hong et al., 2024) uses a simple shared pool to publish role agents' intermediate artifacts, and each agent applies its role profile to filter the pool and pull only relevant memory into context. InteRecAgent (Huang

et al., 2025b) makes the shared state more task-specific by establishing a Candidate Bus: tools repeatedly read the current candidate set and write back filtered candidates, so the set is progressively narrowed, which can avoid prompt length overflow. MAICC (Jiang et al., 2025d) further scales the workspace into a shared experience pool with offline data and an online replay buffer, where agents query with their current sub-trajectory and retrieve top-$k$ similar trajectories to reuse in-context.

**Hybrid.** This setup keeps both a private layer and a shared layer. It uses a policy to decide whether new information is written to the private space or to the shared space. At each call, it builds a permission-limited memory view for the agent, so the agent only sees what it is allowed to see. This gives a balance between maximizing memory reuse and keeping sensitive information isolated. For example, in Collaborative Memory (Rezazadeh et al., 2025a), the system separates all memory fragments into private memory and shared memory. During writing, a write policy decides whether the information is generally useful or user-specific, and then writes it to the shared or private layer. During reading, a read policy uses an access graph provided by the system to build a permission-limited memory view for the current call. Similarly, in MirrorMind (Zeng et al., 2025), each Author Agent maintains a private memory corresponding to their research interests, while sharing foundational disciplinary knowledge through a public memory. This hybrid design forms the structure of an AI Scientist.

**Orchestrated.** The previous three categories mainly differ in where memory lives and who can see it. By contrast, orchestrated designs introduce an explicit controller that coordinates agents in a hierarchical workflow and mediates memory access. Concretely, the controller decomposes the task, assigns subtasks to role agents and decides what each agent can read or write. This centralized coordination is well-suited for multi-stage workflows and strongly constrained settings, but it may also introduce a bottleneck and additional system complexity. ChatDev (Qian et al., 2024a) exemplifies this pattern by running role agents under a predefined ChatChain (design, coding, testing), where stage outputs serve as structured handoffs to reduce cross-stage context overload. MIRIX (Wang & Chen, 2025) similarly orchestrates memory maintenance: a Meta Memory Manager routes updates/retrieval to specialized Memory Managers and aggregates their results into a unified response. Notably, this control layer is orthogonal to storage layout and can be combined with private-only, shared-workspace, or hybrid memory stores. For instance, MGA (Cheng et al., 2025) organizes GUI interaction as an observe, plan and ground agent pipeline. A Planner acts as the controller, while lower-level agents can submit intermediate states in a shared workspace, with the planner selecting what history is retrieved and injected at each step. Similarly, AgentFlow also combines orchestration with a shared workspace: a planner controls all modules and writes key intermediate results into a shared memory for later retrieval (Li et al., 2025l).

### 4.2.2 Memory Routing

Given an architecture, routing describes a set of memory allocation and access rules. When a new task arrives, the system needs to build a separate memory view for agents with different roles: it decides which past memories should be retrieved and how they should be injected into each agent's context. We group routing methods by where the routing decision is made: a central orchestrator, individual agents, or the memory store itself via retrieval and matching.

**Orchestrator-based Routing.** This refers to a setting where a centralized orchestrator makes routing decisions in a unified way. It maintains the global task state and collaboration progress, breaks a complex goal into subtasks, and then assigns subtasks to different worker agents based on their roles and abilities, while also distributing the required memory and deciding the execution order. These decisions can be updated dynamically as the task state changes. This method emphasizes a centralized global workflow, but the cost is that the orchestrator may become a bottleneck for performance and robustness, creating a single point of load and failure risks. For example, in LEGOMem (Han et al., 2025), the orchestrator keeps the global state, generates the next subtask, selects an agent to execute it, and updates the state using the agent's summary; memory is scheduled in the same way, with task memories injected to the orchestrator for planning and subtask memories routed to the selected agent for execution. GameGPT (Chen et al., 2023) shifts the focus to workflow routing: a manager defines a multi-stage pipeline and requires each stage to write key

intermediate outputs into a shared workspace, so later stages can inherit and reuse them. Finally, Westhäußer et al. (2025) extends orchestration to memory-source routing, where the orchestrator using MCP selects which memory sources to call and injects the most relevant snippets; if the injected evidence is insufficient, the Self-Validator asks for more retrieval and updates the context, enabling centralized memory control.

**Agent-Initiated Routing.** Compared with orchestrator-based routing, in this method routing decisions are not assigned by a centralized orchestrator. Instead, each agent initiates them locally based on its role and task. Information is usually published into a shared memory pool first, and then agents use constraint mechanisms to select the memories they need, forming their own memory views. This method is often more flexible, but it also depends more on good filtering design to avoid noise, conflict, or missing important information in the shared pool. In SRMT (Sagirova et al., 2025), each agent keeps a personal memory vector and decides locally how much to read from shared memory: at each step it uses cross attention over its memory to select from the shared memory sequence and updates its personal memory. Taking a more explicit filtering route, $S^3$ (Gao et al., 2023) treats shared memory as a platform-wide message stream and scores items with factors such as forgetting, relevance, and source credibility, retaining only a small subset as the agent's own memory view. In the Talker-Reasoner setup (Christakopoulou et al., 2024), a shared store (*mem*) is written by the Reasoner and read by the Talker, and the Talker can choose what to read or wait for a fresh update before replying, showing that an agent can decide when to read from memory.

**Memory-driven Routing.** Different from the above two, routing here is mostly done by retrieval from the memory store. The system represents the current task as a query, and then performs "retrieval, scoring and reranking, selection" in the memory store to obtain the most relevant subset of memories and inject it into the context. Sometimes it can also use structured links between memories (like graph-based expansion) to extend the retrieved results into a more complete set of experience pieces. As a typical form of memory-driven routing, G-Memory (Zhang et al., 2025c) organizes multi-agent histories as a hierarchical graph, retrieves relevant nodes for a new task, expands them through neighborhood links, and compresses the result into a core subgraph before trimming it into role-specific memory views. CRMWeaver (Lai et al., 2025) instead routes at the level of reusable guidelines: it retrieves the most relevant workflow guideline from past successful trajectories and injects it into the current context, and writes back a new guideline when no match exists. Finally, in Spark (Tablan et al., 2025), each coding problem is treated as a query, and a retrieval agent analyzes intent and selects the most relevant documentation and past experience traces from a shared store.

### 4.2.3 Memory Isolation and Conflicts

Building on multi-agent architecture and memory routing, memory isolation is very important because memory is read and written by multiple agents rather than updated in a single, sequential loop. Multiple agents may produce conclusions in parallel, and their accessible resources and permissions often change over time. If the system writes all the information into the same shared pool without any separation and makes it visible to every agent, there might be consistency conflicts: different agents may write facts that contradict each other, and outdated information may not be removed and still retrieved later, which can mislead reasoning. Agents Thinking Fast and Slow (Christakopoulou et al., 2024) reports that the Talker may produce wrong or rushed answers because it can read an outdated belief state before the Reasoner updates it. Here, memory conflicts are mainly handled in two ways: controlling at write time, or gradually improving consistency through an iterative loop.

**Write Control for Memory Isolation.** A direct strategy is to enforce isolation at the memory write and update stage. In each interaction round, an agent first compares newly extracted candidate facts against the existing memory state and selectively updates memory through a controlled evaluation mechanism, rather than blindly appending new information. In Memory-R1 (Yan et al., 2025b), the memory manager agent is the only agent allowed to mutate memory. This includes four kinds of discrete editing actions, namely *ADD*, *UPDATE*, *DELETE*, and *NOOP*. *ADD* writes a new entry; *UPDATE* rewrites or merges an existing entry (especially when the new information is a refinement or correction of the old one, the system tends to keep the version with more information); *DELETE* removes an entry that is clearly contradicted by new evidence or becomes outdated; and *NOOP* means the information is already covered or is not important for

long-term memory, so no update is made. For each user query, the memory manager observes the current memory state and decides which specific memory slots to operate on. As a result, irrelevant memory entries remain untouched. A related form of write-time isolation is adopted in WebCoach (Liu et al., 2025b), where the memory store is updated only after an episode is completed, so partial trajectories are never written into long-term memory, which isolates episodes and prevents memory conflicts.

**Memory Consistency with Feedback Loop.** In contrast to write control, this line of work treats memory conflicts as an iterative optimization process for consistency. Generally, the multi-agent memory system enforces the task's hard requirements in a stable constraint memory that later iterations can consult, and it keeps a growing feedback memory that records failures from earlier rounds to support learning in subsequent iterations. EvoMem (Fan et al., 2025b) is a representative example. Its verifier compares each candidate with the stored constraint memory and outputs a score. The system accepts a solution only when the score reaches one hundred.

### Learning Policy

**Learning policy** refers to how an agent *learns* to manage memory, **what** to store, **when** to store it, **how** to represent it, **when** to retrieve or discard it, and **where** to store or retrieve, rather than relying on fixed, hand-crafted heuristics. Such policies are typically optimized from data or feedback (e.g., supervised signals, reinforcement learning, or self-improvement).

*Prompting*
Designing structured prompts to guide *when* and *how* the agent updates and uses memory, without changing model parameters.

*Fine-Tuning*
Updating model parameters with curated data to internalize effective memory behaviors and improve robustness and consistency across tasks.

*Reinforcement Learning*
Learning memory management through trial and error to increase or balance performance by optimizing long-horizon rewards

Figure 7: **Memory Evolution and Optimization Policies for Foundation Agent Memory System**. We illustrate how learning policies guide agents in deciding what to store, when to store it, how to represent it, and when and where to retrieve or discard memories. The figure summarizes three common approaches, including *prompting*, *fine-tuning*, and *reinforcement learning*, that progressively improve memory decisions from imprecise memory management toward effective and accurate memory management.

## 5 Memory Evolution and Optimization Policies

This section examines how agents acquire policies to evolve and optimize memory through operations including reading, writing, updating, consolidation, abstraction, forgetting, and pruning. Rather than relying on hard-coded heuristics, these policies determine when and what to remember, how accumulated experience should be transformed into structured knowledge and reusable skills, and how existing memory should be revised according to subsequent interactions and feedback. We categorize these policies into three paradigms based on their primary optimization mechanisms: prompting, fine-tuning, and reinforcement learning, as illustrated in Figure 7.

These paradigms enable memory evolution and optimization at different levels. Prompting directly reorganizes and refines external memory during inference; fine-tuning internalizes memory-operation policies into model parameters; and reinforcement learning optimizes memory decisions according to their long-term effects on agent execution and task outcomes.

## 5.1 Prompt-Driven Memory Evolution and Optimization

Prompt-driven memory optimization refers to inference-time strategies that use prompting, reflection, or summarization to manage memory without modifying model parameters, distinguishing it from fine-tuning (§5.2) and reinforcement learning (§5.3). This paradigm parameterizes the memory policy as natural language prompts. The agent executes these prompts to determine when to access, modify, or prune memory. The primary advantages of this approach include the elimination of expensive model fine-tuning and the high interpretability of the policy. We further categorize this paradigm into Static Prompt-based Control (§5.1.1) and Dynamic Prompt-based Control (§5.1.2).

### 5.1.1 Static Prompt-based Control

Static prompt-based control encodes memory policies as fixed, human-designed rules that remain invariant during execution. Memory decisions are specified at design time through prompt templates or predefined schemas, offering strong interpretability and predictable behavior, but lacking the ability to adapt based on interaction feedback or distributional shifts.

Existing work on static prompt-based memory control can be grouped into three design targets: *static memory OS and organization*, where memory is treated as a structured container or an operating system with a fixed form, enforced through hierarchical partitioning, indexing, summarization, or schema-based representations to mitigate long-horizon context degradation (e.g., SCM (Wang et al., 2023a), MemGPT (Packer et al., 2023), LiCoMemory (Huang et al., 2025e), MemoChat (Lu et al., 2023), A-Mem (Xu et al., 2025e), D-SMART (Lei et al., 2025), MemWeaver (Yu et al., 2025d), and Zep (Rasmussen et al., 2025)); *static memory control in single-agent settings*, where access and retention are constrained by persona identities or domain-specific priors encoded in prompts rather than learned relevance signals (e.g., RoleLLM (Wang et al., 2024e), ChatHaruhi (Li et al., 2023a), Mem-PAL (Huang et al., 2025d), WarAgent (Hua et al., 2023), MemoTime (Tan et al., 2025b), FinMem (Yu et al., 2025e), TradingGPT (Li et al., 2023b), and MemoCRS (Xi et al., 2024)); and *static memory assignment and coordination in multi-agent systems*, where memory is distributed across agents through predefined roles, modular decomposition, and structured communication protocols without learning-based coordination (e.g., MIRIX (Wang & Chen, 2025), LEGOMem (Han et al., 2025), G-Memory (Zhang et al., 2025c), MADRA (Wang et al., 2025g), GameGPT (Chen et al., 2023), and ChatDev (Qian et al., 2024a)).

### 5.1.2 Dynamic Prompt-based Control

Dynamic prompt-based control explores whether memory policies encoded in natural language prompts can be adapted at test time based on experience and feedback, without updating model parameters. Rather than fixing memory behavior at design time, this paradigm treats memory control as a language-mediated and continually revisable process.

Existing work in this space centers on a set of closely related research questions. One line of work asks whether memory usage policies can be corrected through reflection on past outcomes, prompting agents to analyze failures or successes and convert these insights into revised memory instructions that guide future behavior (e.g., Reflexion (Shinn et al., 2023), ReasoningBank (Ouyang et al., 2025), WebCoach (Liu et al., 2025b), and QuantAgent (Wang et al., 2024g)). Another line investigates whether memory representations themselves can be dynamically optimized to improve information efficiency under limited context budgets, treating compression, denoising, and structural reorganization as adaptive, prompt-driven processes (e.g., ACON (Kang et al., 2025c), ACE (Zhang et al., 2025j), SeCom (Pan et al., 2025), Nemori (Nan et al., 2025), CAM (Li et al., 2025d), EvoMem (Fan et al., 2025b), and ViLoMem (Bo et al., 2025)). A further question concerns whether dynamic prompting can distill accumulated experiences into reusable procedural knowledge, such as reasoning templates, execution scripts, or tool-usage strategies that generalize beyond episodic recall (e.g., BoT (Yang et al., 2024b), Memp (Fang et al., 2025b), and ToolMem (Xiao et al., 2025)). Despite their adaptivity, these methods remain fundamentally language-mediated and lack explicit credit assignment, limiting their capacity for long-term policy optimization compared to fine-tuning and reinforcement-learning-based approaches.

## 5.2 Fine-Tuning for Parameterized Memory Policies

Beyond prompt-based adaptation, supervised fine-tuning (SFT) internalizes memory policies into model parameters, enabling more stable and reusable memory behaviors. From a policy learning perspective, SFT-based approaches investigate how memory policies are internalized, stabilized, and executed efficiently once embedded into model weights.

### 5.2.1 Policy Internalization into Parameters

A defining characteristic of SFT-based memory control is that memory policies are internalized into model parameters, transforming memory from an external context manipulation problem into a parametric policy representation. Rather than relying on prompts or explicit buffers at inference time, these approaches embed memory-related behaviors directly into the weight space, enabling stable and reusable memory usage across tasks.

Within this paradigm, existing work explores several closely related research questions concerning how memory control policies can be embedded into model parameters and how such internalization should be structured. Some approaches focus on internalizing memory content itself, distilling short-term contextual information into long-term parametric representations (Wang et al., 2024l; 2025p). Others emphasize internalizing memory access and retrieval behaviors by learning parameterized interfaces or lightweight modules that mediate interaction with external or structured memory stores, rather than absorbing raw content into weights (e.g., Memory3 (Yang et al., 2024a) and MLP Memory (Wei et al., 2025c)). A related line of work investigates how such parameterized memory policies can be organized hierarchically, enabling scalable invocation of large memory components while decoupling long-tail knowledge from core reasoning abilities (Pouransari et al., 2025). Despite their differences, these approaches share the objective of shifting memory control from prompt-level manipulation to parameterized decision rules learned through supervision.

### 5.2.2 Parameterized Policy Stabilization and Boundary Control

Beyond internalization, SFT enables the stabilization of memory policies by learning explicit boundaries on what should be written, corrected, or suppressed once memory control is embedded into parameters. Rather than merely expanding memory capacity, these approaches aim to prevent error accumulation, concept drift, and persona inconsistency under long-term use.

A common theme across these works is the use of supervision to regularize memory updates and enforce boundary constraints. Some approaches train models to perform reflection or self-analysis before committing information to memory, encouraging the storage of high-level, noise-resistant representations aligned with task intent (Liu et al., 2023b; Zhang et al., 2025r; Chen et al., 2025d). Others emphasize identifying and repairing erroneous or outdated memory by learning when existing knowledge should be revised or overridden through correction signals, verifier feedback, or routing mechanisms (e.g., WISE (Wang et al., 2024f), SuperIntelliAgent (Lin et al., 2025), and CRMWeaver (Lai et al., 2025)). In long-horizon interactive settings, defensive boundary control further constrains memory updates to preserve role or identity consistency by restricting which experiences can be retained or reused (e.g., Character-LLM (Shao et al., 2023)). Collectively, these methods treat reflection and self-correction not as isolated prompt-level techniques, but as mechanisms for learning stable and bounded memory policies through supervision.

### 5.2.3 Parameterized Policy Efficiency and Retrieval Refinement

Beyond learning what to store and how to stabilize memory, SFT is also used to refine how parameterized memory policies are executed at inference time, particularly during memory reading and retrieval. Rather than relying on exhaustive context access, these approaches treat retrieval itself as a learning policy and optimize how queries are formulated, iteratively refined, and applied to compressed memory representations. Through supervised training, models learn to generate precise retrieval cues for targeted memory access (Qian et al., 2025b), to execute multi-hop or progressive retrieval that refines queries across reasoning steps (Ko et al., 2024), and to optimize compression-aware retrieval by internalizing or reversibly refining memory representations (Cao et al., 2025b; Wang et al., 2025m), thereby improving reasoning robustness while

reducing inference-time overhead. Despite these gains, the resulting retrieval policies remain fixed after training and do not incorporate explicit credit assignment over extended decision horizons.

### 5.3 Reinforcement Learning for Memory Policies

Reinforcement learning (RL) (He et al., 2026) introduces a fundamentally different paradigm for memory control by enabling memory policies to be optimized through interaction and reward feedback. Unlike prompt-based or supervised approaches, RL allows downstream task outcomes to influence earlier memory-related decisions, making memory construction itself a learnable policy. Existing work can be understood as progressively extending the temporal scope over which reinforcement signals shape memory behavior.

#### 5.3.1 Step-Level Memory Decisions

At the shortest temporal scope, reinforcement learning is applied to memory control by treating memory management as a sequence of step-level decisions. In this setting, memory operations are modeled as actions selected by a learning policy and optimized based on their immediate or short-horizon impact on task reward.

One line of work studies how memory editing can be formalized as an explicit action space. Memory-R1 (Yan et al., 2025b) defines atomic memory operations such as adding, updating, deleting, or skipping entries, and learns a memory policy that selects among these actions based on task-level rewards. MemAct (Zhang et al., 2025p) extends this formulation by incorporating finer-grained editing actions, including trimming and summarization, directly into the agent's unified policy space. A closely related problem concerns step-level memory decisions under explicit capacity constraints. MemAgent (Yu et al., 2025b) and RMM (Tan et al., 2025c) address this setting by learning, through interaction-driven feedback, which information should be written into a fixed-size memory buffer when processing extremely long contexts. Mem-$\alpha$ (Wang et al., 2025q) generalizes this paradigm by framing memory construction itself as a sequential decision-making problem, where agents learn, via reinforcement learning, how to populate and update structured multi-component memories (e.g., core, semantic, and episodic memory) to maximize long-horizon task performance. Together, these approaches frame step-level memory control as the optimization of local memory actions through reinforcement signals, without explicitly modeling the long-term effects of memory state construction.

#### 5.3.2 Trajectory-Level Memory Representation

As tasks extend over longer horizons, the value of memory decisions often emerges only through their cumulative influence on future reasoning and action selection. Reinforcement learning enables this setting by allowing delayed task outcomes to shape how trajectory-level memory states are constructed, updated, and maintained by a learning policy.

Within this scope, existing work studies how compact, decision-sufficient memory representations can be learned when interaction histories can no longer be evaluated step by step. Rather than treating memory as a transient buffer, these approaches view trajectory-level memory as part of the agent's Markov state, whose quality is assessed through downstream decision performance (Chen et al., 2025c; Wu et al., 2025d). A closely related question concerns how long interaction histories should be abstracted into such representations. Several studies treat summarization, folding, or compression as policy decisions whose effectiveness can only be evaluated through reinforcement signals propagated from future outcomes (Lu et al., 2025b; Sun et al., 2025b; Li et al., 2025g). Trajectory-level memory also raises the issue of how memory states should evolve over time as new interactions unfold. MemSearcher (Yuan et al., 2025a) addresses this problem by maintaining an iteratively updated compact memory state and propagating advantages across contexts to refine memory representations. Together, these works characterize trajectory-level memory as a learned state representation whose utility is defined by its long-term contribution to decision making under reinforcement learning.

#### 5.3.3 Cross-Episode and Multi-Agent Memory

When memory extends beyond individual trajectories, it no longer serves only immediate reasoning but accumulates experience whose value emerges across repeated episodes or interactions. At this scope, reinforcement

learning becomes particularly valuable, as long-term and cross-episode reward signals provide explicit credit assignment over which memories should persist, adapt, or be revised by the memory policy.

Research at this level focuses on how experience should be represented, reused, and coordinated once memory spans multiple episodes or agents. Rather than preserving raw interaction histories, cross-episode memory aims to distill higher-level decision-relevant knowledge, such as reusable strategies, self-correction rules, or abstracted behavioral patterns, whose utility is evaluated through repeated reinforcement signals. This perspective underlies approaches such as MCTR (Li et al., 2025h) and graph-based experience abstraction (Xia et al., 2025), which encode experience as transferable decision knowledge learned through interaction. Crucially, such experience is retrieved and applied in a context-dependent manner governed by reinforcement learning, as exemplified by reflective retrieval policies in Retroformer (Yao et al., 2024) and Memento (Zhou et al., 2025a). As memory further extends across agents or representation spaces, reinforcement learning enables feedback to propagate beyond individual trajectories. This includes latent or non-textual memory representations such as MemGen (Zhang et al., 2025d), retrieval-path optimization and callback mechanisms in ReMemR1 (Shi et al., 2025a), as well as shared or decentralized memory policies in multi-agent systems such as MAICC (Jiang et al., 2025d) and SRMT (Sagirova et al., 2025). These studies frame cross-episode and multi-agent memory as the broadest scope of reinforcement-learning-based memory control, where memory policies evolve through accumulated interaction and reward feedback rather than predefined rules or one-shot supervision.

## 6 Evolving and Scaling: Memory, Contexts, and Environments

As LLMs and agents are deployed in increasingly realistic settings, the experience and contextual information accumulated during their operation grow rapidly along three scaling axes in open-world environments: interaction horizon, environmental complexity, and the number of interacting environments, tools, and agents. While traditional evaluations often rely on static, context-limited settings that overlook environmental dynamics, real-world utility requires agents to accumulate, retain, consolidate, and update knowledge across extended timeframes and heterogeneous data structures. Simply expanding the context window is insufficient, because long-running interactions introduce redundant, outdated, conflicting, and increasingly heterogeneous experience that must be selectively organized and transformed. This section explores how memory emerges as the essential architectural solution to this scaling challenge, transforming from a simple interaction log into an adaptive substrate for managing growing experience and supporting continual agent evolution. By selectively retaining task-relevant state, abstracting trajectories into reusable knowledge and skills, tracking evolving user preferences, and coordinating experience across agents, scalable memory enables long-horizon execution, persistent personalization, and collective adaptation in open-world environments.

### 6.1 Context-Limited Simple Environments

The majority of LLM and agent benchmarks today are still configured in *context-limited simple environments*, where an agent is placed in a compact and closed world and interacts over only a short-horizon task instance (Hendrycks et al., 2021b) or with synthetic, non-real users (Maharana et al., 2024). While such settings facilitate reproducibility and experimental comparison, they substantially under-specify the memory demands faced by real-world agents. Therefore, strong benchmark performance often reflects proficiency in in-context pattern matching or short-term reasoning, rather than the ability to accumulate, retain, and reuse knowledge across extended interactions and evolving user preferences and contexts. This mismatch leads to a notable utility gap: agents that achieve high scores on existing benchmarks frequently fail to exhibit long-term adaptation, user personalization, or task-specific skill reuse in open-ended, dynamic environments.

A substantial portion of existing LLM and agent evaluations remains confined to static, context-limited environments. Classic general question answering benchmarks, including SQuAD (Rajpurkar et al., 2018), HotpotQA (Yang et al., 2018), and KILT (Petroni et al., 2021), operate over frozen Wikipedia snapshots, while MS MARCO (Craswell et al., 2021), Natural Questions (Kwiatkowski et al., 2019), SearchQA (Dunn et al., 2017), and TriviaQA (Joshi et al., 2017) replace live information access with pre-collected queries and passages. These designs yield stable, bounded, and well-controlled information sources under which modern systems achieve near-saturating performance. However, retrieval and reasoning are strictly episodic and instance-isolated: agents neither maintain cross-query state nor confront temporal drift, source inconsistency,

or evolving knowledge. As a result, such benchmarks impose minimal requirements on persistent memory, primarily testing short-term retrieval and in-context reasoning rather than long-term knowledge accumulation or adaptive context management. Similar limitations extend to tool-use and web interaction benchmarks. Frameworks such as ToolBench (Qin et al., 2024), WebShop (Yao et al., 2022), and WorkArena (Drouin et al., 2024) rely on fixed APIs or self-hosted environments that abstract away authentication, failure recovery, and long-term user or task state. Even systems that interact with the live web, including WebGPT (Nakano et al., 2021) and WebVoyager (He et al., 2024a), remain constrained by restricted browsing interfaces, curated site lists, and strict interaction budgets. Sequential and interactive environments such as ScienceWorld (Wang et al., 2022), ALFWorld (Shridhar et al., 2021), TextWorld (Côté et al., 2018), and Jericho (Hausknecht et al., 2020), along with programming benchmarks like APPS (Hendrycks et al., 2021a), SWE-bench (Jimenez et al., 2024), SWE-Bench Pro (Deng et al., 2025), and SWE-Lancer (Miserendino et al., 2025), increase task complexity but remain episodic, reset-centric, and evaluation-driven. Consequently, these benchmarks systematically fail to assess knowledge accumulation across episodes and long-horizon consistency, thereby obscuring the critical role of memory for robust real-world agent deployment.

## 6.2 Context-Exploded Real-World Environments

Unlike context-limited benchmark evaluations in research settings, real-world deployments expose agents to environments where context scales along multiple axes. It accumulates over long interaction horizons, becomes increasingly complex due to structured and heterogeneous environmental states, and spans multiple environments, agents, and tools. Moreover, real-world context does not merely increase in volume: it continually changes as users revise their preferences, environments evolve, tools produce new observations, and agents acquire new knowledge and skills. Scalable memory must therefore support both the compression of growing context and the continual evolution of the agent's retained state.

### 6.2.1 Evolving with Long-Horizon Interactions

**Evolving User Memory over Extended Interaction Horizons.** In persistent user–agent interactions, agents are required to maintain coherent behavior and decision consistency aligned with the user across extended dialogue horizons. Unlike in short, self-contained dialogues, information introduced in early rounds, such as user preferences, identity attributes, or implicit task constraints, may not immediately influence responses but often becomes decisive in later stages of interaction. Therefore, interaction history functions less as static input and more as an evolving representation of user state that must be selectively maintained and revised over time. This requirement for information persistence arises from the relative stability of some user attributes alongside the gradual evolution of preferences and goals across different turns (Li et al., 2016), the long-term structure of dialogue tasks involving planning or sustained goals, and the importance of identity and goal consistency for user trust and usability (Zhang et al., 2018).

Long-horizon personalization must consequently balance consistency and adaptation: an agent should preserve enduring user characteristics while identifying when recent interactions provide sufficient evidence to update previously inferred preferences or constraints. However, under fixed context-window constraints, long-horizon dialogue systems frequently exhibit failure modes such as early-context forgetting and progressive context drift as interaction length increases (Liu et al., 2024b; Xu et al., 2022a). These failures are not isolated reasoning errors but the cumulative consequence of long-term context mismanagement. They may cause an agent either to forget persistent preferences or to remain anchored to outdated information, both of which undermine continual preference alignment.

Existing mitigation strategies, such as sliding context windows, heuristic truncation, and summary-based memory mechanisms (Park et al., 2023), as well as explicit long-term structures such as user profiles or persona memories (Xu et al., 2022a), improve scalability but often degrade recall reliability. In practice, seemingly peripheral information may be irreversibly discarded despite its potential future relevance, exposing a fundamental trade-off between context-budget control and long-term information accessibility (Liu et al., 2024b). The challenge is therefore not simply to retain more interaction history, but to continually determine which information remains stable, which has become obsolete, and which should modify the agent's current representation of the user.

**Accumulated Execution Memory for Multi-Turn Tool Use.** Context explosion can be exacerbated in agents that rely on multi-turn tool use and reasoning architectures. Beyond conversational history, tool-based agents must retain tool inputs, execution outputs, and intermediate states, many of which are repeatedly referenced in subsequent reasoning steps (Schick et al., 2023). In frameworks such as ReAct (Yao et al., 2023) and Planner–Executor architectures (Wang et al., 2023b), this accumulation is particularly severe: planning traces, tool feedback, and reflective reasoning are explicitly preserved to maintain coherence and decision consistency (Shinn et al., 2023). Consequently, context size can grow rapidly with interaction length and the number or complexity of tool calls.

In ReAct-style agents, explicit reasoning traces themselves become part of the context, enabling interpretability and complex reasoning but simultaneously introducing a substantial and persistent contextual burden. Additionally, for many real-world applications, large or heterogeneous tool outputs, such as web-search results (Wei et al., 2025e) and database-query results (Jing et al., 2025), can also accumulate dramatically. Naively removing or compressing these traces risks breaking causal dependencies between reasoning and action steps and undermining subsequent decisions (Yao et al., 2023). Effective memory must instead preserve task-critical states, unresolved dependencies, prior tool effects, and execution failures while compressing redundant observations and reasoning traces.

Thus, while multi-turn tool use substantially enhances agent capability, it also exposes scalability limits on finite context windows, underscoring a core challenge for long-horizon agent design (Liu et al., 2024b). Beyond maintaining the state of an ongoing task, agents should also consolidate repeated tool-use experience into reusable procedures and skills, allowing future tasks to benefit from prior execution rather than reproducing the same long reasoning traces.

### 6.2.2 Evolving with Environmental Complexity

As environments scale in complexity, an agent's context must accommodate heterogeneous data modalities, asynchronous tool interactions, and protocol- or permission-constrained external state. In such settings, context can no longer be treated as a linear interaction trace appended to a prompt. Real-world deployments require reasoning over structured artifacts such as API responses, files, databases, logs, and configuration states, whose semantics depend on schemas, provenance, update rules, and temporal validity. Naively flattening these artifacts into token sequences is both inefficient and structurally lossy, undermining interpretability, precise retrieval, and targeted updates (Modarressi et al., 2024). Consequently, increasing environmental complexity shifts memory from an implicit byproduct of prompt accumulation to an explicit, system-level component responsible for structured context management. The memory challenge thus transitions from merely retaining past information to maintaining coherent, queryable, and updatable representations of environmental state across diverse sources and lifetimes. For self-evolving agents, these representations must further adapt as schemas, interfaces, permissions, and environmental dynamics change, while preserving the validity and provenance of previously acquired knowledge and skills.

Recent agent systems address this challenge by externalizing memory beyond the prompt and exposing explicit read–write interfaces that decouple storage from reasoning (cauri, 2025). Externalized memory enables schema-aware retrieval, versioning, targeted edits, and access control—operations that are difficult or infeasible within prompt-based context alone (Yakobi & Sadon, 2025). Protocol-based interfaces such as the Model Context Protocol (Anthropic, 2024) standardize how agents access external tools and contextual resources, while skill-oriented abstractions such as Claude Agent Skills (Anthropic, 2025) package procedural instructions, executable resources, and domain-specific workflows into reusable units. Together, these abstractions separate persistent state, environment access, and reusable skills from the transient reasoning context, thereby improving modularity, auditability, and behavioral safety (Keen, 2025).

As environmental complexity increases, persistence becomes unavoidable: user models, system configurations, and task artifacts represent durable state that must remain consistent across sessions while respecting governance constraints such as privacy, permissions, and rollback (Sarin et al., 2025; Wu & Shu, 2025). These demands introduce additional challenges, including schema drift, concurrent updates, and path-dependent evaluation, underscoring the need for principled abstractions that treat memory as a first-class infrastructure for context management in complex, open-world environments. Memory must therefore evolve

at both the content and structural levels: it should update environmental facts and task state, reorganize representations when schemas change, and revise previously learned skills when their underlying interfaces or action preconditions are no longer valid.

### 6.2.3 Evolving across Multiple Environments and Agents

**Memory as an Interface across Tool Environments.** In open-world settings, context complexity arises not only from prolonged interaction within a single environment, but also from the need to operate across multiple heterogeneous tool environments, each defined by distinct state representations, action spaces, and access protocols. For example, a personal embodied assistant may alternate between a physical household environment, where it performs long-horizon behaviors grounded in perceptual feedback, and a digital web environment, where it retrieves information such as weather forecasts through browser-based interactions. Supporting such behavior requires memory mechanisms that can preserve, differentiate, and reconcile environment-specific states across diverse action and observation modalities (Glocker et al., 2025; Hong et al., 2025). In OSWorld (Xie et al., 2024), a GUI agent may issue search queries in a browser, retrieve results from a news application, and consult documents in a file viewer, with each interaction producing environment-specific observations and state transitions. While tools enable localized interaction within each environment, memory provides the interface function that persists, organizes, and contextualizes information across environment boundaries. As a result, memory systems are required not merely to log past tool calls, but to maintain structured and separated representations of environment state that support long-horizon reasoning without exceeding the context window (Burtsev et al., 2020; Rae et al., 2019).

**Memory in Multi-Human-Agent Systems.** As agent systems scale to include multiple agents and human participants, effective coordination increasingly depends on structured communication and shared memory mechanisms rather than naïve context sharing, which quickly fragments under limited context windows (Chen et al., 2025f). Recent approaches introduce agent-aligned or semi-shared memory abstractions that encode relational histories, inter-agent dependencies, and task-relevant state across long interaction horizons. For example, Intrinsic Memory Agents equip agents with role-specific memory templates that preserve specialized perspectives while enabling integration into a shared contextual substrate, substantially improving long-horizon planning stability (Yuen et al., 2025). Beyond memory, structured communication protocols play a central role in coordination: hierarchical and role-aware dialogue schemes reduce noise and bias in inter-agent exchanges (Wang et al., 2025s), while cognitively adaptive orchestration frameworks dynamically adjust communication patterns based on inferred collaborator states (Zhang et al., 2025h). In open-world deployments involving multiple humans and agents, coordination further extends to alignment, conflict resolution, and collective decision-making under disagreement. Empirical studies show that debate-based protocols and decision rules such as consensus or majority voting significantly influence performance across reasoning and knowledge tasks (Kaesberg et al., 2025; Samanta et al., 2025), including multimodal extraction settings where a few debate rounds measurably improve accuracy (Huang & Caragea, 2026), while adaptive and consensus-free debate mechanisms balance computational cost, robustness, and conformity effects (Fan et al., 2025a; Cui et al., 2025b). As these systems scale, explicit graph-structured representations increasingly underpin orchestration and communication, organizing relational dependencies into optimizable communication topologies and reframing context management as a problem of distributed memory, coordination, and alignment in open-world systems (Qian et al., 2025a; Zhang et al., 2025f;e).

## 7 Evaluation

Evaluating foundation agent memory is fundamentally about measuring whether stored information and experience are accurate, useful, reliable, and efficiently accessible under long-horizon interactions. In the following, we summarize commonly used metrics in three different basic categories, including *accuracy-based*, *similarity-based*, and *LLM-as-a-judge*, in Section 7.1. In addition, we collect the commonly used benchmarks to assess the foundation agent's performance, as shown in Section 7.2.

Table 3: **Metrics used in Foundation Agent Memory Evaluations**.

| Metric | Short Description | Representative Benchmark(s) |
|---|---|---|
| **Accuracy-Based Metrics** | | |
| Accuracy / Memory Accuracy | Proportion of instances answered correctly, computed at the benchmark granularity (e.g., question-, turn-, session-, or task-level). | HotpotQA (Yang et al., 2018), PerLTQA (Du et al., 2024), MemoryBank (Zhong et al., 2024) |
| F1 Score | Harmonic mean of precision and recall, computed at token level and optionally aggregated to question, turn, or session level. | MuSiQue (Trivedi et al., 2022), MSC (Xu et al., 2022a), PerLTQA (Du et al., 2024) |
| Recall@K | Evaluates retrieval success by checking if relevant evidence appears within the top-$K$ results. | LongMemEval (Wu et al., 2025b), PerLTQA (Du et al., 2024) |
| Mean Average Precision (MAP) | Averages precision at ranks of relevant items, then averages across queries. | PerLTQA (Du et al., 2024), PMR (Kohar & Krishnan, 2025) |
| NDCG@K | Measures ranked relevance at cutoff $K$ by prioritizing top positions. | LongMemEval (Wu et al., 2025b), PMR (Kohar & Krishnan, 2025) |
| Success Rate (SR) / Goal Completion (GC) | Fraction of interactive tasks completed under environment-defined success checkers, typically measured at the task or episode level. | WebArena (Zhou et al., 2024), OSWorld (Xie et al., 2024), MineDojo (Fan et al., 2022) |
| Pass@K / Resolved Rate (RR) | Probability that at least one correct solution appears among $K$ sampled attempts (Pass@K), or issues resolved end-to-end (RR). | HumanEval (Chen et al., 2021), SWE-Bench (Jimenez et al., 2024) |
| Memory Integrity (MI) | Completeness of memory extraction, measured by coverage or recall over memory points. | HaluMem (Chen et al., 2025a) |
| False Memory Rate (FMR) | Rate of introducing hallucinated memories, including fabricated or incorrect updates. | HaluMem (Chen et al., 2025a) |
| **Similarity-Based Metrics** | | |
| ROUGE | Measures $n$-gram and longest-common-subsequence overlap between generated and reference text. It is widely used for summarization. | LoCoMo (Maharana et al., 2024), MemoryBench (Ai et al., 2025) |
| BLEU | $n$-gram precision overlap with reference. It is commonly used for dialogue generation. | DuLeMon (Xu et al., 2022b), LoCoMo (Maharana et al., 2024) |
| Distinct-$n$ | Calculates the ratio of unique $n$-grams to measure lexical diversity and discourage repetition. | DuLeMon (Xu et al., 2022b), MADial-Bench (He et al., 2025a) |
| BERTScore | Embedding-based semantic similarity between candidate and reference. | LoCoMo (Maharana et al., 2024), MADial-Bench (He et al., 2025a) |
| FactScore | Fact-level faithfulness: extracts atomic claims and measures the fraction supported by retrieved evidence. | LoCoMo (Maharana et al., 2024), Face4Rag (Xu et al., 2024b) |
| Perplexity | Likelihood-based metric of predicting reference text, typically token-level and aggregated over sessions. | MSC (Xu et al., 2022a), DuLeMon (Xu et al., 2022b) |
| **LLM-as-a-Judge Metrics (JUDGE)** | | |
| Response Correctness | A strong LLM judges whether the response answers the query or satisfies constraints. | LongMemEval (Wu et al., 2025b), MemTrack (Deshpande et al., 2025) |
| Faithfulness / Groundedness | Judge checks that response claims are grounded in retrieved context or memory, or checks whether the citations support the claims. | SeekBench (Shao et al., 2025), LiveResearchBench (Wang et al., 2025f) |
| Preference Following | Judge evaluates preference following by checking whether the output satisfies user-stated preferences or constraints. | PrefEval (Zhao et al., 2025c), BEAM (Tavakoli et al., 2025), ConvoMem (Pakhomov et al., 2025) |

### 7.1 Metrics

Table 3 summarizes metrics most commonly used in foundation agent memory evaluations. There are multiple dimensions to evaluate foundation agents and their memory module performance. Some tasks have clear ground truth answers and can be scored with exact correctness (Du et al., 2024; Dunn et al., 2017), while others are open-ended (dialogue (Budzianowski et al., 2018), summarization (Maharana et al., 2024), preference following (Zhao et al., 2025c)) where multiple outputs are acceptable and reference-based scoring is not reliable. As a result, existing work typically combines outcome-level correctness with retrieval-oriented assessment and judge-based rubrics, depending on whether the benchmark exposes a memory module, uses long-context prompting, or evaluates agents acting in a specific environment, scenario, or task (Patlan et al., 2025; Yadav et al., 2025).

**Accuracy-based Metrics.** When tasks have a clear objective outcome, accuracy-based metrics are used. For answering questions about long histories, accuracy or memory accuracy directly assesses the alignment of the final response with the ground truth answer (Yang et al., 2018; Zhong et al., 2024; Du et al., 2024). F1 relaxes the exact-match criterion by giving credit for partial overlap at the token level. This is especially prevalent when responses do not match exactly but have some variance (Trivedi et al., 2022; Deng et al., 2023). When the benchmark evaluates an explicit memory module, Recall@K, Mean Average Precision (MAP), and Normalized Discounted Cumulative Gain (NDCG)@K become central because they separate "did the system retrieve the right evidence" from "did the generator phrase the answer well." Recall@K measures the fraction of relevant items that appear among the top-$K$ retrieved results (Wu et al., 2025b). MAP and NDCG@K, on the other hand, also take into account the quality of the ranking and give systems that put relevant memories first a higher score (Kohar & Krishnan, 2025). For interactive agents, correctness is instead defined by environment evaluators, so Success Rate (SR) or Goal Completion (GC) represents whether the agent finishes tasks end-to-end (Zhou et al., 2024; Zheng et al., 2025a). For embodied agents, additional metrics such as path-length weighted success rate (SPL), navigation efficiency, and interaction success rate are commonly used to capture execution quality beyond binary completion. For code and tool-use settings, Pass@K and Resolved Rate (RR) measure whether at least one of $K$ sampled attempts solves the task or resolves an issue (Yao et al., 2025; Jimenez et al., 2024). In addition, memory-centric benchmarks increasingly add failure-mode assessments that are hard to capture with end-task accuracy alone. Memory Integrity quantifies whether extracted memories cover the required memory points, and False Memory Rate measures how often systems introduce fabricated or incorrect memories during storage, update, or use (Chen et al., 2025a).

**Similarity-based Metrics.** Similarity-based metrics are most prevalent in dialogue generation and summarization, where the output is free-form, and correctness cannot be fully checked by the accuracy or F1 score (Chen et al., 2024d). BLEU, ROUGE, and ROUGE-L measure lexical overlap with a reference, which is useful for tracking surface similarity but can underestimate valid paraphrases and overestimate fluent yet ungrounded responses (Gehrmann et al., 2023; Ai et al., 2025). Distinct-$n$ complements overlap metrics by measuring lexical diversity, discouraging repetitive generations that can inflate similarity scores without improving faithfulness (He et al., 2025a). When lexical overlap is too strict, BERTScore provides an embedding-based approximation of semantic similarity (Maharana et al., 2024), and FactScore evaluates memory faithfulness by checking agreement at the level of atomic factual claims (Min et al., 2023), which is particularly relevant when summarization is used as a compression mechanism for long histories (Saxena et al., 2025). Perplexity is also used in some long dialogue benchmarks as a likelihood-based metric for generation quality over sessions (Xu et al., 2022a), but it remains an indirect indicator for memory performance, because it does not verify whether the model's content is grounded in the correct historical evidence (Durmus et al., 2022).

**LLM-as-a-judge Metrics.** LLM-as-a-judge metrics are used when ground truth answers, or references, are incomplete, when multiple responses are acceptable, or when evaluation requires a rubric that is hard to encode as string matching (Yu et al., 2025a). Response Correctness asks a strong model to decide whether the answer satisfies the user query (Deshpande et al., 2025), which is convenient for open-ended responses but introduces judge dependence and sensitivity to prompting. Faithfulness or Groundedness uses a judge to verify that claims are supported by retrieved context or memory (and in some settings, whether provided citations support the response) (Shao et al., 2025; Wang et al., 2025f), which helps distinguish helpful but

hallucinated answers from evidence-supported ones. Preference Following uses a judge to determine whether the output respects explicit user constraints or stated preferences (Zhao et al., 2025c; Tavakoli et al., 2025; Pakhomov et al., 2025), which is essential for personalization benchmarks where correctness is defined by user alignment rather than a single factual label (Wang et al., 2024b).

Across these metrics, a practical implication is that evaluation becomes more attributable to memory mechanisms when the benchmark can separate retrieval or selection from generation (Chen et al., 2025a). Retrieval and ranking metrics (Recall@K/MAP/NDCG@K) assess whether the memory interface surfaces the right items, while integrity and hallucination-oriented metrics (MI/FMR) expose failure modes that may not change end-task accuracy until they accumulate (Zhang et al., 2025a). In contrast, similarity-based metrics remain useful for tracking fluency and summarization quality, but they should be paired with grounding checks (FactScore or judge-based faithfulness) to avoid rewarding ungrounded paraphrases (Aralikatte et al., 2021). In addition, some benchmarks also emphasize cost and feasibility (e.g., capacity and efficiency) (Tan et al., 2025a), motivating reporting not only what the agent remembers, but also the computational and storage trade-offs required to achieve that performance.

## 7.2 Benchmarks

To assess the memory improvement on the foundation agent task in diverse scenarios, we categorize existing benchmarks into two primary domains: **user-centric** benchmarks and **agent-centric** benchmarks. User-centric benchmarks, such as MSC (Xu et al., 2022a) and MemoryBank (Zhong et al., 2024), primarily evaluate conversational consistency, measuring an agent's ability to retain persona information, recall user preferences, and sustain coherent interactions across multi-session or multi-turn dialogue. On the other hand, agent-centric benchmarks, including OSWorld (Xie et al., 2024) and WebArena (Zhou et al., 2024), focus on the functional application of memory for complex problem-solving, measuring success rates in tasks that require multi-hop reasoning and tool usage. We present the commonly used user-centric and agent-centric benchmarks in Section 7.2.1 and Section 7.2.2, respectively.

### 7.2.1 User-Centric Evaluation Benchmarks

User-centric benchmarks evaluate a foundation agent in personalized dialogue, where the goal is to remain consistent with a specific user's evolving profile and the shared interaction history over long horizons. Compared to agent-centric tasks, whose evaluation metrics are largely user-invariant (Mo et al., 2025), user-centric settings are inherently user-dependent (Zhao et al., 2025d). The agent must decide what to store from the conversation, retrieve relevant information when needed, revise it when the user changes their preference, and stay calibrated when evidence is missing (Terranova et al., 2025). Table 4 summarizes representative benchmarks with the interaction scale (#sessions, #questions, maximum context length), data resource (REAL/SIM/MIX), and memory ability and evaluation coverage using ✓/✓/✗.

We define ten user-centric memory abilities to characterize what an agent must remember and use over long-horizon interactions (Wu et al., 2025b; Pakhomov et al., 2025): Fact Extraction (FE) is the ability to identify and store reusable facts from dialogue (e.g., user attributes, constraints, key events) so they can be recalled later; Multi-Session Reasoning (MR) requires integrating evidence that is distributed across multiple sessions and turns rather than contained in a single session; Temporal Reasoning (TR) covers reasoning over time series (ordering, timestamps, recency) and selecting the correct state when information changes; Update & Refresh (UR) captures explicitly revising memory when new evidence contradicts old content (overwriting outdated facts and following the latest state under conflicts); Compression & Summarization (CS) is the ability to condense long interaction histories into compact memory representations that remain faithful and usable; Forgetting & Retention (FR) reflects maintaining long-range information while selectively forgetting obsolete or irrelevant content to reduce interference; User Facts & Preferences (UP) focuses on user-centric memory subjects (persona, preferences, relationships, recurring habits or events) and their evolution; Assistant Facts (AS) tracks the assistant's own prior statements, recommendations, or commitments so the agent maintains coherence with what it previously said; Implicit Inference and Connection (IC) measures whether the agent can link scattered clues and perform multi-hop or implicit inference (e.g., applying a previously mentioned limitation to a new recommendation without being reminded); and Abstain & Boundary Handling (AB)

Table 4: **Overview of user-centric memory benchmarks. Memory Abilities**— FE: Fact Extraction; MR: Multi-Session Reasoning; TR: Temporal Reasoning; UR: Update & Refresh; CS: Compression & Summarization; FR: Forgetting & Retention; UP: User Facts & Preferences; AS: Assistant Facts; IC: Implicit Inference & Connection; AB: Abstain & Boundary Handling. **Marks**— ✓: *explicitly covered.* The benchmark defines this ability as a target with dedicated task types or annotations and evaluates it directly; ✓: *partially or indirectly covered.* The ability may be required in some instances or implied by the setup, but lacks dedicated labeling or ability specific evaluation, so coverage is weak or not cleanly attributable; ✗: *not covered.* No corresponding task, annotation, or rubric exists, and the evaluation does not require or measure this ability. **Resource**— REAL (human-authored or real-world conversations), SIM (fully synthetic or simulated), MIX (mixture of real and synthetic). **Evaluation metrics**— AR: Accurate Retrieval; TTL: Test-Time Learning; LRU: Long-Range Understanding; SF: Selective Forgetting; MM-R: Multi-Message Relevance; RC: Response Correctness; CC: Contextual Coherence; MA: Memory Accuracy; MI: Memory Integrity; FMR: False Memory Rate; MRS: Memory Retention Score; JUDGE: LLM-as-a-judge.

| Name | #Sess. | #Q | Max Tok. | FE | MR | TR | UR | CS | FR | UP | AS | IC | AB | Res. | Link | Evaluation |
|---|---|---|---|---|---|---|---|---|---|---|---|---|---|---|---|---|
| MSC | 5K | – | 1K | ✓ | ✓ | ✓ | ✓ | ✓ | ✗ | ✓ | ✓ | ✓ | ✗ | REAL | ⦿ | Perplexity |
| DuLeMon | 27,501 | – | 1K | ✓ | ✓ | ✗ | ✓ | ✓ | ✗ | ✓ | ✓ | ✓ | ✗ | REAL | ⦿ | F1, Recall@K, BLEU, Distinct-$n$ |
| MemoryBank | 300 | 194 | 5K | ✓ | ✓ | ✓ | ✓ | ✓ | ✓ | ✓ | ✓ | ✓ | ✗ | SIM | ⦿ | Retrieval Accuracy, RC, CC |
| PerLTQA | 3,409 | 8,593 | 1M | ✗ | ✓ | ✗ | ✗ | ✗ | ✗ | ✓ | ✓ | ✓ | ✗ | SIM | ⦿ | Accuracy, F1, Recall@K, MAP |
| LoCoMo | 1K | 7,512 | 10K | ✓ | ✓ | ✓ | ✗ | ✓ | ✓ | ✓ | ✓ | ✓ | ✓ | MIX | ⦿ | F1, ROUGE, BLEU, MM-R |
| DialSim | ∼1,300 | 1M | 367K | ✗ | ✓ | ✓ | ✓ | ✓ | ✓ | ✓ | ✓ | ✓ | ✓ | MIX | ⦿ | Accuracy |
| LOCCO | 3,080 | 2,981 | – | ✓ | ✗ | ✓ | ✗ | ✗ | ✓ | ✓ | ✗ | ✗ | ✗ | SIM | ⦿ | Accuracy, MRS |
| MemoryAgentBench | 130 | 207 | 1.44M | ✓ | ✓ | ✓ | ✓ | ✓ | ✓ | ✓ | ✗ | ✓ | ✓ | SIM | ⦿ | Accuracy, F1, AR, TTL, LRU, SF |
| LongMemEval | 50K | 500 | 1.5M | ✓ | ✓ | ✓ | ✓ | ✓ | ✓ | ✓ | ✓ | ✓ | ✓ | MIX | 🔗 | JUDGE, Recall@K, NDCG@K |
| HaluMem | 1,387 | 3,467 | 1M | ✓ | ✓ | ✓ | ✓ | ✗ | ✓ | ✓ | ✓ | ✓ | ✓ | MIX | ⦿ | MA, MI, FMR |
| PersonaMem | 60 | ∼6K | 1M | ✓ | ✓ | ✓ | ✗ | ✓ | ✓ | ✓ | ✗ | ✓ | ✗ | SIM | ⦿ | Accuracy |
| PrefEval | – | 3,000 | 100K | ✓ | ✓ | ✓ | ✓ | ✗ | ✓ | ✓ | ✗ | ✓ | ✓ | MIX | 🔗 | JUDGE, Accuracy |
| MemBench | 65K | 53K | 100K | ✓ | ✓ | ✓ | ✓ | ✓ | ✓ | ✓ | ✓ | ✓ | ✗ | MIX | ⦿ | Accuracy, Capacity, Efficiency |
| MemoryBench | 20K | – | – | ✓ | ✓ | ✓ | ✓ | ✓ | ✓ | ✓ | ✗ | ✓ | ✓ | MIX | ⦿ | Accuracy, F1, JUDGE |
| ConvoMem | 300 | 75,335 | 3M | ✓ | ✓ | ✓ | ✓ | ✗ | ✓ | ✓ | ✓ | ✓ | ✓ | SIM | ⦿ | Accuracy |

measures recognizing unknown or unanswerable cases, conflicts, or false premises and avoiding fabrication, for example by explicitly saying "I don't know" when the needed information was never stated.

A key pattern in Table 4 is that MR and UP are the most consistently covered abilities, reflecting the dominant framing of user memory as retrieving user-related facts from long dialogue and using them later. In contrast, operational abilities are much less uniformly defined. Early long dialogue benchmarks such as MSC (Xu et al., 2022a) and DuLeMon (Xu et al., 2022b) emphasize multi-session coherence and persona usage, but their evaluation relies heavily on similarity-based or generation-based metrics (e.g., BLEU, Distinct-$n$, Perplexity, ROUGE), which weakly isolates memory. Fluency and generic helpfulness can mask incorrect recall, making FE, UR, and FR difficult to attribute. Newer benchmarks increasingly move toward mechanism attributable evaluation by defining specific task types or assessments. MemoryBank (Zhong et al., 2024) and HaluMem (Chen et al., 2025a) introduce explicit memory records and operation-level searching, turning extraction, updating, and memory integrity failures into measurable targets. LongMemEval (Wu et al., 2025b) further strengthens attribution by explicitly categorizing question types according to memory abilities, enabling cleaner localization of failures to temporal reasoning, updating, or abstention behaviors. However, CS and FR remain comparatively under-evaluated and less systematically assessed in current benchmark designs. Compression is explicit in only a few benchmarks (Hu et al., 2025c), and selective forgetting or retention (Jia et al., 2025b) is frequently partial or absent, despite being essential for long-horizon assistants operating under finite memory budgets and evolving user states. AB is also inconsistently required (Ai et al., 2025; Jiang et al., 2025a). Only a few benchmarks explicitly reward abstention under missing evidence, leaving a gap for evaluating safe memory behavior that prevents confident hallucinations.

Evaluation in Table 4 follows several main aspects that reflect how much evaluation the benchmark provides. First, many benchmarks reduce evaluation to answer-level correctness on ground truth questions, reporting

accuracy and F1 (e.g., PerLTQA (Du et al., 2024), LoCoMo (Maharana et al., 2024), DialSim (Kim et al., 2024a), PersonaMem (Jiang et al., 2025a), MemoryAgentBench (Hu et al., 2025c)), and in some cases, using retrieval-style scoring such as Recall@K or ranked relevance metrics (MAP/NDCG@K) when the task is explicitly framed as selecting supporting memories rather than generating free-form text (e.g., PerLTQA (Du et al., 2024), LongMemEval (Wu et al., 2025b)). Second, several user-centric benchmarks introduce memory-specific assessment that goes beyond final answers and directly evaluates memory system behavior. HaluMem reports memory accuracy and integrity as well as false memory rate to quantify hallucinated or incorrect memory operations (Chen et al., 2025a), LOCCO (Jia et al., 2025b) reports memory retention (MRS), and MemBench (Tan et al., 2025a) explicitly measures capacity and efficiency to capture performance and cost trade-offs under fixed memory budgets. Third, when outputs are open-ended or reference answers are insufficient, benchmarks increasingly rely on LLM-as-a-judge to score response correctness or preference adherence (e.g., LongMemEval (Wu et al., 2025b), PrefEval (Zhao et al., 2025c), and MemoryBench (Ai et al., 2025)). Compared to exact-match scoring, judge-based evaluation expands coverage to realistic assistant behavior, but it is more sensitive to the evaluator model and rubric.

### 7.2.2 Agent-Centric Evaluation Benchmarks

Agent-centric benchmarks evaluate a foundation model as a *semi-autonomous agent* that must execute multi-step actions in an environment to reach a clearly specified goal, rather than only producing a response to a static prompt. These benchmarks typically provide a task specification (e.g., an initial state with constraints) together with an objective success checker, and summarize performance with end-to-end metrics such as Accuracy/F1 for text QA tasks or Success Rate (SR) for interactive environments. Because correctness is defined by the environment's state, the target outcome is largely **user-invariant**: under the same task setting, different users issuing the same request should obtain the same completion signal, even if the agent follows different trajectories (Trivedi et al., 2024; Zhou et al., 2024).

We summarize commonly used agent-centric benchmarks in Table 5. Beyond tagging each benchmark by task environment (`Env`) and interface (`Interact`), we further annotate (i) **resource type**, indicating whether the task world and evidence are constructed with real-world or simulated data, or a hybrid of both, and (ii) **core agent abilities** (`Abilities`) most directly exercised by the benchmark. We define a set of *core ability tags* commonly used to characterize agent and memory benchmarks: `TEMP` (temporal/sequence reasoning over event order and time dependencies), `STATE` (tracking and updating environment/task state across multi-step interaction), `GROUND` (grounding natural-language instructions into concrete environment targets/actions), `PLAN` (planning and re-planning multi-step actions toward a goal), `TOOL` (selecting and correctly invoking tools/APIs to solve subtasks), `MHOP` (multi-hop reasoning that composes multiple pieces of evidence), `DIAL` (goal-directed dialogue management such as clarification and consistency across turns), and `DEBUG`, `CODEGEN`, and `PATCH` (diagnosing failures, generating new code, and repairing existing code in software-engineering environments); optionally, `TTL` denotes test-time learning, where agents improve later performance by accumulating experience in memory *without* parameter updates. In these works, **memory is treated as a tool and module for cross-session knowledge transfer and experience sharing**, enabling agents to preserve intermediate findings, tool outputs, and state updates across long horizons and limited context windows to improve downstream task completion ability.

Across environments, agent-centric evaluation covers `TEXT/WEB` information seeking (Yang et al., 2018; Trivedi et al., 2022; Deng et al., 2023; Yao et al., 2022; Mialon et al., 2024; Wei et al., 2025a), `OS/APP` computer use (Xie et al., 2024; Trivedi et al., 2024; Yao et al., 2025; Barres et al., 2025), `CODE` software engineering (Chen et al., 2021; Qiu et al., 2025b), embodied `ROBOT/GAME` control (Shridhar et al., 2020; 2021; Fan et al., 2022), and long-form `VIDEO/PAPER` workflows (Mangalam et al., 2023; Fu et al., 2025; Wu et al., 2024a; Starace et al., 2025). These environments induce distinct memory pressures. Text and multi-hop QA emphasize evidence tracking and state bookkeeping over intermediate facts (Ho et al., 2020; Trivedi et al., 2022). Web, desktop, and app settings stress episodic action memory. Foundation agents must remember visited pages, filled fields, downloaded files, prior tool outputs, constraints discovered during interaction and avoid redundant exploration (Deng et al., 2023; Zhou et al., 2024; Yao et al., 2022). Code and paper workflows require working memory, such as files, patches, hypotheses, and experiment logs, to support cross-stage continuity (Starace et al., 2025; Miao et al., 2025). These requirements make compression, selection, and retrieval

Table 5: Overview of agent-centric task benchmarks (§7.2.2). Each benchmark is annotated by task environment (**Env**), interaction mode (**Interact**), data source (**Resource**), core abilities (**Abilities**), and evaluation metrics (**Evaluation**). **Env**—TEXT: Text/Document; WEB: Web; OS: Operating System/Desktop; APP: Application/API-centric; CODE: Code/Software Engineering; ROBOT: Embodied Robotics; GAME: Game/Simulation; VIDEO: Video (long-form); PAPER: Scientific Paper/Research. **Interact**—QA: Question Answering; MT: Multi-turn; GUI: Graphical UI; API: Tool/API Invocation; EXEC: Execution-based; MM: Multimodal; ACT: Action/Control. **Resource**—REAL: real-world; SIM: simulated/synthetic; MIX: mixed real+sim. **Abilities**—MHOP: Multi-hop Reasoning; PLAN: Planning/Acting; STATE: State Tracking; GROUND: Grounding; TOOL: Tool Use; DEBUG: Debugging; CODEGEN: Code Generation; PATCH: Patch/Repair; TEMP: Temporal Reasoning; DIAL: Dialogue Management; TTL: Test-Time Learning. **Evaluation**—SR: Success Rate / Solve Rate; PR: Pass Rate; GC: Goal Completion; RR: Resolved Rate; LCCS: Longest Consecutive Correct Sequence.

| Name | #Data | Env | Interact | Resource | Core Abilities | Link | Evaluation |
|---|---|---|---|---|---|---|---|
| HotpotQA | 113K | TEXT | QA | REAL | MHOP, STATE | Github | Accuracy, F1 |
| 2WikiMultiHopQA | 193K | TEXT | QA | REAL | MHOP, STATE | Github | Accuracy, F1 |
| MuSiQue | 25K | TEXT | QA | REAL | MHOP, STATE | Github | F1 |
| HLE | 2.5K | TEXT | QA | REAL | MHOP, STATE | Website | Accuracy, RMSE |
| BrowseComp | 1,266 | WEB | QA, GUI | REAL | PLAN, TOOL, MHOP, STATE | Website | Accuracy, PR |
| Mind2Web | 2.35K | WEB | GUI | REAL | GROUND, PLAN, STATE | Website | Accuracy, F1, SR |
| WebArena | 812 | WEB | GUI | SIM | GROUND, PLAN, STATE | Website | SR |
| WebShop | 12.1K | WEB | GUI | MIX | GROUND, PLAN, STATE | Github | Task Score, SR |
| GAIA | 466 | WEB | QA, GUI | REAL | TOOL, MHOP, PLAN, STATE | Website | Accuracy, SR |
| OSWorld | 369 | OS | GUI, MM | REAL | GROUND, PLAN, STATE | Website | SR |
| AppWorld | 750 | APP | API, MT | SIM | TOOL, CODEGEN, PLAN, STATE | Website | SR |
| $\tau$-Bench | 165 | APP | API, MT | SIM | TOOL, PLAN, STATE, DIAL | Github | Pass^1, Pass^k |
| $\tau$-Bench2 | 2.3K | APP | API, MT | SIM | TOOL, PLAN, STATE, DIAL | Github | Pass^1, Pass^k |
| HumanEval | 164 | CODE | EXEC | REAL | CODEGEN, DEBUG | Github | Pass@1 |
| SWE-Bench | 2.3K | CODE | EXEC | REAL | PATCH, DEBUG, STATE | Website | RR |
| LoCoBench | 8K | CODE | EXEC | SIM | CODEGEN, PATCH, DEBUG, STATE | Github | Multi-metric |
| LoCoBench-Agent | 8K | CODE | EXEC | SIM | TOOL, PLAN, CODEGEN, PATCH, DEBUG, STATE | Github | Multi-metric |
| PaperBench | 20 | PAPER | EXEC, MT | REAL | TOOL, PLAN, CODEGEN, DEBUG, STATE | Website | Replication Score |
| RECODE-H | 102 | PAPER | EXEC, MT | REAL | CODEGEN, PATCH, DEBUG, DIAL, STATE | Github | Recall, PR |
| ALFRED | 25.7K | ROBOT | ACT, MM | SIM | GROUND, PLAN, STATE | Website | SR, GC |
| ALFWorld | 3.8K | ROBOT | ACT, MT | SIM | PLAN, STATE, GROUND | Website | SR |
| MineDojo | 3.1K | GAME | ACT, MM | MIX | PLAN, STATE, GROUND | Website | SR |
| EgoSchema | 5,031 | VIDEO | QA, MM | REAL | TEMP | Website | Accuracy |
| Video-MME | 2,700 | VIDEO | QA, MM | REAL | TEMP | Website | Accuracy |
| LongVideoBench | 6.7K | VIDEO | QA, MM | REAL | TEMP | Website | Accuracy |
| MT-Mind2Web | 720 | WEB | GUI, MT | REAL | GROUND, PLAN, STATE, DIAL | Github | Accuracy, F1, SR |
| MPR | 10.8K | TEXT | QA | SIM | MHOP, STATE | Github | Accuracy |
| StoryBench | 397 | GAME | MT, ACT | SIM | PLAN, STATE, TEMP | – | Accuracy, LCCS |
| Evo-Memory | ∼3,700 | TEXT | QA, MT | MIX | TTL, PLAN, STATE | – | Accuracy, SR |
| LifelongAgentBench | 1,396 | APP, OS | API, MT | SIM | TTL, TOOL, PLAN, STATE | Website | SR |
| OdysseyBench | 602 | APP | GUI, MT | MIX | PLAN, TOOL, STATE, TEMP | Github | PR |

fidelity critical to complete tasks (Jimenez et al., 2024; Qiu et al., 2025a). Video benchmarks additionally test sensory memory over long clips, where critical cues may appear far before the question is asked (Fu et al., 2025; Wu et al., 2024a). As benchmarks become more realistic and long-horizon, foundation agents cannot retain all task relevant context in the prompt, and must instead externalize state via explicit memory operations to preserve and retrieve critical information over time.

Evaluation methods for benchmarks in Table 5 mainly fall into a few categories. Answer-level *Accuracy/F1* for `TEXT` (Trivedi et al., 2022; Phan et al., 2025), goal-based `SR, GC` for interactive `WEB/OS/APP/ROBOT/GAME` (Xie et al., 2024; Trivedi et al., 2024; Shridhar et al., 2020; 2021; Fan et al., 2022), and execution-centric metrics (e.g., `Pass@1, RR`) for `CODE` (Jimenez et al., 2024; Qiu et al., 2025a). For memory-centric analysis, two additional dimensions are crucial: (1) **dependency distance**—how far apart the required information and its later use occur, such as within-turn, cross-turn, or cross-session, and (2) **memory correctness under interaction**—whether stored items remain faithful, non-contradictory, and policy-consistent as the environment evolves. This motivates complementing end-to-end success with memory-sensitive measurements such as: (i) retrieval faithfulness and coverage for required facts or tool outputs, (ii) error modes in state tracking (drift, omission, contradiction), (iii) persistence under interruptions (resume after long gaps), and (iv) efficiency trade-offs (memory size, update frequency, and retrieval cost). We expect next-generation benchmarks to go beyond a single end-to-end accuracy or success rate and instead incorporate memory-related metrics, so that evaluation can also be attributed to the memory mechanism rather than to short-horizon prompting or incidental heuristics.

## 8 Applications

Memory transforms LLMs into dynamic, persistent agents, representing a fundamental shift in recent research. When implemented in complex real-world scenarios, agentic memory has emerged not merely as a storage utility, but as the cognitive substrate that enables continuity, learning, and personalization, bridging an agent's past experiences with its future actions. Recent work has broadly investigated memory-enabled capabilities in LLM agents where the ways of storing, operating, and managing memory vary significantly. To provide insights into how improvements in memory design boost further abilities, this section discusses and summarizes recent representative works across education, scientific research, gaming and simulation, robotics, healthcare, dialogue systems, workflow automation, software engineering, online streaming and recommendation, information search, finance and accounting, and legal and consulting. The application domains are illustrated in Figure 8, and the summarization is shown in Table 6.

**Education.** Educational agents require sustained, personalized interactions spanning a long period of time, making memory essential for tracking learner progress, adapting instruction, and maintaining pedagogical coherence (Chu et al., 2025). Without memory, agents treat every interaction as an isolated event, unable to build on a student's prior knowledge or maintain pedagogical consistency. Recent models illustrate this shift toward more sophisticated memory modules. For instance, LOOM (Cui et al., 2025a) utilizes a learner memory graph mapping educational concepts with prerequisite dependencies to facilitate personalized curriculum generation. Agent4Edu (Gao et al., 2025b) explicitly replicates the Ebbinghaus Forgetting Curve to simulate knowledge decay for teacher training. WebCoach (Liu et al., 2025b) uses persistent cross-session memory to enable self-evolving instructional guidance. These systems reveal that in the educational domain, memory functions less as a historical log and more as a cognitive digital twin (Zheng et al., 2022) of the student. Future works should move toward interoperable memory protocols that allow a learner's cognitive profile to persist across different educational platforms, effectively creating an evolving record of their intellectual development.

**Scientific Research.** Scientific research represents a frontier where the process has been lengthy and costly, requiring agents to synthesize vast literature, manage provenance, and maintain reasoning continuity across multi-stage endeavors. In recent studies, General Agentic Memory (GAM) (Yan et al., 2025a) employs a specialized researcher agent for deep research over a universal page-store, enabling dynamic context reconstruction for complex multi-hop reasoning. IterResearch (Chen et al., 2025c) maintains a workspace preserving only the evolving report and immediate results to prevent context suffocation. MirrorMind (Zeng et al., 2025) simulates collective intelligence through hierarchical architecture retrieving specific cognitive styles and knowledge bases. AISAC (Bhattacharya & Som, 2025) implements hybrid memory combining

Table 6: **Summarization of representative agentic memory applications.**

| Application | Memory Utilization | Works |
|---|---|---|
| Education | Tracks learner progress and simulates knowledge decay to provide personalized pedagogical guidance. | LOOM (Cui et al., 2025a), Agent4Edu (Gao et al., 2025b), WebCoach (Liu et al., 2025b), CAM (Li et al., 2025d), Classroom Simulacra (Xu et al., 2025b), TeachTune (Jin et al., 2025a), EduAgent (Xu et al., 2024a), EvaAI (Lagakis & Demetriadis, 2024), OATutor (Pardos et al., 2023), MEDCO (Wei et al., 2024) |
| Scientific Research | Synthesizes vast literature and maintains reasoning provenance across multi-stage discovery processes. | IterResearch (Chen et al., 2025c), GAM (Yan et al., 2025a), MirrorMind (Zeng et al., 2025), AISAC (Bhattacharya & Som, 2025), Lee et al. (2024), ChemDFM (Zhao et al., 2025e), AI-coscientist (Gottweis et al., 2025), SciAgents (Ghafarollahi & Buehler, 2025), Agent Laboratory (Schmidgall et al., 2025), NovelSeek (Team et al., 2025) |
| Gaming & Simulation | Enables bottom-up skill acquisition and the emergence of complex social dynamics through episodic memories. | Voyager (Wang et al., 2025c), GITM (Zhu et al., 2023), Generative Agents (Park et al., 2023), GameGPT (Chen et al., 2023), M2PA (Zhou et al., 2025b), Jiang et al. (2025d), WarAgent (Hua et al., 2023), S3 (Gao et al., 2023), AvalonBench (Light et al., 2023), Mosaic (Liu et al., 2025c) |
| Robotics | Bridges high-level reasoning with low-level control by maintaining spatial graphs and trajectory summaries. | Memo (Gupta et al., 2025), MG-Nav (Wang et al., 2025b), JARVIS-1 (Wang et al., 2024p), KARMA (Wang et al., 2025t), VIPeR (Ming et al., 2025), SAM 2 (Ravi et al., 2024), LRLL (Tziafas & Kasaei, 2024), VideoAgent (Fan et al., 2024), Kim et al. (2023a), RAP (Kagaya et al., 2024), GridMM (Wang et al., 2023d) |
| Healthcare | Maintains longitudinal records of physiological trends and emotional states to build user trust and adherence. | TheraMind (Hu et al., 2025a), DAM (Lu & Li, 2025), Mem-PAL (Huang et al., 2025d), ReSurgSAM2 (Liu et al., 2025d), CARE-AD (Li et al., 2025e), AgentMD (Jin et al., 2025b), MedConMA (Wang et al., 2025k), MDAgents (Kim et al., 2024b), MedAgents (Tang et al., 2024), ChatCAD (Tang et al., 2025a) |
| Dialogue Systems | Manages context window constraints and persona consistency to simulate persistent human-like relationships. | A-Mem (Xu et al., 2025e), MemGPT (Packer et al., 2023), O-Mem (Wang et al., 2025i), MemoChat (Lu et al., 2023), Mem0 (Chhikara et al., 2025), SEAL (Wang et al., 2025d), LiCoMemory (Huang et al., 2025e), Lu & Li (2025), Terranova et al. (2025), LightMem (Fang et al., 2025a), RGMem (Tian et al., 2025) |
| Workflow Automation | Induces reusable workflow templates and learns tool-usage patterns from successful execution histories. | AWM (Wang et al., 2025v), ToolMem (Xiao et al., 2025), Synapse (Zheng et al., 2024), WebArena (Zhou et al., 2024), Wheeler & Jeunen (2025), Wang et al. (2025u), WALT (Prabhu et al., 2025), Mobile-agent-v2 (Wang et al., 2024c), AutoAgents (Chen et al., 2024a), SIT-Graph (Li et al., 2025f) |
| Software Engineering | Maintains global code context and recalls failure trajectories to improve multi-file debugging and development. | MetaGPT (Hong et al., 2024), ChatDev (Qian et al., 2024a), SWE-bench (Jimenez et al., 2024), SWE-Effi (Fan et al., 2025c), TroVE (Wang et al., 2024o), Self-organized agents (Ishibashi & Nishimura, 2024), Openhands (Wang et al., 2025n), Masai (Arora et al., 2024), DeepCode (Li et al., 2025m) |
| Online Streaming & Recommendation | Distills high-throughput multimodal feeds into persistent representations to recognize long-range temporal patterns. | WorldMM (Yeo et al., 2025b), GCAgent (Yeo et al., 2025a), XMem++ (Bekuzarov et al., 2023), VideoScan (Li et al., 2025c), Xiong et al. (2025b), Qian et al. (2024b), VideoLLM-online (Chen et al., 2024c), VideoLLM-MoD (Wu et al., 2024c), Di et al. (2025) |
| Information Search | Transforms static retrieval into active workspaces for synthesizing conflicting reports and tracking search provenance. | AgentFold (Ye et al., 2025b), MemSearcher (Yuan et al., 2025a), MoM (Zhao et al., 2025a), ReSum (Wu et al., 2025d), Memento (Zhou et al., 2025a), MLP Memory (Wei et al., 2025c), MemAgent (Yu et al., 2025b), Wang et al. (2025p), MemoryLLM (Wang et al., 2024l) |
| Finance & Accounting | Maintains strategic consistency across volatile market cycles and balances quantitative signals with qualitative historical precedents. | FinCon (Yu et al., 2024a), FinMem (Yu et al., 2025e), QuantAgent (Wang et al., 2024g), FLAG-Trader (Xiong et al., 2025a), Investor-Bench (Li et al., 2025b), TradingAgents (Xiao et al., 2024), Trading-GPT (Li et al., 2023b), Open-FinLLMs (Huang et al., 2024a) |
| Legal & Consulting | Manages multi-document provenance and synthesizes conflicting statutes into coherent advice across long-term case histories. | MALR (Yuan et al., 2024b), StaffPro (Maritan, 2025), Blair-Stanek et al. (2025), LegalMind (Vara et al., 2025), CaseGPT (Yang, 2024), Dallma (Westermann, 2024), AgentCourt (Chen et al., 2025b), Legal-GPT (Shi et al., 2024), Feat (Shen et al., 2025) |

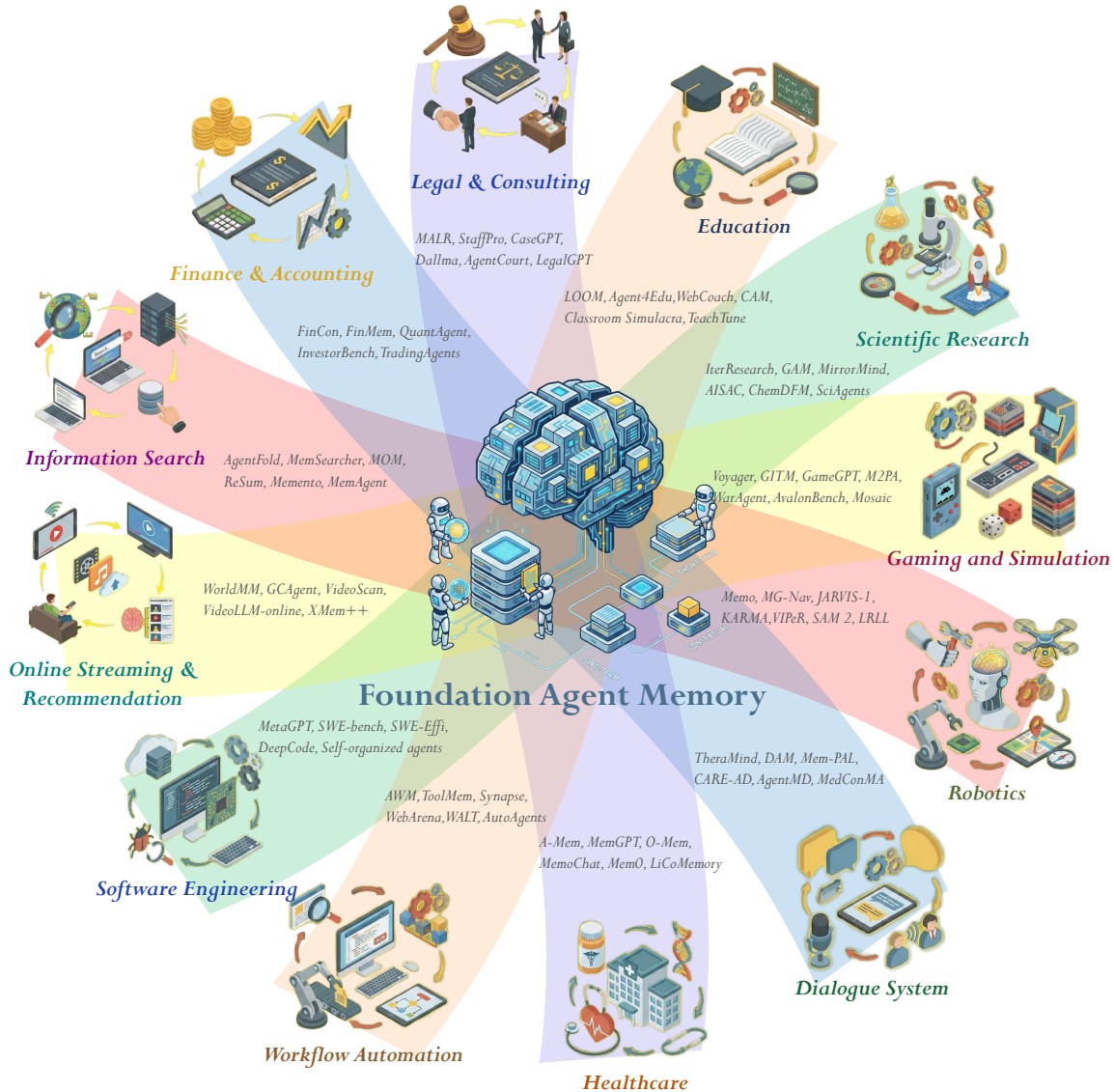

Figure 8: **Applications of the Foundation Agent Memory System**. The diagram introduces the general application domains of the foundation agent memory system, including *education, scientific research, gaming and simulation, robotics, dialogue systems, healthcare, workflow automation, software engineering, online streaming and recommendation, information search, finance and accounting,* and *legal and consulting.*

semantic retrieval with structured SQLite logs for reproducibility. These research agents exemplify a paradigm shift where memory serves as a verification layer for the research process, maintaining a transparent lineage of how a conclusion was reached. Future systems will evolve from solitary research assistants into lab-scale collective intelligences, where multiple agents share a unified, evolving knowledge graph of a specific scientific field, updating it in real-time as new papers are published and synthesized.

**Gaming and Simulation.** In open-ended gaming environments and simulations, memory enables skill acquisition, spatial exploration, and the emergence of complex social dynamics. Agents must utilize procedural memory to retain learned skills and episodic memory to maintain believable social histories. For example, Voyager (Wang et al., 2025c) stores successful actions as executable code in a skill library for compounding abilities. GITM (Zhu et al., 2023) employs hierarchical text-based memory where a planner records structured

sub-goal summaries. Generative Agents (Park et al., 2023) uses a memory stream where agents reflect to synthesize high-level insights into relationships and plans. GameGPT (Chen et al., 2023) applies memory as shared state for multi-agent game development, managing versioning and conflict resolution. In these works, memory modules allow behavior to emerge bottom-up from the accumulation of experiences rather than top-down programming. The next frontier in this domain could be the development of social alignment and forgetting mechanisms. As simulations run for extended periods, agents should mimic human-like memory decay, ensuring personalities evolve organically without being paralyzed by the noise of infinite, trivial historical data.

**Robotics.** Embodied agents operating in physical worlds face the challenge of partial observability (Fung et al., 2025), requiring memory to link visual inputs to semantic concepts and maintain spatial representations over time. Memory must be compressed yet sufficiently detailed to support navigation and manipulation in non-static environments. Memo (Gupta et al., 2025) introduces periodic summarization tokens compressing trajectories for long-horizon navigation. MG-Nav (Wang et al., 2025b) constructs spatial memory graphs with landmark regions rather than dense point clouds, mimicking human navigation. JARVIS-1 (Wang et al., 2024p) extends embodied agency with multimodal memory retrieving experiences based on visual and semantic similarity. These applications demonstrate that memory is the bridge between high-level reasoning and low-level control. Future research should focus on multimodal memory integration, enabling agents to simulate the physical affordances based on past successes and failures stored in their procedural memory.

**Healthcare.** In the domain of healthcare, memory enables agents to track longitudinal health trends, emotional trajectories, and the efficacy of interventions. TheraMind (Hu et al., 2025a) introduces a dual-loop architecture separating immediate responses from strategic cross-session memory updates for therapeutic strategy adjustment. DAM (Lu & Li, 2025) treats memory units as confidence distributions over sentiment polarities for stable probabilistic emotion modeling. Mem-PAL (Huang et al., 2025d) employs H²Memory architecture distinguishing between objective physiological logs and subjective dialogue to infer health metric correlations. The deployment of agentic memory reveals that affective continuity is as critical as clinical accuracy, leading to a measurable increase in user trust and adherence. However, this domain faces challenges regarding privacy and ethics. Future architectures should implement privacy-preserving memory by design mechanisms that balance the utility of long-term memory with the imperative of patient confidentiality.

**Dialogue Systems.** For general-purpose assistants, memory creates the illusion of a continuous personalized relationship, managing the context window while providing conversation history. This domain focuses on actively managing the trade-off between retention and context window constraints. MemGPT (Packer et al., 2023) introduces a context management system explicitly moving data between main and external context for longer conversations. O-Mem (Wang et al., 2025i) uses tri-component memory to extract and update holistic user personas for aligned responses. MemoChat (Lu et al., 2023) employs instructional tuning to train models on writing structured memos for improved long-range consistency. These dialogue architectures demonstrate a shift toward OS-level memory management, where the agent acts as a kernel managing its own resources. The future of dialogue systems lies in self-optimizing memory. Rather than relying on fixed heuristic rules for what to remember, next-generation agents will likely learn personalized memory policies.

**Workflow Automation.** LLM agents also work as assistants to boost productivity through workflow automation. Automation agents mainly orchestrate multi-step processes, coordinate tool usage, and adapt procedures based on execution feedback. Memory mechanisms enable agents to accumulate procedural knowledge, optimize workflow, and maintain task context across complex automation pipelines. AWM (Wang et al., 2025v) induces reusable workflow templates from successful trajectories as parameterized procedural memory. ToolMem (Xiao et al., 2025) implements semantic memory of tool usage patterns, learning effective tools for specific task types. On WebArena (Zhou et al., 2024), agents that retain episodic memory of web interaction sequences substantially outperform memoryless baselines. Synapse (Zheng et al., 2024) introduces trajectory-as-exemplar prompting, storing successful control sequences as episodic memory for analogical reasoning. In these works, the integration of memory facilitates the transition from rigid script execution to adaptive procedural learning, which dramatically increases robustness and efficiency. Future systems will not just follow human-defined operation procedures but should actively rewrite their own instructions based on long-term execution logs, evolving from simple task executors into process architects that autonomously refine enterprise workflows for maximum efficiency.

**Software Engineering.** For software engineering, agents operate in complex codebases requiring long-horizon reasoning, multi-file coordination, and accumulated debugging experience. Memory enables these agents to maintain code context, learn from implementation attempts, and navigate large-scale repositories. MetaGPT (Hong et al., 2024) implements procedural memory for development workflows while maintaining shared semantic memory of project specifications. ChatDev (Qian et al., 2024a) extends this with episodic memory of development iterations for learning from debugging sessions. Evaluations on SWE-bench (Jimenez et al., 2024) report that memory mechanisms improve issue resolution by maintaining context across multi-file edits and leveraging prior debugging experience. Critically, effective coding agents do not merely generate code; they recall the trajectory of previous failures and fixes to achieve higher success rates in passing unit tests. Future applications in this domain will likely move beyond local project memory toward shared, anonymized knowledge repositories where distributed coding agents contribute to and query a universal pool of algorithmic solutions and error patches, accelerating the global pace of automated software development.

**Online Streaming and Recommendation.** In the era of online streaming and recommendation systems, agents process high-throughput multimodal inputs where the relevance of information shifts dynamically over time. Memory allows agents to maintain temporal consistency and recognize long-range patterns across video frames and user interactions. For instance, WorldMM (Yeo et al., 2025b) utilizes dynamic multimodal memory storing visual-linguistic features for complex reasoning over long-duration video streams. GCAgent (Yeo et al., 2025a) introduces dual-structured episodic memory separating schematic knowledge from narrative sequences for structured video understanding. Similarly, Xiong et al. (2025b) implements memory-enhanced knowledge buffers supporting multi-round interactions with retained context. These applications suggest that in streaming contexts, memory acts as a temporal filter that distills transient data into persistent representations. Future research should focus on forgetting-aware recommendation memories that can distinguish between a user's fleeting interests and their long-term preferences, optimizing the balance between novelty and relevance in real-time feeds.

**Information Search.** Beyond simple retrieval, agents for information search must synthesize conflicting reports, track evolving stories, and manage vast document spaces without losing reasoning depth. Memory serves as the organizational framework that transforms static search results into an active workspace for knowledge synthesis. AgentFold (Ye et al., 2025b) addresses long-horizon web navigation through proactive context management, folding irrelevant trajectories to prevent overflow while preserving critical findings. MemSearcher (Yuan et al., 2025a) employs reinforcement learning for joint searching and memory management. MoM (Zhao et al., 2025a) utilizes scenario-aware memories, dynamically routing queries to specialized memory banks. Furthermore, Rajesh et al. (2025) bridges RAG with episodic memory, maintaining a repository of the search process itself. These systems demonstrate that effective search is not just about finding data, but about managing the cognitive load of the search trajectory. The next generation of information search agents will likely evolve toward collaborative memory structures, where multiple agents verify facts and update a shared belief graph among agents in response to breaking news cycles.

**Finance and Accounting.** The financial domain is characterized by high-frequency volatility, a mixture of quantitative data and qualitative news, and the critical need for long-term strategic consistency. Memory is crucial here because trading and accounting require agents to process real-time signals and recall historical market regimes and maintain a persistent trading character to avoid erratic decision-making. Recent works have introduced specialized architectures for this purpose. FinMem (Yu et al., 2025e) implements a layered memory system that separates immediate market observations from long-term investment experience, allowing the agent to refine its personality and risk profile over time. FinCon (Yu et al., 2024a) utilizes a multi-agent setup where memory serves as a repository for conceptual verbal reinforcement, enabling the system to learn from past financial decisions through reflective feedback loops. QuantAgent (Wang et al., 2024g) further pushes this by seeking investment grails through a self-improving mechanism where success trajectories are stored as procedural memory for future strategy refinement. Despite these gains, the primary challenge remains the signal-to-noise ratio in financial memory. Agents must learn to distinguish between transient market fluctuations and fundamental shifts. Future research should explore forgetting-aware financial agents that can prune outdated economic assumptions while retaining core risk management principles.

**Legal and Consulting.** In legal and consulting services, agents must navigate massive volumes of heterogeneous documents, where the precise provenance of every claim is mandatory. Memory is the cognitive

substrate that allows these agents to perform multi-step reasoning over long-duration cases, ensuring that advice remains consistent with previously cited statutes or client history. For instance, MALR (Yuan et al., 2024b) utilizes a multi-agent framework to improve complex legal reasoning by maintaining an interaction history that simulates collaborative debate between legal experts. StaffPro (Maritan, 2025) focuses on the consulting side by using memory to profile workers and project requirements over time, enabling dynamic staffing through a feedback loop of past performance data. Blair-Stanek et al. (2025) demonstrates the power of memory in discovering novel tax-minimization strategies by synthesizing thousands of pages of evolving statutes and case law into a persistent reasoning graph. The core challenge in this domain is the high stakes of hallucinated memory. A single misremembered clause can lead to legal liability. Future directions will likely focus on verifiable memory architectures that link every retrieved insight back to a cryptographically signed source document, ensuring the highest levels of professional integrity and accountability.

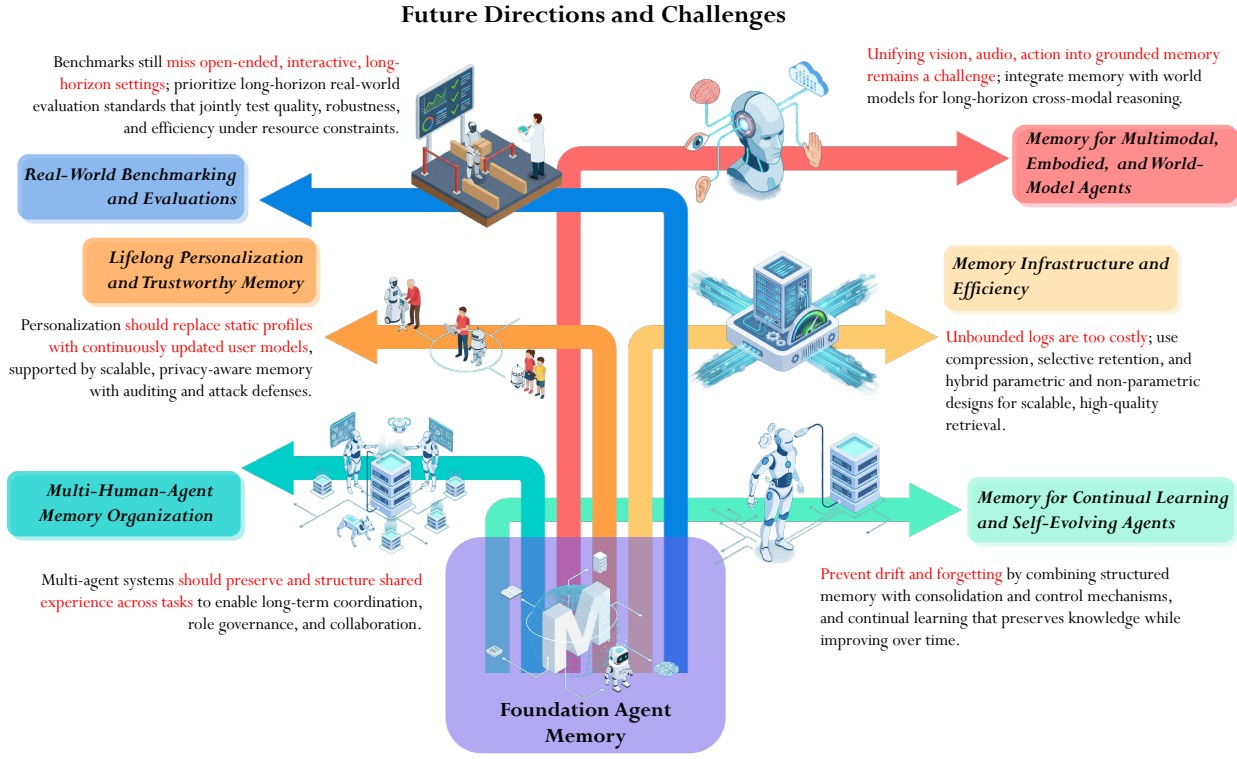

Figure 9: **Future Directions and Challenges in Foundation Agent Memory**. The diagram highlights key opportunities for future agentic memory works, including *memory for continual learning and self-evolving agents, multi-human-agent memory organization, memory infrastructure and efficiency, lifelong personalization and trustworthy memory, memory for multimodal, embodied, and world-model agents*, and *real-world benchmarking and evaluations*. Each direction pairs a challenge with a promising research path.

# 9 Future Directions

Building on the preceding analysis, we highlight six open challenges that we consider most consequential for foundation agent memory. Figure 9 summarizes them together with the directions we see as most promising for each.

## 9.1 Memory for Continual Learning and Self-Evolving Agents

A fundamental challenge in memory-enabled self-evolving agents lies in managing memory dynamics across both intra-task and cross-task timescales. At the intra-task level, agents must continuously decide what

information to retain, compress, or discard from heterogeneous streams such as tool outputs, search results, feedback signals, and intermediate reasoning traces, all under strict context-window constraints (Gao et al., 2025a). Existing systems largely rely on heuristic memory controllers, leaving the internal coupling between memory evolution and reasoning behavior poorly understood. At the cross-task level, agents are expected to accumulate experience across episodes and task distributions (Wei et al., 2025d), yet current approaches primarily emphasize inference-time reuse rather than principled consolidation and generalization. In comparison, classical continual learning methods focus on preventing catastrophic forgetting through replay, regularization, or parameter isolation (Rebuffi et al., 2017; Lopez-Paz & Ranzato, 2017; Shin et al., 2017; Kemker & Kanan, 2018), but they typically treat memory as a static mechanism for knowledge retention. This framing is insufficient for agent-based systems, where memory must also track evolving interaction states, user-specific information, and procedural behaviors. Moreover, stable post-training adaptation from accumulated experience remains underexplored, with unresolved risks of negative transfer, uncontrolled drift, and semantic inconsistency (Ke et al., 2025b).

Future research should therefore reframe continual learning around a richer, agent-centric notion of memory that integrates semantic, episodic, and procedural components (Ke et al., 2025a; Sumers et al., 2023). Beyond explicit textual logs, latent and structured memory representations offer a promising direction for scalable and efficient adaptation, enabling compact storage while preserving causal and behavioral abstractions. Progress will likely require moving beyond inference-time heuristics toward principled post-training paradigms that leverage accumulated agent experience for continual improvement. This includes designing consolidation mechanisms that selectively distill long-term knowledge, align evolving memory with model parameters, and mitigate forgetting without sacrificing plasticity. Correspondingly, new benchmarks are needed that evaluate not only task-level retention, but also sustained adaptation, relevance-aware memory management, and behavioral stability under non-stationary objectives and environments (Ke et al., 2024). Establishing a unified framework that connects classical continual learning objectives with structured memory design remains a key open direction for self-evolving foundation agents.

## 9.2 Multi-Human-Agent Memory Organization

Recent multi-agent LLM frameworks, such as AutoGen (Wu et al., 2024b) and AgentLite (Liu et al., 2024c), enable task decomposition and role-based coordination through structured message passing and prompt-driven control. In practice, such systems increasingly operate in *human-agent collaborative settings*, where artificial agents interact not only with other agents but also with human users or supervisors through iterative feedback, correction, and delegation (Lu et al., 2025c; Zou et al., 2025b;c), with benchmarks beginning to evaluate such systems under configurable human participation and explicit role and permission structures (Wu et al., 2026). However, despite this growing complexity, coordination remains largely *episodic and transient*: interactions are scoped to a single task instance, and little experience is retained once the task is completed (Suzgun et al., 2025). As a result, both agent-agent and human-agent collaborations are repeatedly re-established from scratch, limiting the system's ability to adapt interaction strategies, personalize behavior, or improve collaboration quality across repeated tasks or deployments.

Enabling persistent and adaptive collaboration among interacting entities (including both foundation agents and humans) would inspire long-term research questions (Han et al., 2024; Li et al., 2024c). One important direction is *collaborative (social) memory*, where agents retain experience about their collaborators, such as communication preferences, domain expertise, feedback patterns, or historical interaction outcomes, allowing them to adapt signaling strategies, calibrate trust, and reduce coordination overhead over time. At the same time, agents may benefit from *role-specific workflow and procedural memory*, accumulating experience about their own recurring workflows (Wang et al., 2025v), including task decomposition patterns, execution strategies, and common failure modes, so that agents assuming stable functional roles can gradually refine their behavior through experience-driven specialization. Introducing persistent memory in such multi-entity settings also raises *memory governance and coordination* challenges, including questions of ownership, access, responsibility, and how divergent perspectives or human corrections should be handled. Addressing these issues is essential for preventing uncontrolled error propagation and for sustaining reliable collaboration as multi-agent systems scale in size, heterogeneity, and task complexity.

### 9.3 Memory Infrastructure and Efficiency

As foundation agents are increasingly deployed in long-horizon, interactive, and open-ended environments, memory infrastructure has emerged as a central efficiency bottleneck (Chhikara et al., 2025; Qiu et al., 2025b). Most existing agent memory designs remain text-centric, treating memory as an ever-growing collection of past interactions, summaries, or episodic logs that are retrieved and injected into the prompt (Xu et al., 2025e; Zhong et al., 2024). While this strategy can improve task performance, it induces substantial token overhead, with memory contexts routinely expanding to thousands of tokens and exhibiting diminishing marginal returns (Chhikara et al., 2025). This linear growth in memory cost (Kwon et al., 2023) directly translates into higher inference latency and reduced scalability, particularly in multi-turn or lifelong settings. More fundamentally, current approaches conflate memory capacity with prompt length, implicitly assuming that more context implies better reasoning. This assumption overlooks the need for selective retention, structured access, and consolidation of experience. Moreover, memory is often managed externally through heuristics such as summarization or truncation, rather than being integrated into the agent's reasoning and learning process. These limitations highlight a core challenge: how to design memory systems that enable agents to retain, abstract, and reuse experience efficiently under strict resource constraints, without relying on unbounded context expansion.

Future research on memory infrastructure and efficiency can be viewed as a progression toward increasingly abstract and integrated representations of experience. In the near term, a promising direction lies in organized text-based memory, where textual memories are explicitly structured for efficient access rather than maximal coverage. Recent work has explored schema-based or graph-structured memory representations (Edge et al., 2024; Chhikara et al., 2025), but these efforts primarily target reasoning accuracy rather than efficiency. An open opportunity is to design structure-aware storage and precision-oriented retrieval mechanisms that expose only reasoning-critical spans, minimizing unnecessary context injection. Beyond textual organization, efficiency gains may be achieved through compressed latent memory, where episodic, semantic, or procedural experiences are encoded into compact vector representations that function as persistent memory units rather than mere similarity indices. At a deeper level of integration, internalized or parametric memory offers a path toward constant-sized memory, where long-term experience is absorbed into internal states or model parameters. Frameworks such as MEM1 (Zhou et al., 2025c) and Mem-$\alpha$ (Wang et al., 2025q) exemplify this shift by training agents, via reinforcement learning, to consolidate, update, and discard memory as part of the reasoning process itself, enabling bounded memory even in long-horizon tasks. Realizing these directions will also require robust environment infrastructure capable of supporting controlled, multi-step interactions and scalable evaluation. Platforms such as NeMo Gym (NVIDIA, 2025), which decouple environment logic from training and provide modular reward and verification services, represent an essential component of this ecosystem. Together, these advances suggest a future in which memory is no longer an external prompt-management artifact but a core, learned subsystem co-evolving with agent reasoning and decision-making, realized through integrated memory architectures that combine structured latent representations (e.g., hierarchical vector tables with differentiable read/write interfaces), joint optimization of memory and policy via end-to-end reinforcement learning or meta-learning objectives, and adaptive memory controllers that dynamically allocate, compress, and retire memory units based on task relevance, uncertainty estimates, and long-term utility.

### 9.4 Lifelong Personalization and Trustworthy Memory

Lifelong personalization seeks to equip foundation agents with the ability to continuously adapt to individual users across sessions, tasks, and extended time horizons (Wang et al., 2024i). Unlike conventional personalization approaches that rely on static user profiles or transient contextual signals, this setting requires agents to maintain evolving user representations that capture gradual preference shifts, long-term goals, and behavioral regularities. While recent efforts on persistent memory and dynamic user modeling have made initial progress (Zhong et al., 2024; Tan et al., 2025c; Zhang et al., 2025o), existing systems largely depend on heuristic aggregation of interaction histories or unstructured memory retrieval, which limits their ability to distill reliable, interpretable, and causally grounded user knowledge (Pink et al., 2025). Moreover, long-horizon personalization introduces non-trivial challenges in memory staleness, concept drift, and credit assignment: agents must decide which past interactions remain relevant, how to reconcile conflicting signals over time, and how to prevent outdated preferences from dominating current behavior. These issues are further exacerbated

by scalability constraints, as naively retaining or replaying long interaction histories leads to prohibitive storage, retrieval, and inference costs, especially when deployed in real-world, always-on assistant settings.

A key research direction is the design of scalable and dynamic memory systems that can incrementally update user modeling while bridging fine-grained episodic traces with higher-level abstractions such as preferences, habits, or long-term intents. Promising approaches include hierarchical memory architectures that separate short-term episodic buffers from distilled semantic user profiles (Tan et al., 2025c), learned memory controllers that regulate when to write, compress, or overwrite user information (Zhang et al., 2025o), and continual representation learning techniques that mitigate forgetting under distribution shift (De Lange et al., 2021; Parisi et al., 2019). In parallel, the field requires new evaluation benchmarks tailored to lifelong personalization, moving beyond single-turn accuracy toward metrics that assess long-term consistency, adaptability to preference changes, and robustness under extended interactions (Xu et al., 2025e). Equally important is the development of trustworthy memory infrastructures. Persistent user memory raises substantial risks, including privacy leakage (Wang et al., 2025a), memory poisoning (Tan et al., 2024b), and adversarial manipulation (Dong et al., 2025), which can accumulate silently over time. Recent work on secure and auditable memory modules (Wei et al., 2025b; Wang et al., 2025a) highlights the need for user-controllable mechanisms that support inspection, editing, and revocation of stored memories, alongside defenses against unauthorized access and malicious writes. Ultimately, robustness, transparency, and security should be treated as first-class objectives, on par with adaptability, when designing and evaluating lifelong personalized foundation agents (Yu et al., 2025c).

### 9.5 Memory for Multimodal, Embodied, and World-Model Agents

A central challenge for next-generation foundation agents lies in designing memory systems that can faithfully represent, align, and abstract heterogeneous sensory streams, including vision, audio, language, tactile feedback, and proprioceptive signals, into coherent internal states (Bei et al., 2026). While textual memory mechanisms have achieved notable success in long-horizon reasoning and personalization (Xu et al., 2025e), existing approaches largely assume unimodal or language-dominant representations. Early efforts in multimodal agent memory (Long et al., 2025; Bo et al., 2025; Liu et al., 2025g) reveal that naively extending text-based memory to high-dimensional perceptual inputs leads to severe inefficiencies, semantic misalignment across modalities, and brittle retrieval behaviors. These challenges are further amplified in embodied settings, where agents operate in closed-loop environments and must reason over temporally extended perception–action–outcome trajectories. In such scenarios, memory must go beyond storing episodic observations and instead encode grounded knowledge about dynamics, affordances, and physical constraints (Wang et al., 2025t). However, current systems lack principled mechanisms for action-conditioned memory updates, cross-modal abstraction, and consistency maintenance across episodic, semantic, and procedural memory layers. As a result, embodied agents often struggle with skill fragmentation, long-horizon planning failures, and compounding errors caused by misaligned or stale memories.

Looking forward, a promising research direction is to elevate agent memory into an explicit, predictive world model that treats memory not as a passive log, but as a controllable internal state evolving over time. World-model-based formulations (Hafner et al., 2023; Ha & Schmidhuber, 2018) provide a unifying perspective in which memory updates can be modeled as latent state transitions conditioned on perception and action. This opens the door to *proactive memory planning*, where agents simulate the long-term consequences of storing, compressing, or forgetting information before committing updates (Schrittwieser et al., 2020; Silver et al., 2017). Within this framework, memory operations become internal actions optimized jointly with external decision-making, enabling agents to balance immediate utility with long-term consistency and task performance. Moreover, integrating multimodal memory with structured world representations, such as spatial maps, object-centric graphs (Singh et al., 2023), or skill graphs (Wang et al., 2025c; Feng et al., 2025a), can support abstraction across time and modality while improving retrieval efficiency. Finally, memory and world models should be co-trained in a mutually reinforcing loop: stable, structured memory can provide long-term state cues that improve world-model prediction, while world models can regularize memory evolution to prevent identity drift, goal inconsistency, and behavioral instability (Savinov et al., 2019). Advancing this synergy is key to building scalable, reliable multimodal and embodied agents capable of long-horizon autonomy in complex real-world environments.

### 9.6 Real-World Benchmarking and Evaluations

A central challenge in real-world benchmarking for memory-enabled foundation agents lies in the persistent mismatch between research-level benchmark abstractions and real-world deployment complexities, for both user-centric and agent-centric memory. On the user side, most existing benchmarks reduce long-term personalization to synthetic factual recall, where agents retrieve static user attributes embedded in long contexts or scripted interaction histories (e.g., persona facts, preferences, or conversations). While such settings facilitate controlled evaluation, they fail to capture real user satisfaction, which depends on preference drift, conflicting signals, partial observability, and delayed feedback over weeks or months. Benchmarks such as LoCoMo (Maharana et al., 2024) emphasize long-context retrieval accuracy, yet implicitly assume stationary user intent and unambiguous ground truth, overlooking critical failure modes such as stale preference reuse, incorrect overwriting of long-term user state, or unsafe retention of sensitive information. Even PersonaMem (Jiang et al., 2025a), which explicitly targets evolving preferences, evaluates them over fully simulated sessions and does not assess memory update or refresh. On the agent-centric side, interactive benchmarks including WebArena (Zhou et al., 2024) and OSWorld (Xie et al., 2024) improve realism through execution-based evaluation, but remain bounded by curated environments, reset-centric task design, and short evaluation horizons. Recent extensions begin to relax the first of these: InterruptBench augments WebArena-Lite with mid-task additions, revisions, and retractions of the user's goal, and reports that handling them remains difficult even for strong backbones (Zou et al., 2026). These constraints obscure whether agents can accumulate, revise, and safely exploit experience across episodes, especially under non-stationary tools, policies, or environments. As a result, agents may optimize for short-horizon success while silently failing at memory-critical competencies such as provenance tracking, contradiction resolution, and long-term policy consistency. Probing hidden user state directly makes this gap measurable: task completion saturates even for a memoryless baseline, while recovery of the user's evolving state remains moderate and degrades further under top-$K$ retrieval (Ma et al., 2026). This gap mirrors observations in general assistant benchmarks such as GAIA (Mialon et al., 2024), where failures often arise not from isolated reasoning errors but from brittle state transitions and incorrect memory updates across multimodal and tool-mediated interactions.

Future research should move toward closed-loop, longitudinal, and execution-grounded evaluation paradigms that explicitly stress persistent memory under realistic constraints, for both users and agents. For user-centric memory, benchmarks should incorporate recurring interactions with controlled preference drift, ambiguous feedback, and real-user rewards, enabling direct measurement of satisfaction-aligned memory behaviors such as compression, selective forgetting, and safe overwriting, rather than static recall accuracy (e.g., extending user history with multi-month preference evolution or counterfactual feedback). For agent-centric memory, evaluation should go beyond simulated resets toward partially open or continuously evolving environments, where experience accumulation has real consequences, such as financial trading sandboxes, long-running web services, or competitive control tasks with delayed payoffs, enabling comparison between memory-augmented agents and memory-free baselines under identical conditions. Execution-based frameworks like OSWorld (Xie et al., 2024) can be extended with memory-sensitive invariants, requiring agents to version, audit, and roll back persistent state, and to attach provenance metadata to stored knowledge. In parallel, standardized tool-mediation layers (e.g., MCP-style interfaces) can enable reproducible logging, permission enforcement, and replay, supporting fine-grained evaluation of memory–policy interactions under realistic constraints. Finally, benchmarks should explicitly quantify resource–utility trade-offs, measuring memory quality as a function of token budget, storage cost, and latency, reflecting the bounded-memory conditions of real deployments. Collectively, these directions reframe benchmarking from episodic task completion toward systems-level evaluation of memory as a first-class capability, jointly shaping user trust, agent autonomy, and long-term utility across evolving environments.

## 10 Conclusions

Memory is becoming the key component for foundation agents operating in long-horizon, context-exploding, and user-dependent environments such as agentic coding, deep research, and computer use. In this survey, we unify the design along three dimensions, including memory substrates (internal and external), cognitive mechanisms (sensory, working, episodic, semantic, and procedural), and memory subjects (user-centric

personalization and agent-centric experience), and analyze how memory is operated under single- and multi-agent systems, as well as how it is increasingly shaped by prompting-, fine-tuning-, and reinforcement-learning-based evolution and optimization policies. In addition, we summarize the metrics and benchmarks used to assess foundation agent performance, and categorize current works into representative application domains. Throughout, memory shifts from passive storage toward the substrate of agent self-evolution, and we close with six key challenges pointing to a reliable, scalable, and trustworthy memory infrastructure.

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
