# OpenReview forum: "Rethinking Memory Mechanisms of Foundation Agents in the Second Half: A Survey"
_TMLR — Decision pending for TMLR_

### Review · Reviewer_yaUX · 2026-03-01

**Summary Of Contributions:**

This article presents a comprehensive survey on the memory of foundamation model based agents. Authors unify memory design across three dimensions: memory substrate (internal and external), cognitive mechanisms (sensory, working, episodic, semantic, and procedural), and memory subjects (user- and agent-centric). They analyze how memory operates in single-agent and multi-agent systems and how it is increasingly influenced by prompting, fine-tuning, and reinforcement learning strategies. Furthermore, they summarize metrics and benchmarks for evaluating foundational agent performance and categorize current work into representative application areas. Finally, they outline six key challenges for future research.

**Audience:**

Yes

**Audience Explanation:**

Memory plays a key role for foundamation model based agents to work in real world. This article presents a comprehensive survey on memory mechanisms of such agents.

**Claims And Evidence:**

No

**Claims Explanation:**

This is a survey paper that does not provide any experimental evidence to support its claims.

**Requested Changes:**

I believe this is an excellent paper; it diligently organizes and analyzes relevant literature, forming a comprehensive review. However, I would like to ask: what is the unique contribution of this paper compared to existing reviews? Is it merely a compilation and organization of knowledge? I suggest clarifying this point in the revised version of the paper. Furthermore, what was the motivation for writing this paper? In the era of large-scale models, what is the value of a review? I hope the authors can answer these questions.

---

> ### Author Response · Authors · 2026-04-28
>
> We thank the reviewer yaUX for the accurate summary. The reviewer correctly notes that our survey unifies memory design across memory substrate, cognitive mechanisms, and memory subjects; analyzes memory in single-agent and multi-agent systems; discusses prompting, fine-tuning, and reinforcement learning strategies; summarizes metrics and benchmarks; categorizes representative application areas; and outlines key challenges for future research.
>
> ## Requested Changes
>
> > R1: Clarify the unique contribution compared with existing reviews
>
> Our survey goes beyond compilation in three main ways.
>
> First, we provide a three-dimensional taxonomy of foundation-agent memory organized by memory substrate, cognitive mechanism, and memory subject. This taxonomy connects implementation form, functional role, and optimization target, which are often discussed separately in prior reviews. In particular, we highlight user-centric memory for user modeling and agent-centric memory for skill evolution.
>
> Second, we analyze memory from a system-design perspective. Rather than only grouping prior work by applications or memory types, we examine how memory is instantiated, operated, updated, and evaluated within agent systems, including single-agent and multi-agent topologies.
>
> Third, we position memory as a central mechanism for the “second half” of AI development, where the focus shifts from benchmark performance to real-world utility in long-horizon, dynamic, and user-dependent environments.
>
> > R2: Clarify the motivation for writing this survey
>
> The motivation for writing this survey comes from the rapid shift of foundation agents from short, isolated benchmark settings toward long-horizon, dynamic, and user-dependent environments. In such settings, agents must continuously accumulate, update, retrieve, and selectively reuse information across interactions. Memory therefore becomes a central mechanism for enabling personalization, task continuity, tool-use adaptation, and long-term behavioral consistency. However, the current literature is highly fragmented across RAG systems, long-context modeling, cognitive memory, personalization, continual learning, reflection, and agent evaluation. This fragmentation makes it difficult for researchers to understand how different memory mechanisms relate to one another and how to design memory systems for realistic agent deployments.
>
> > R3: Clarify the value of a review in the era of large-scale models
>
> This is a thought-provoking meta-question. This is a thought-provoking meta-question. In the era of large-scale models, we believe the value of a survey lies not only in summarizing existing work, but also in identifying key research directions and, importantly, in clarifying the essence of an increasingly crowded and emerging research domain.

---

### Review · Reviewer_atvc · 2026-03-18

**Summary Of Contributions:**

The paper presents a comprehensive survey on memory mechanisms for foundation agents, positioning memory as the critical component for agents to operate effectively in long-horizon, dynamic, and user-dependent environments. As the field transitions from idealized benchmark evaluations to real-world utility, the authors argue that agents must continuously accumulate, manage, and selectively reuse large volumes of information.

To conceptualize this, the authors propose a unified taxonomy of foundation agent memory organized along three distinct dimensions: memory substrate (internal vs. external), cognitive mechanism (episodic, semantic, sensory, working, and procedural), and memory subject (user-centric vs. agent-centric). Beyond the taxonomy, the survey systematically analyzes how memory operations are instantiated across different agent topologies (single vs. multi-agent) and highlights the emerging trend of using learning policies to optimize memory management. Finally, the paper reviews evaluation benchmarks and metrics for assessing memory utility, and outlines open challenges to guide future research.

**Audience:**

Yes

**Audience Explanation:**

See above.

**Claims And Evidence:**

Yes

**Claims Explanation:**

**Strengths**
1. The survey addresses a highly critical and rapidly expanding bottleneck in AI research (this is a timely survey). As agents face "context explosion," understanding how to build scalable memory systems is essential for real-world deployment.
2. The three-dimensional taxonomy offers a robust, system-level design perspective.
3. The paper has done an excellent job covering the entire lifecycle and ecosystem of agent memory.


**Weaknesses**
1. The hard distinction between "user-centric" and "agent-centric" memory subjects might be somewhat rigid. In practical applications like customized coding assistants or personalized deep research, an agent relies heavily on a blend of user preference history and task-oriented states, making the boundary less clear. (Also that Figure 2 has too many texts and the text color blends into the background/sub-figures, making it difficult to read. Especially the top-right corner part).
2. The survey occasionally sacrifices deep technical scrutiny. For example, while it mentions internal weight updates via continual learning and model editing, it stops short of thoroughly analyzing the algorithmic trade-offs (e.g., computational overhead) of these methods compared to external RAG-based systems.

**Requested Changes:**

**Suggestions**
1. Is that possible to add a table about the computational and operational trade-offs between internal parametric memory and external non-parametric memory? (including dimensions such as retrieval latency, storage costs, ease of updating/editing, and context window overhead would be highly valuable for the community)

---

> ### Author Response · Authors · 2026-04-28
>
> We thank the reviewer atvc for the accurate summary of our paper. Here is our response.
>
> > W1: User-centric and agent-centric memory may be too rigidly separated
>
> Indeed, in real-world scenarios, user-aligned preferences and task-oriented goals are typically intertwined in human-involved activities. However, (1) current benchmarks rarely incorporate user preferences while simultaneously requiring task completion, leading existing research to naturally diverge into two categories; and (2) these represent two largely orthogonal dimensions for model improvement, with most tasks emphasizing one dimension over the other. For example, in coding assistants, task completion is generally prioritized over user preferences. The relevant memory is the current code repository rather than the user’s historical coding patterns, since these are primarily productivity-oriented tasks. In contrast, recommendation tasks inherently require deep modeling of user preferences.
>
> We agree that, in practice, user-centric and agent-centric memory often interleave, and we will add a discussion of hybrid memory to avoid implying a rigid boundary. There is indeed prior work such as MemSkills that leverages both user-centric and agent-centric benchmarks. We will further clarify that these notions should be understood as conceptual orientations rather than mutually exclusive categories.
>
> Zhang, Haozhen, et al. "MemSkill: Learning and Evolving Memory Skills for Self-Evolving Agents." arXiv preprint arXiv:2602.02474 (2026).
>
> > W1: Figure 2 readability
>
> We agree that Figure 2 contains too much text and that some text blends into the background. We revise Figure 2 by reducing text density, increasing font size, improving color contrast, and simplifying visually dense regions, especially the top-right part. Some explanatory content will be moved from the figure into the caption or main text.

---

> ### Author Response · Authors · 2026-04-28
>
> > W2 & R1: Need deeper technical scrutiny of memory-substrate trade-offs
>
> We appreciate this comment. We will expand the discussion of technical trade-offs between memory substrates, especially internal weight-based memory, latent-state memory, KV-cache memory, and external RAG-based memory.
>
> We will analyze computational overhead, retrieval latency, storage cost, ease of update, risk of interference or catastrophic forgetting, context-window overhead, scalability, and robustness to noisy memory. This will make the discussion more useful for system builders and researchers designing practical memory architectures.
>
> **Add a table about computational and operational trade-offs**
>
> Table X: Comparative analysis of memory substrate trade-offs. We evaluate internal and external memory substrates across eight operational dimensions. Retrieval latency indicates the time cost of accessing stored information. Storage cost reflects the memory footprint required to maintain the substrate. Update flexibility captures the ease of modifying stored content. Context overhead denotes the additional burden imposed on the model's context window. Persistence indicates whether information survives across sessions. Scalability measures the substrate's capacity to accommodate growing memory demands. Forgetting risk quantifies susceptibility to knowledge degradation upon modification. Retrieval precision reflects the accuracy of targeting relevant stored information. Representative works are drawn from the taxonomy in §3.1.
>
> | Dimension | Internal Memory (§3.1.2) | External Memory (§3.1.1) |
> |----------|--------------------------|--------------------------|
> | Retrieval Latency | Low; knowledge is accessed through the forward pass or served from GPU-resident caches | Variable; dependent on index structure, corpus size, and retrieval algorithm |
> | Storage Cost | High; encompasses model parameters, hidden-state activations, and KV cache entries that scale with model depth and sequence length | Moderate; scales with the number of stored entries, independent of model architecture |
> | Update Flexibility | Low; parametric updates require fine-tuning, model editing, or distillation; latent states and KV caches are transient and non-editable | High; supports explicit insert, update, and delete operations without modifying model parameters |
> | Context Window Overhead | Significant for latent-state and KV cache substrates, which directly occupy the context budget; parametric memory imposes no overhead | Moderate; retrieved entries must be serialized and injected into the prompt |
> | Cross-Session Persistence | Partial; parametric knowledge persists permanently in weights, whereas latent states and KV caches are discarded at session termination | Permanent; entries are maintained in external storage across sessions |
> | Scalability | Constrained; parametric capacity is fixed at training time; KV cache size is bounded by available GPU memory | High; storage capacity grows independently of model architecture and can be expanded on demand |
> | Forgetting Risk | High for parametric substrates, which are susceptible to catastrophic forgetting upon weight updates; KV caches carry no forgetting risk but lack durability | Negligible; original entries are preserved in their stored form unless explicitly modified or deleted |
> | Retrieval Precision | Implicitly determined by learned associations for parametric memory; high within-session precision for KV cache via full attention over cached tokens | Variable; sensitive to embedding quality, similarity metrics, and ranking algorithms |
> | Representative Works | MemoryLLM, WISE, SELF-PARAM (parametric); vLLMPA, Titans, ChunkKV, EpiCache (latent-state/KV cache) | Mem0, Zep, A-Mem, HippoRAG2, MIRIX, MemTree |

---

### Review · Reviewer_3UtD · 2026-04-18

**Summary Of Contributions:**

**Summary**: This survey paper reviews more than 500 papers for recent advancement of memory research of agentic AI. It analyze foundation agent memory in three dimensions: (1) memory substrate, (2) memory cognitive mechanism, and (3) memory subject. It also review evaluation benchmarks and metrics.

**Strengths**
- Focused on memory system for web agents.
- Very comprehensive review of recent memory research paper for agents.
- Proposing well structured taxonomy of memory system (Fig. 4)
- Well organized contrast of cognitive memory and agent-AI's memory (Table 1)
- Good to have links for some references (e.g., Table 4)

**Weaknesses**
- It is missing that what is particularly important for memory for agents and memory for other AI systems
- Captions of figures should be self-contained or at least having pointers for the detailed contents.
 - Fig. 2's caption has no pointer to relevant sections for the detailed description.
 - For Fig. 3's caption, it is not clear what "external" and "internal" means in the left most figure. It is understood after reading whole section 3 (partly clear by referring to Fig. 2).
 - Fig. 5, dots and squares are not well described. What is the connection?
 - Table 4's caption is messy and has no pointer to corresponding sections.
- Some sections are not very clear (refer to 'requests or questions to the authors' section).
- Embodied AI agents are largely ignored. For example, metrics for embodied agents should include "success rate", "path length weighted success rate" and etc. But they are not addressed (also in other sections (than metric) as well). Although it is perfectly fine to confine the scope of the survey to web-agents, it is better to specify that the paper focuses on web-agents not embodied ones (though there are multiple locations that mention about it (e.g., Section 9.5) but not very serious on this). Or include embodied agents as a part of survey.

**Additional Comments:**

N/A

**Audience:**

Yes

**Audience Explanation:**

As there is no comprehensive survey on web agents with this much details (at least to my best knowledge), this gives a guideline for researchers in the field around web agents.

**Broader Impact Concerns:**

No concerns.

**Claims And Evidence:**

Yes

**Claims Explanation:**

Most of the descriptions are quite clearly stated and correct to the reviewer's best knowledge. But I didn't check all the words very precisely, thus there might be some inaccuracy in statements.

**Requested Changes:**

**Requests or questions to the authors**
- Define "foundation agent" clearly at the beginning of introduction. As there are so many different definition of agents, it is better to clearly define what the "foundation agent" refers to. It is in Section 2.1 and not immediately clear from the introduction.
- Why does "learning policy" contain "prompt-based learning"? What does "prompt-based learning"?

---

> ### Author Response · Authors · 2026-04-28
>
> We thank the reviewer 3UtD for recognizing the comprehensiveness of our survey and giving the detailed review of our paper. Here is our response.
>
> ## Weaknesses
> > W1: Missing what is particularly important for memory for agents and memory for other AI systems.
>
> We agree that the key points should be stated more explicitly. In the paper, we propose several future directions and argue that more realistic, long-horizon task settings are critical for the next generation of self-evolving, lifelong agents (Sections 6 and 9). For the other AI systems that focus on fixed tasks, effiency and cost would the key to design the memory systems. We will emphasize this more clearly in the introduction.
>
> > W2: Figure and table captions are not sufficiently self-contained
>
> The figure (2, 3, 5) and the table (4) caption will be revised. For instance, we update the Figure 2:
> The taxonomy of Foundation Agent Memory. Three orthogonal dimensions: Memory Substrate (§3.1), distinguishing internal memory (weights, latent states, KV caches; §3.1.2) from external memory (vector indices, text records, structural/hierarchical stores; §3.1.1); Memory Cognitive Mechanism (§3.2), including sensory (§3.2.1), working (§3.2.2), episodic (§3.2.3), semantic (§3.2.4), and procedural (§3.2.5); and Memory Subject (§3.3), separating user-centric (§3.3.1) from agent-centric (§3.3.2) perspectives. The bottom row shows representative application scenarios.
>
> > W3: Embodied AI agents are largely ignored
>
> We thank the reviewer for highlighting the importance of embodied agents. We would like to clarify that our survey primarily focuses on foundation agents, including emobdied agnets. The current scope is partly observed by the current literature landscape: only a limited number of embodied-agent papers explicitly place memory at the center of their contribution or use memory-related terms as core keywords, with representative examples such as KARMA. Nevertheless, we agree that embodied agents should be better acknowledged, and we will include additional embodied-agent works in the revised manuscript.
>
> Note that the current version already discusses embodied-agent evaluation, like success rate (SR) in Table 2 under accuracy-based metrics, and we also cover several open-ended embodied-agent benchmarks such as MineDojo, EmbodiedBench, and ALFWorld. Following the reviewer’s suggestion, we will further expand the evaluation section by adding more embodied-agent metrics, including path-length weighted success rate, navigation efficiency, trajectory completion, and interaction success.
>
> ## Requested Changes
>
> > R1: Define “foundation agent” clearly at the beginning of the Introduction
>
> We agree. The current definition appears in Section 2.1, but we will move a concise definition to the beginning of the Introduction:
>
> > In this survey, we define a foundation agent as an autonomous or semi-autonomous system whose core decision-making is driven by a foundation model, such as a large language model, vision-language model, or learned world model, and which is augmented with mechanisms for perception, reasoning, planning, tool use, action execution, and memory management.
>
> The more detailed discussion will remain in Section 2.1 as background.
>
> > R2: Clarify why “learning policy” contains “prompt-based learning” and what “prompt-based learning” means
>
> We agree that the term “prompt-based learning” may be not clear under the learning policy. To avoid ambiguity, we will rename this category to prompt-driven memory optimization policies. We will define it at the beggining of the section:
> Prompt-driven memory optimization refers to inference-time optimization strategy that use prompting, reflection, summarization, or self-generated feedback to update, organize, or retrieve memory mechanism without modifying model parameters.
> This will distinguish it from fine-tuning and reinforcement learning.
>
>
> We thank the reviewer again for the constructive comments and will incorporate the above revisions.

---

> > ### Comment · Reviewer_3UtD · 2026-04-28
> > **Let me know once the revision has been posted.**
> >
> > I appreciate the authors for detailed answer to my questions. When the revision is uploaded, I will take a close look at it.

---

> ### Author Response · Authors · 2026-05-02
> **Revision Update**
>
> Thanks for your time and response. We have incorporated all the requested changes in the updated revision (changed content marked as blue). We look forward to any further feedback from the reviewer.

---

### Author Response · Authors · 2026-06-21
**Follow up on the review progress**

Dear Reviewers and Action Editor,

We hope you are doing well. We would like to kindly follow up on our revised submission, which was uploaded on May 2 after incorporating the reviewers’ requested changes. The revised content has been marked in blue for ease of checking.

We sincerely appreciate the reviewers’ constructive feedback and the Action Editor’s time in handling our submission. We understand that the review process takes time, but we would be grateful for any update or further feedback when available.

Please let us know if there are any remaining concerns or additional revisions we should address.

Thank you again for your time and consideration.

Best regards,
The Authors

---

> ### Comment · Reviewer_3UtD · 2026-06-21
> **Sorry for the delay.**
>
> Dear authors,
>
> Due to my medical emergency, I have been suffering from reviewing the manuscript so far. I may be able to submit my review by June 30. Is that too late?
>
> Reviewer 3UtD

---

### Decision · Action_Editor_R9wU · 2026-06-29

**Recommendation:** Accept as is

**Additional Comments:**

The paper provides a timely survey of memory mechanisms for foundation agents. The paper offers a good coverage of memory architectures, evaluation, and future directions. While this is a fast-developing research area, this survey would be a great resource for summarizing past literature and inspiring future research.

**Audience:**

Yes

**Audience Explanation:**

The paper would be interesting to a wide range of audiences who work in foundation models and LLM agents.

**Claims And Evidence:**

Yes

**Claims Explanation:**

The paper clearly surveys foundation agent memory in three dimensions: memory substrate, memory cognitive mechanism, and memory subject. The paper also reviewed evaluation benchmarks and metrics for memory assessment. The paper has a good coverage of related literature.